# Cytoneme-mediated transport of active Wnt5b–Ror2 complexes in zebrafish

Chengting Zhang[1,2], Lucy Brunt[1,2], Yosuke Ono[1], Sally Rogers[1] & Steffen Scholpp[1✉]

Chemical signalling is the primary means by which cells communicate in the embryo. The underlying principle refers to a group of ligand-producing cells and a group of cells that respond to this signal because they express the appropriate receptors[1,2]. In the zebrafish embryo, Wnt5b binds to the receptor Ror2 to trigger the Wnt–planar cell polarity (PCP) signalling pathway to regulate tissue polarity and cell migration[3,4]. However, it remains unclear how this lipophilic ligand is transported from the source cells through the aqueous extracellular space to the target tissue. In this study, we provide evidence that Wnt5b, together with Ror2, is loaded on long protrusions called cytonemes. Our data further suggest that the active Wnt5b–Ror2 complexes form in the producing cell and are handed over from these cytonemes to the receiving cell. Then, the receiving cell has the capacity to initiate Wnt–PCP signalling, irrespective of its functional Ror2 receptor status. On the tissue level, we further show that cytoneme-dependent spreading of active Wnt5b–Ror2 affects convergence and extension in the zebrafish gastrula. We suggest that cytoneme-mediated transfer of ligand–receptor complexes is a vital mechanism for paracrine signalling. This may prompt a reevaluation of the conventional concept of characterizing responsive and non-responsive tissues solely on the basis of the expression of receptors.

Chemical signals are one of the main modes of intercellular communication in embryonic development. Signals are released by signal-producing cells in the form of secreted molecules called ligands. About a century ago, Spemann postulated that the target cells have the ability to react and respond to these signals by changing their cellular behaviour[1]. The ability to transduce such signals depends principally on the availability of the correct set of receptors. Consequently, cells and tissues lacking the appropriate receptors are considered non-responsive or blind to such signals. The concept of Spemann still dominates the current thinking that signal activation in responsive cells but not in non-responsive cells enables precise tissue specification, organ development and formation of the entire embryonic body[2].

During embryonic development, one of the most critical signalling systems is the Wnt signalling network, which comprises evolutionarily conserved and entangled pathways that regulate multiple processes that are crucial for embryogenesis and tissue homeostasis in all metazoans via an interplay between the Wnt proteins and their receptors and co-receptors[5,6]. Wnt–PCP signalling is one pathway of this network, and regulates cell polarization and migration[3,4]. In vertebrates, crucial components of the Wnt–PCP pathway include the Wnt5 ligands Wnt5a and Wnt5b[7], the seven-pass transmembrane Frizzled receptors[6] and the vital receptor tyrosine kinase pseudokinases Ror1 and Ror2 (ref. 8). Wnt5b–Fzd–Ror2 regulates Jun N-terminal kinase (JNK), Rac and RhoA signalling[9,10], and in zebrafish, Wnt5b–Ror2 governs the concomitant narrowing and lengthening of the embryonic body axis, the convergence and extension movement[11,12]. However, it is still unclear how the lipid-modified and membrane-associated Wnt5a and Wnt5b ligands,

which are produced at the embryonic margin, are transported over several hundreds of micrometres to cells of the overlying epiblast that express receptors to activate paracrine PCP signalling and thus control convergence and extension in the embryo.

Cytonemes are thin and actin-rich membranous protrusions that transport essential signalling components such as ligands and receptors between cells[13,14]. Similar to filopodia, cytonemes contain a Cdc42-dependent actin cytoskeleton, are highly dynamic, and can form and retract within minutes[15]. Their emergence is influenced by the cytoneme-producing cell and the extracellular space that they traverse. Cytonemes have been reported to transport signalling components of the Wnt signalling family in invertebrates and vertebrates[14]. The Wnt-bearing cytonemes transport Wnt8a over tens of micrometres in the zebrafish gastrula to regulate the patterning of the neural plate[15]. Similarly, Wnt3 is loaded on cytonemes in gastric cancer to facilitate tumour proliferation[16]. The emergence of Wnt-transporting cytonemes is affected by Wnt–PCP signalling in the source cell[17,18]. However, it is unclear how Wnt proteins loaded on cytonemes are handed over to the receiving cells to engage with their cognate receptors.

Here, we explore the dissemination of the Wnt–PCP signalling component Wnt5b and its receptor Ror2 in zebrafish gastrulation. First, we provide evidence that fibroblasts and epiblast cells in the zebrafish embryo form cytonemes decorated with Wnt5b and Ror2. Our studies further suggest that these ligands and receptors contribute to a complex handover to the receiving cell. Then, using in vivo quantitative imaging with single-molecule sensitivity, our data suggest that the cohesiveness of the Wnt5b–Ror2 pathway is maintained

[1]Living Systems Institute, School of Biosciences, Faculty of Health and Life Sciences, University of Exeter, Exeter, UK. [2]These authors contributed equally: Chengting Zhang, Lucy Brunt.
✉e-mail: s.scholpp@exeter.ac.uk

in the producing cells, during transport along cytonemes and in the receiving cell. Notably, the complex remains active during transport, regulates PCP–JNK signalling in the target cell, and influences the convergence and extension movement in the zebrafish, even when these cells lack functional Ror2 receptors. In summary, our results suggest that cytoneme-mediated transfer of the Wnt–PCP ligand–receptor complex is a form of paracrine Wnt–PCP signalling and challenges the typical categorization of tissues as responsive or non-responsive on the basis of receptor expression.

## Wnt5b–Ror2 is expressed on protrusions

Wnt5a and Wnt5b belong to the primary regulating ligands of Wnt–PCP signalling in vertebrate embryonic development[19]. However, it is unclear how these lipid-modified ligands are disseminated with precision in an embryonic tissue such as the zebrafish gastrula to regulate complex tissue movements such as convergence and extension. Therefore, we explored how Wnt5b (also known as pipetail) spreads between cells. Using an antibody against Wnt5a and Wnt5b[20] (Wnt5a/b), we detected the ligand on more than 65% of long protrusions of zebrafish fibroblasts (PAC2 cells) (Fig. 1a,b and Extended Data Fig. 1a,b,e). We also observed the localization of Ror2, a crucial receptor of Wnt5a/b[21], on cellular extensions (Fig. 1c and Extended Data Fig. 1c,d). Fluorescently tagged ligand Wnt5b co-localized with endogenous Ror2, and conversely, the fluorescently tagged receptor Ror2 co-localized with endogenous Wnt5a/b (Fig. 1d,e). Next, we over-expressed the tagged constructs of the ligand Wnt5b and the receptor Ror2, individually and in combination in PAC2 fibroblasts (Fig. 1f). We found Wnt5b–GFP and mCherry–Ror2 on protrusions, and detected co-localization of Wnt5b and Ror2 in puncta in the neighbouring cells. The number of puncta of Wnt5b–GFP, Ror2–mCherry and (Wnt5b–GFP)–(Ror2–mCherry) in the neighbouring Pac2 cells was significantly higher than puncta of the membrane marker mCherry, suggesting a specific handover of Wnt5b together with Ror2 signalling components (Extended Data Fig. 1f). We further characterized these protrusions and found that they also carry the filopodia tip markers unconventional MyoX and N-WASP[15] (Extended Data Fig. 1j,k). Next, we generated a Ror2-knockout PAC2 cell line using CRISPR–Cas9 (Extended Data Fig. 1l,m), which we co-cultured with wild-type PAC2 cells (Fig. 1g). To distinguish between $ror2^{-/-}$ cells from Ror2 wild-type cells, we used the prominent membrane localization of Ror2 (Fig. 1g and Extended Data Fig. 1c,d). We found that endogenous Ror2 localized to the plasma membrane in the wild-type cells, and identified Ror2 puncta in adjacent $ror2^{-/-}$ cells, suggesting that Ror2 protein produced in the wild-type cells is transferred to the $ror2^{-/-}$ cells (Fig. 1g and Extended Data Fig. 2a).

## Cytoneme-based delivery of Wnt5b–Ror2

To address the orientation of the transported ligand–receptor complex, we expressed Wnt5b–GFP, N-terminally GFP-tagged Ror2 (GFP–Ror2) or C-terminally tagged Ror2–GFP in fibroblasts and co-cultured these with cells stably expressing a secreted anti-GFP-nanobody coupled to mCherry (secVhh–mCherry[22]). After 24 h of co-culture, we found Wnt5b–GFP and the N-terminally GFP-tagged Ror2 co-localizing with secVhh–mCherry on cytonemes (Fig. 1h,i and Extended Data Figs. 1h,i and 2b). By contrast, the C-terminally tagged Ror2–GFP, did not co-localize with secVhh–mCherry (Fig. 1j). Next, we explored how Wnt5b–Ror2 is taken up in the receiving cell. We expressed GFP–Ror2 in PAC2 cells and exposed these cells to secVhh–mCherry to label membrane-bound Ror2. We then treated these cells with dynasore, a small molecule inhibitor of dynamin[23] and recorded the uptake of (GFP–Ror2)–(secVhh–mCherry) in the neighbouring PAC2 cells. After 2 h and 20 h treatments, we saw a reduction of (GFP–Ror2)–(secVhh–mCherry) in the receiving cells (Fig. 1k). Quantification of the Vhh–mCherry-positive vesicles in the neighbouring cells suggests that

GFP–Ror2 and Wnt5b–GFP are exposed to the extracellular space, are taken up at a similar ratio, and that uptake is blocked by inhibition of dynamin (Extended Data Fig. 2c). These results suggest that the ligand and the N-terminal part of the receptor face the extracellular side of the membrane during transport and that the ligand–receptor complex is taken up by dynamin-dependent endocytosis into the receiving cells (Fig. 1k and Extended Data Fig. 1g–i).

We then investigated the intercellular co-transport of Wnt5b and Ror2 in embryonic margin cells in the zebrafish embryos (Extended Data Fig. 2a,d,e). In the embryo, we observed the formation of a cytoneme, the handover of Wnt5b–GFP and Ror2–mCherry, and the subsequent retraction of the protrusion occurring within 4 min (Extended Data Fig. 2d and Supplementary Video 1). The transport of (Wnt5b–GFP)–(Ror2–mCherry) was accompanied by the independent membrane marker mem-BFP, suggesting a transfer of the cytoneme tip vesicle to the receiving cell. Next, we generated clones expressing Ror2–mCherry in a transgenic Rab5–GFP-expressing zebrafish line to visualize early endosomes (Extended Data Fig. 2f). At 6 hours post-fertilization (hpf), we found co-localization of Ror2–mCherry and Rab5–GFP in the producing cells, as well as in cells contacted by cytonemes; further, the uptake was reduced by co-expression of the dominant-negative dynamin Dyn2$^{K44A}$ (Extended Data Fig. 2f,g). These data support our early results indicating that dynamin function is involved in the uptake of the Ror2 into early endosomes of the neighbouring cells.

Our data suggest that Wnt5b and Ror2 are loaded together on signalling filopodia, hereafter referred to as Wnt–PCP-bearing cytonemes. Our findings further imply that Wnt5b–Ror2 is transported on cytonemes towards the tip. After contact formation, the Wnt5b–Ror2 is handed over to the neighbouring cell and endocytosed (Fig. 1l).

## Wnt5b–Ror2 complexes during transport

Given the unexpected finding that Wnt5b–Ror2 is translocated as a complex to neighbouring cells, we aimed to characterize the intermolecular interaction within this ligand–receptor complex during transport in the living zebrafish embryo using fluorescence lifetime imaging microscopy–Förster resonance energy transfer (FLIM–FRET) and fluorescence cross-correlation spectroscopy (FCCS). In the living zebrafish embryo, we measured the FRET efficiency of the glycosylphosphatidylinositol (GPI)-anchored fluorescent protein mem-GFP, and the anti-GFP nanobody Vhh–mCherry (Fig. 2a and Extended Data Fig. 3l) at the plasma membrane of mesenchymal cells of living zebrafish embryos as a positive control and compared these measurements to our negative control—mem-GFP and a cytosolic mCherry (cyto-mCherry) (Fig. 2b and Extended Data Fig. 3l). (mem-GFP)–(Vhh–mCherry) and (mem-GFP)–(cyto-mCherry) exhibited significant differences in GFP lifetime (2.24 ns and 2.41 ns, respectively), FRET efficiency (10.92% and 4.6%, respectively) and distance between fluorophores (79.34 Å and 92.7 Å, respectively) (Fig. 2g and Extended Data Fig. 3e). In parallel, we determined the cross-correlation of the same fluorescent pairs, (mem-GFP)–(Vhh–mCherry) and (mem-GFP)–(cyto-mCherry). We found a strong cross-correlation between mem-GFP and Vhh–mCherry (Fig. 2a and Extended Data Fig. 3l). For this positive control, we measured a dissociation constant ($K_d$) of 229 nM (Extended Data Fig. 3l), indicating a high binding affinity. We then determined the binding affinity of mem-GFP to cyto-mCherry as a negative control (Extended Data Fig. 3l). In this case, there was no cross-correlation and we calculated a $K_d$ value of more than 9,301 nM, indicating a low binding affinity.

Next, we measured the lifetime, the FRET efficiency and the distance between the donor Wnt5b–GFP and the N-terminally tagged acceptor molecule mCherry–Ror2. mCherry–Ror2 is biologically active (Extended Data Fig. 6g) and is able to partially rescue loss of Ror2 function in the embryo (Fig. 3a). We detected a robust signal at the plasma membrane of the producing cell, on cellular protrusions emitting from these source cells, and in the adjacent receiver cell (Fig. 2c–e). We

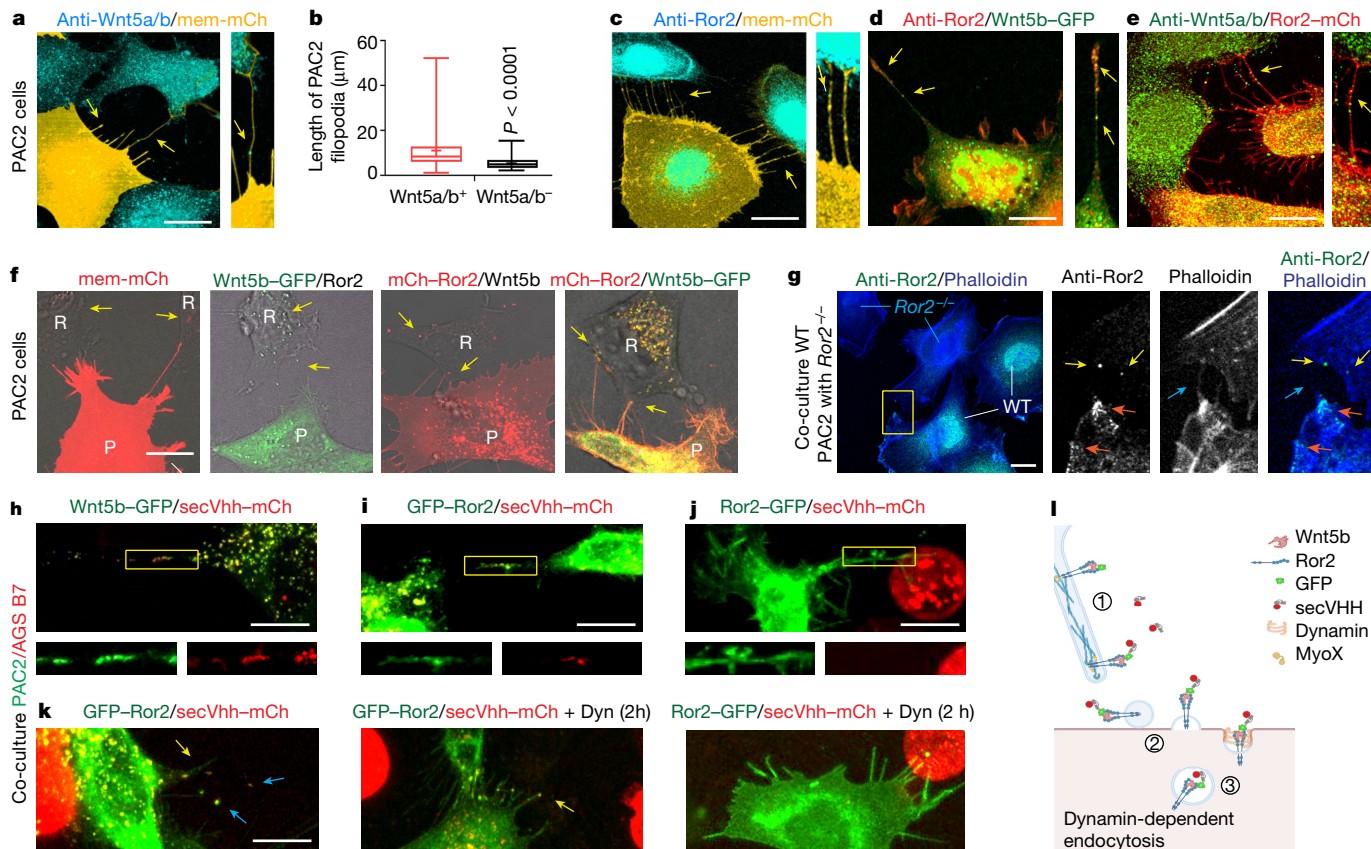

**Fig. 1 | Wnt5b–Ror2 complexes are transported from the producing cells to the receiving cells via cytonemes. a,c–e**, PAC2 cells transfected with mem-mCherry (**a,c**), Wnt5b–GFP (**d**) or Ror2–mCherry (**e**) and stained with an antibody against Wnt5a/b (**a,e**) or against Ror2 (**c,d**). Yellow arrows, Wnt5a/b–Ror2 on cytonemes. mCh, mCherry. Scale bars 10 µm. **b**, Length of filopodia with or without Wnt5a/b (*n* = 17 cells, 3 biological repeats). Two-tailed *P* values by Mann–Whitney test. Box and whisker plots show median, and top and bottom quartile ranges, with whiskers extending to minimum and maximum values. **f**, PAC2 cells were transfected with indicated markers and imaged live at 24 h post-transfection. Yellow arrows, Wnt5b (second from left) Ror2 (third from left) and Wnt5b–Ror2 (right) in the non-transfected neighbouring cells. R, receiving cells; P, producing cells. Scale bar, 5 µm. Left to right: *n* = 36, 38, 42 and 28 cells, 3 biological repeats. **g**, Left, wild-type (WT) PAC2 cells were co-cultivated with

*Ror2*[−/−] cells for 24 h, then stained with a Ror2 antibody. The area bound in yellow is magnified in the other images. Orange arrows, Ror2 puncta at the plasma membrane; yellow arrows, Ror2 puncta in the adjacent *Ror2*[−/−] PAC2 cells; blue arrows, filopodia. Scale bar, 10 µm. *n* = 46 cells, 3 biological repeats. **h–k**, PAC2 cells transfected with indicated constructs, co-cultured with AGS cells expressing secVhh–mCherry, PAC2 cells transfected with Wnt5b–GFP (**h**), GFP–Ror2 (**i,k**), Ror2–GFP (**j**) and treated with 40 µM dynasore (Dyn) (**k**). **h–j**, The region bound in yellow shows protrusions, and is magnified below the main image. *n* = 24 (**h**), 22 (**i**), 36 (**j**) cells, 3 biological repeats. **k**, Co-localization on cytonemes (yellow arrows) and in receiving cells (blue arrows). Left to right: *n* = 22, 38 and 9 cells, 3 biological repeats. Scale bars, 10 µm. **l**, Schematic of the handover mechanism: (1) transport along cytonemes; (2) deposition of cargo-positive vesicles; and (3) endocytosed vesicles.

measured a similar GFP lifetime of around 2.16–2.19 ns, FRET efficiency of around 11.3–13%, and distance of 75.7–75.0 Å between Wnt5b–GFP and mCherry–Ror2 in the producing cell, on cytonemes, and in the receiving cells (Fig. 2c–e,g and Extended Data Fig. 3e) (the FLIM–FRET assay records the distance between the fluorescent tags of the ligand and the receptor as a proxy for the distance between Wnt5b and Ror2). Using in vivo FCCS, we determined the $K_d$ values for Wnt5b–Ror2 in the producing cell, on cytonemes, and in the receiving cell to be 311–476 nM (Extended Data Fig. 3l). Similar to the FLIM–FRET analysis, we found an equal distribution of the individual $K_d$ measurements at the different subcellular localizations (Fig. 2h), suggesting that the ligand–receptor complex is maintained during translocation from the producing cell to the receiving cell. As a negative control, we measured FLIM–FRET with a GFP lifetime of 2.43–2.66 ns, FRET efficiency of 0–1.61%, and a non-detectable distance of Wnt5b to Ror2 lacking the Wnt binding domain (comprising the cysteine-rich domain[9] (CRD)) (Fig. 2f,g and Extended Data Fig. 3e). We also did not observe cross-correlation ($K_d$ > 6,678 nM; Extended Data Fig. 3l). As a further negative control experiment for FLIM–FRET, we measured the GFP lifetime, FRET efficiency and the distance between Wnt5b–GFP and a Ror2, which carries

a C-terminal mCherry tag. In this setting, the increased space between the fluorophores and the membrane between the fluorophores acting as an insulator should reduce FRET efficiency significantly (Extended Data Fig. 3m). Indeed, we measured a GFP lifetime of 2.40 ns, FRET efficiency of 3.6%, and distance between the fluorophores of 101.8 Å. On the basis of the in vivo FLIM–FRET and FCCS analysis, we infer that the Wnt5b–Ror2 complex is maintained during transport.

To challenge the integrity of the complexes, we over-expressed Wnt5b–GFP and untagged Ror2 together in the source cells and over-expressed Ror2–mCherry in the receiving cells. However, we found no cross-correlation between Wnt5b–GFP and Ror2–mCherry ($K_d$ = 3,560 nM; Extended Data Fig. 3n), suggesting that the (Wnt5b–GFP)–Ror2 complex stays largely intact, even if there is an excess of unbound Ror2 in the target cell. Finally, we set the binding affinity of mem-GFP-cyt-mCherry to 0 and that of (mem-GFP)–(Vhh–mCherry) to 100—Wnt5b and Ror2 had similar relative binding affinities at the different locations (Extended Data Fig. 3o).

These data indicate that Ror2 binds to Wnt5b with high affinity at the plasma membrane of the producing cell and that the structural integrity of the complex is maintained—as indicated by the proximity

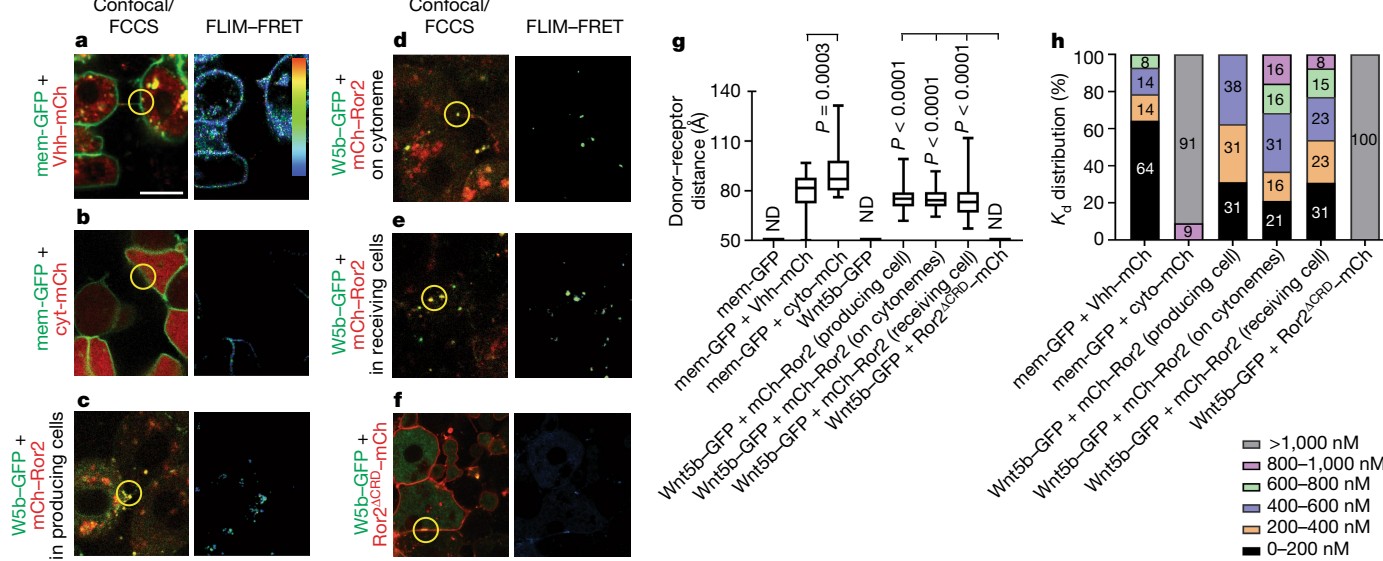

**Fig. 2 | In vivo FLIM–FRET and FCCS imaging reveal maintenance of Wnt5b–Ror2 complex cohesiveness during transport. a–f**, Wild-type zebrafish embryos were injected with the mRNA for mem–GFP plus Vhh–mCherry (**a**), mem–GFP plus cyt-mCherry (**b**), Wnt5b (W5b)–GFP plus mCherry–Ror2 in producing cells (**c**), Wnt5b–GFP plus mCherry–Ror2 in cytonemes (**d**), Wnt5b–GFP plus mCherry–Ror2 in receiving cells (**e**), or Wnt5b–GFP plus Ror2$^{\Delta CRD}$–mCherry (**f**) at the 8-cell stage in one blastomere to generate local clones and imaged live at 6 hpf. Representative fluorescence images and corresponding FLIM–FRET analysis are shown. Example spots for FCCS analysis are marked with yellow circles. Scale bar, 10 μm. The colour bar in **a** indicates FRET efficiency in FLIM–FRET analysis, along a range of blue (low) to red (high). **g**, Donor–receptor distance. Left to right: $n$ = 8, 47, 25, 21, 36, 23, 33 and 11 regions of interest in 3 biological repeats. One-way ANOVA test plus Tukey multiple comparisons test. $P$ values shown are in comparison to the corresponding negative controls indicated on the graph. Box and whisker plots show median, and top and bottom quartile ranges, with whiskers extending to minimum and maximum values. ND, not detectable. **h**, Distribution of Wnt5b–Ror2 binding affinities in the indicated bins. Left to right: $n$ = 14, 11, 17, 19, 13 and 7 filtered measurements in 3 biological repeats. The individual percentage is shown on the bar.

of fluorescently tagged constructs–during both transportation and subsequent uptake into the receiving cells.

## Wnt5b–Ror2 regulates cytoneme formation

To address the underlying transport mechanism, we generated zebrafish that lack either Ror1 or Ror2, as well as fish deficient for both receptors (Methods), as the two pseudokinases share many functions[24]. The phenotype of the *ror1* crispant embryos in the maternal–zygotic *ror2*[t13] mutant background (*ror1*[cr]/MZ*ror2*[t13]) is similar to the phenotype observed in the double-crispant zebrafish embryos *wnt5a/wnt5b*[cr] and in the double-mutant embryos for *wnt5b/wnt11*[25,26] (*wnt11* is also known as *silberblick*) (Fig. 3a and Extended Data Fig. 4e,g). Next, we tested whether Wnt5b–Ror2 signalling in the sender cells affects cytoneme formation[17,18] by overexpression of the indicated PCP constructs in clones in the zebrafish embryo (Extended Data Fig. 4l). We found that overexpression of Wnt5b, Ror2 and Wnt5b–Ror2 led to the formation of fewer but much longer filopodia (Fig. 3b and Extended Data Fig. 4l,m). We also detected a significantly reduced number of filopodia in the *ror1*[cr]/MZ*ror2*[t13] mutant (Fig. 3b and Extended Data Fig. 4l,m), whereas the length of filopodia was only slightly reduced in the MZ*ror2*[t13] mutant and in the *ror1*[cr]/MZ*ror2*[t13] mutant. Finally, we tested whether the formation of Ror2-dependent filopodia requires stabilization via the actin cytoskeleton. IRSp53, a multidomain I-BAR protein, promotes filopodia formation by linking the filopodia membrane with the cytoskeleton[27]; to block Wnt-PCP cytoneme formation, we used the mutated protein[15] IRSp53[4K]. The formation of the long cytonemes induced by Wnt-PCP is significantly reduced by co-expression of IRSp53[4K], independent of the expression of Wnt5b or Ror2 (Fig. 3b and Extended Data Figs. 4l,m, 5c–f and 7h). Filopodia formation is dependent on JNK signalling[28,29] and, specifically, the upkeep of Wnt8a cytonemes[18]; indeed, we observed the long Wnt5b–Ror2-positive cytonemes fragment minutes after treatment with the

JNK inhibitor SP600125 (Extended Data Fig. 5a,b and Supplementary Videos 2 and 3).

Next, we tested whether the Ror2-dependent emergence of filopodia would affect the distribution of Wnt5b. We found that the number of Wnt5b–GFP puncta on the membrane of the producing cells and in the neighbouring cells is significantly reduced in *ror1*[cr]/MZ*ror2*[t13] embryos (Fig. 3c,d and Extended Data Fig. 4q). Consistently, this phenotype is reversed if the Wnt5b–GFP expressing clones co-express Ror2. These experiments suggest that Wnt-PCP signalling is involved twofold: first, Wnt-PCP signalling induces long Wnt5b-Ror2-bearing cytonemes, which facilitates Wnt5b (and Ror2) spreading; second, long Wnt5b-Ror2-bearing cytonemes are stabilized through IRSp53 and Wnt-JNK signalling.

## Paracrine Ror2 activates JNK signalling

Next, we tested whether the transferred Wnt5b–Ror2 complexes maintain their activity in the target cells. We used the JNK kinase translocation reporter KTR–mCherry[30] (Extended Data Fig. 6b), and measured its cytoplasmic and nuclear localization as an indicator of JNK signalling strength in near real time[18] (Extended Data Fig. 6b,c). Small clones expressing mem-GFP do not interfere with JNK signal activation in the lateral mesoderm at 6 hpf (Fig. 4a); however, we found that clonal Wnt5b–Ror2 expression activates concentration-dependent paracrine JNK signalling (Fig. 4a,b). We further demonstrated that Wnt5b-positive clones–as well as Ror2-expressing clones–activate the JNK reporter (Extended Data Fig. 6d–h). Notably, local Wnt5b expression activated the JNK reporter only in the clonal cells and in a smaller halo around the clone compared with the Ror2-positive clones. This effect seems to be Wnt-PCP specific, as Wnt8a clones did not activate JNK above basal levels (Extended Data Fig. 6t,u). To test for cytoneme-dependent transport, we used the dominant-negative mutants IRSp53[4K] and Cdc42[T17N] (Fig. 4c,d and Extended Data Figs. 5c–f and 6l–n) and observed that

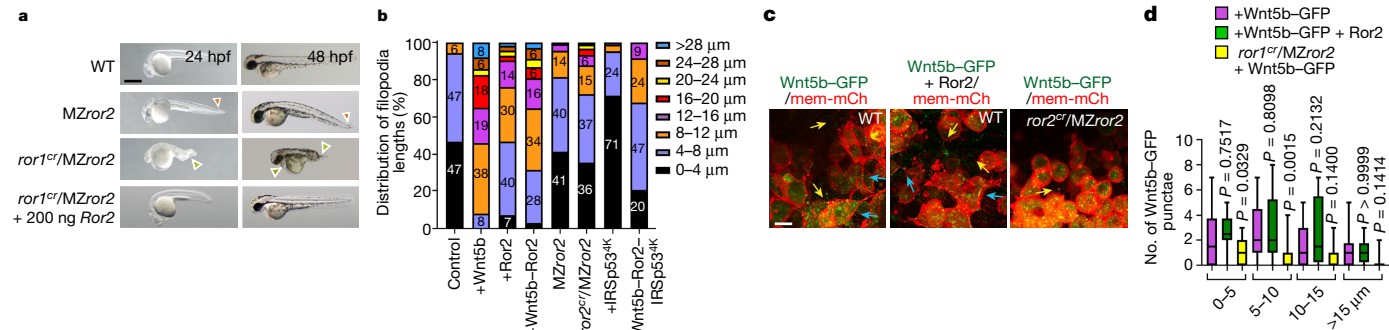

**Fig. 3 | Wnt5b–Ror2–JNK signalling promotes the formation of long cytonemes. a**, Morphology phenotypes for indicated mutant lines. *Ror2* mRNA (200 ng) was injected into the embryo as indicated. Arrowheads show abnormal tail morphology and heart oedema. Scale bar, 500 μm. **b**, Filopodia length in zebrafish embryos, split into indicated bins. Left to right: *n* = 367, 63, 113, 68, 312, 267, 41 and 59 measurements in 3 biological repeats. **c**, Zebrafish embryos were injected with indicated constructs and guide RNA (gRNA) and imaged at 6 hpf.

Yellow arrows, spread of Wnt5b–GFP; blue arrows, Wnt5b–GFP on the plasma membrane. Scale bar, 10 μm. **d**, Wnt5b–GFP puncta spreading from producing clone, split into indicated bins. Left to right: *n* = 8 (Wnt5b–GFP), 5 (Wnt5b–GFP + Ror2) and 8 (*ror1cr*/MZ*ror2* + Wnt5b–GFP) embryos, 3 biological repeats. Two-way ANOVA with Dunnett's multiple comparisons test. Box and whisker plots show median, and top and bottom quartile ranges, with whiskers extending to minimum and maximum values. *P* values are indicated.

reduction of cytonemes mediated by IRSp53[4K] or Cdc42[T17N] significantly reduced the ability of Wnt5b–Ror2-positive clones to activate paracrine JNK signalling. Our data indicate that cytonemes activate paracrine Wnt–PCP signalling but are dispensable for autocrine signalling (Extended Data Fig. 7). To test whether the transferred Wnt5b–Ror2 complex remains active in the target cells, we clonally activated Wnt5b and Ror2 in the *ror1cr*/MZ*ror2t13* mutant background and found a strong autocrine and paracrine JNK signalling (Fig. 4e,f), in contrast to clones expressing Wnt5b but not Ror2, which had significantly reduced activation of the JNK reporter. Similarly, overexpressing the dominant-negative form Ror2[3i]—which blocks the formation of homodimers and heterodimers[31,32]—reduces JNK signal activation (Extended Data Fig. 6i–k). Blockade of Ror2–PCP signalling by ubiquitous overexpression of Ror2[3i] can be overcome by paracrine Wnt–PCP from the clone (Extended Data Figs. 6i–k).

Next, we tested whether the downstream component Vangl2 (refs. 18,33) is required for Wnt–JNK activation triggered by the paracrine-transferred complexes. Wnt5b–Ror2-expressing clones were unable to activate the JNK reporter in the Vangl2 zebrafish mutant *vangl2m209* (Fig. 4g,h) and in cells expressing the dominant-negative Vangl2 mutant S5-17A/S76-84A (also known as Vangl2[10A]), in which 10 serine and threonine residues are replaced by alanine[18,33] (Extended Data Fig. 6o–q). These data suggest that Vangl2 is a mediator of Wnt–PCP signalling that is required for the activity of endogenous and transferred Wnt5b–Ror2 complexes in the zebrafish embryo.

To address whether endogenous Ror2 protein also activates paracrine Wnt–PCP signalling, we performed these experiments in the sensitized background of the MZ*ror2t13* mutant line. We found that embryonic margin cells—which express endogenous levels of Wnt5a, Wnt5b and Ror2—induce JNK activation in the MZ*ror2t13* mutant neighbouring cells (Extended Data Fig. 6a,v,w), whereas cells grafted from MZ*ror2t13* mutant embryos do not alter JNK signal activation around the clone (Extended Data Fig. 6v,w). Reduced cytoneme formation as a result of co-expressing IRSp53[4K] significantly reduced paracrine JNK signalling (Extended Data Fig. 6v,w). On the basis of these experiments, we conclude that Ror2 directly handed over by cytonemes is likely to be required for paracrine Wnt–JNK activation.

## Paracrine Ror2 represses β-catenin signalling

In parallel, we analysed the effects of autocrine and paracrine signalling functions of the same Wnt–PCP clones on endogenous gene expression in zebrafish gastrulation. In addition to activating JNK

signalling, Wnt–PCP signalling acts antagonistically to Wnt–β-catenin pathways[34,35]. We found that Wnt5b–Ror2 clones effectively suppress paracrine *lef1* expression (Extended Data Fig. 7a–f) if the clones are able to form cytonemes (Extended Data Fig. 7g–m). Furthermore, endogenous receptor function and dynamin-dependent endocytosis are not required to repress *lef1* expression (Extended Data Fig. 6r,s); however, Vangl2 is essential as an intracellular mediator in the receiving cells (Extended Data Fig. 7n–s).

Next, we addressed whether convergence and extension is affected by cytoneme-mediated dissemination of active Wnt–PCP complexes. We generated small clones in the lateral mesoderm expressing the indicated PCP constructs and quantified the width of the notochord in Tg(Gsc-GFP[CAAX]) embryos[36] (Fig. 5a–j and Extended Data Fig. 8a,c) at 10 hpf and the formation of the neuroectoderm at 32 hpf (Extended Data Fig. 8d–f). We found that clones expressing Wnt–PCP led to a significant broadening of the notochord anlage (Fig. 5a–j and Extended Data Fig. 8c) and interfered with convergence and extension by increasing the width of the neuroectoderm significantly in the developing embryo (Extended Data Fig. 8d–f). These Wnt5b–Ror2-induced defects were partially rescued by reducing cytoneme formation (Fig. 5c,e and Extended Data Fig. 8d).

Finally, we tested whether ectopic clones expressing Ror2 would rescue the convergence and extension defects observed in embryos lacking functional Ror receptors (Extended Data Fig. 4). We found that a clonal Ror2 source partially rescued the convergence and extension phenotype as well as the circularity of notochord cells in a double-crispant *ror1cr*/*ror2cr*, Tg(Gsc-GFP[CAAX]) zebrafish line at 10 hpf (Fig. 5g–j and Extended Data Fig. 8b,c). In addition, the ability of the clone to rescue the *ror1cr*/*ror2cr* phenotype was limited by blocking cytoneme formation via IRSp53[4K] co-expression (Fig. 5i). Notably, these embryos had two populations of cells: one that clustered at the midline, and another that remained in the lateral plate mesoderm, suggesting that the cells remaining at the lateral plate mesoderm require cytoneme-based transport of an appropriate Wnt–PCP signal for proper migration. Together, these results show that cytoneme-mediated Wnt5b–Ror2 dissemination influences convergence and extension in zebrafish development.

## Discussion

Wnt–PCP signalling orchestrates essential processes in vertebrate development, such as cell polarity and tissue migration. Wnt proteins act as autocrine and paracrine signalling proteins in these processes[37].

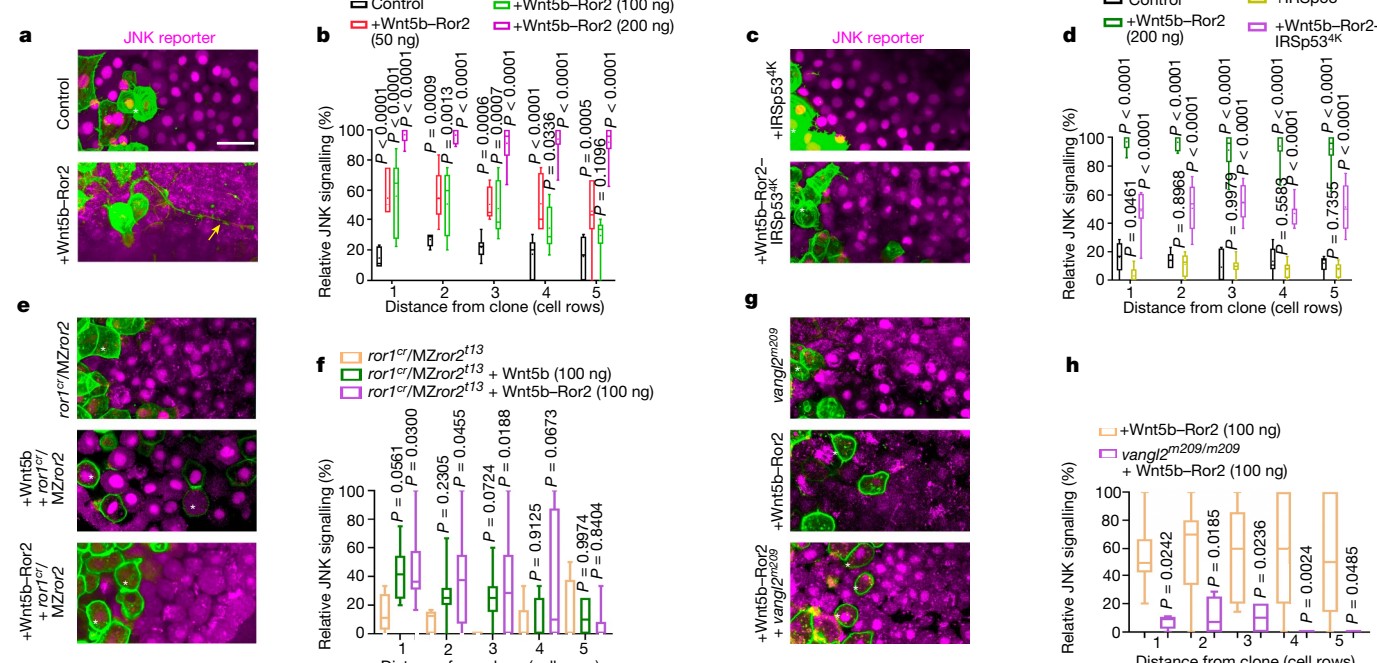

**Fig. 4 | Wnt5b–Ror2-expressing clones activate paracrine JNK signalling.** **a,c,e,g**. Wild-type (**a,c**), *ror1cr/ror2t13* (**e**) and *vangl2m209* (**g**) zebrafish embryos ubiquitously expressing the JNK reporter KTR–mCherry, with clones expressing Wnt5b-Ror2 (**a**), Wnt5b–Ror2-IRSp53⁴ᴷ or IRSp53⁴ᴷ(**c**), Wnt5b or Wnt5b–Ror2 (**e**), or Wnt5b–Ror2 (**g**) at the embryonic margin were imaged alive at 6 hpf. White stars, signal-producing cells; yellow arrow, an example of a cytoneme approximately 80 μm long. Scale bar, 20 μm. **b**, Relative JNK signalling within five cell rows distance of the clone when injected with different amounts of clonal Wnt5b–Ror2 in wild-type embryos. Left to right: n = 7 (control), 7 (50 ng), 12 (100 ng) and 12 (200 ng) embryos, 3 biological repeats. **d**, Relative JNK signalling within five cell rows distance of the clone when injected with indicated constructs. Left to right: n = 6 (control), 12 (Wnt5b-Ror2), 8 (IRSp53⁴ᴷ) and

9 (Wnt5b–Ror2-IRSp534K) embryos, 3 biological repeats. **f**, Relative JNK signalling within five cell rows distance of the clone when injected with clonal Wnt5b or Wnt5b–Ror2 in *ror1cr/ror2t13* embryos. Left to right: n = 5 (*ror1cr/MZror2t13*), 12 (*ror1cr/MZror2t13* + Wnt5b (100 ng)) and 9 (*ror1cr/MZror2t13* + Wnt5b–Ror2 (100 ng)) embryos, 3 biological repeats. **b,d,f**, Two-way ANOVA with Dunnett's multiple comparisons test. **h**, Relative JNK signalling within five cell rows distance of the clone when injected with clonal Wnt5b–Ror2 in wild-type and *vangl2m209* embryos. n = 15 (Wnt5b-Ror2) and 4 (*vangl2m209* + Wnt5b-Ror2) embryos, 3 biological repeats. Two-way ANOVA with Sidak's multiple comparisons test. Box and whisker plots show median, and top and bottom quartile ranges, with whiskers extending to minimum and maximum values. *P* values are indicated.

For example, WNT5B protein forms gradients to determine the spatial identity and influence the behaviour of target cells in the mouse limb bud[33]. However, it is still unclear how Wnt5b protein is transported through the tissue after lipid modification by Porcupine-mediated palmitoleation[38]. It has been reported that Wnt proteins are loaded on extracellular vesicles and activate Wnt signalling in target cells[39]. For instance, Wnt5b protein is loaded on extracellular vesicles that activate the invasion of breast cancer cells[40], and Wnt5b-loaded extracellular vesicles are produced in the choroid plexus to orchestrate cerebellar morphogenesis in mice[41] and also promote cancer cell migration and proliferation[42]. However, it is unclear how Wnt5b exhibits paracrine activity in a densely packed embryonic tissue such as the zebrafish gastrula. The zebrafish glypican Gpc4 (also known as knypek) has been suggested to control cell polarity during convergence and extension in zebrafish[35]. Recently, Gpc4 has been shown to be localized on signalling filopodia in zebrafish to aid Wnt–PCP signalling[43]. Here we show that Wnt5b is loaded on cytonemes, facilitating paracrine signalling. These signalling protrusions promote the exchange of lipophilic and membrane-tethered signalling components in zebrafish[14]. Similarly, the lipid-modified Wnt proteins Wnt8a and Wnt3 are disseminated by cytonemes in zebrafish and gastric cancer cells, respectively[15,16]. Furthermore, we find that the Wnt5b receptor Ror2 is also localized on cytonemes. Although we focus on the Wnt5b–Ror2 interaction during transport and signalling, interaction with the Fzd receptors is also likely to occur. In support of this idea, the Wnt receptor Fz in *Drosophila* and Fzd7 in chicks are also located on cytonemes and retrogradely transport active Wnt receptors[44,45]. As an expansion of the idea of

cytoneme transport[14], we show that Wnt5b binds to Ror2 in the producing cell and is then anterogradely transported along cytonemes to the receiving cells. We further observe that Ror2 promotes the formation of long cytonemes. These observations are consistent with data from *Drosophila*, in which the Hh co-receptor Interference hedgehog (Ihog) activates long cytonemes, which are stabilized—a prerequisite for signalling[46,47].

Intramolecular sensors utilizing FRET have been used for the analysis of conformational changes of Wnt signalling components such as Fzd5 and Fzd6[48,49] or Dishevelled[50] when exposed to Wnt5a. Here we developed this procedure to quantify ligand–receptor integrity during cytonemal transport in the living zebrafish embryo using FLIM–FRET. Our results indicate that the integrity of the complex is preserved during transport. Ligand–receptor affinities have also previously been quantified in the Wnt signalling network. For example, an in vitro binding assay based on biolayer interferometry[51] has suggested a binding affinity of Wnt5b to Fzd8 of about 36 nM, and NanoLuc Binary Technology (NanoBiT) and bioluminescence resonance energy transfer (BRET)-based analysis has suggested binding affinities of Wnt3a and Fzd receptors of below 10 nM in HEK 293 cells[52]. Here we used in vivo FCCS to quantify the strength of receptor–ligand interactions in the living zebrafish and measured $K_d$ values of $440 \pm 80$ nM for Wnt5b and Ror2. Differences in measuring approaches and accessory proteins in complex microenvironments could explain the differences in the $K_d$ values. Similar to our analysis, FCCS analysis in *Xenopus* suggests that Ror2 cross-correlates with Wnt5b, although the $K_d$ was not determined in that study[53]. More recently, surface plasmon resonance was used to

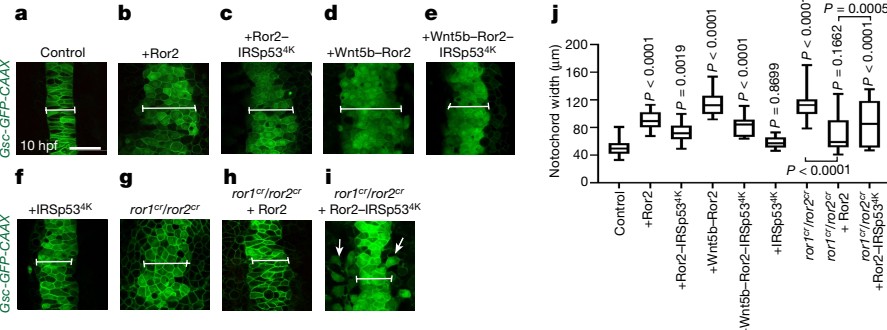

**Fig. 5 | Alterations of cytoneme-mediated dissemination of Wnt5b–Ror2 complexes affect convergence and extension in the zebrafish embryo.** **a**–**i**, Tg(*–6gsc:EGFP–CAAX*) zebrafish embryos were injected with control mRNA (**a**) or mRNA encoding Ror2 (**b**), Ror2–IRSp53[4K] (**c**), Wnt5b–Ror2 (**d**), Wnt5b–Ror2–IRSp53[4K] (**e**), IRSp53[4K] (**f**), or CRISPR–ribonucleoprotein (RNP) complexes targeting ror1 and ror2 without (**g**) or with mRNA encoding Ror2 (**h**) or Ror2–IRSp53[4K] (**i**), and imaged live at 10 hpf. Scale bar, 50 μm. The white line indicates the notochord width. White arrows mark cells that remain located lateral to the midline. **j**, The notochord width was measured in embryos treated as in **a**–**i**. Left to right: *n* = 20, 13, 12, 8, 12, 9, 6, 8 and 10 per embryo, 3 biological repeats. One-way ANOVA plus Tukey multiple comparisons test. Box and whisker plots show median, and top and bottom quartile ranges, with whiskers extending to minimum and maximum values. *P* values are indicated.

determine the binding affinity of the co-receptor RYK (a homologue of Drl) to DWnt-5 immobilized on a sensor chip[54]. The mean $K_d$ for binding of the extracellular region of Drl to immobilized DWnt-5 across multiple repeats was 720 ± 160 nM and, thus in accordance with our results in the zebrafish.

In our study, the binding affinity of Wnt5b to Ror2 was not altered during transport of the complex from the producing cells to the receiving cells, indicating high structural integrity of the ligand–receptor complex. This analysis does not diminish the importance of Fzd receptors during PCP signalling—it remains highly likely that Fzds are also part of this complex, and influence the binding affinity of Wnt5b to Ror2.

We further report that Wnt5b–Ror2 is transported on the extracellular side, as indicated by Vhh–mCherry binding to the accessible GFP tag on these proteins. This is in accordance with recently published data indicating that externally applied proteinase K decreases the Wnt5a content of extracellular vesicles[55]. Following the handover, we observed Wnt5b–Ror2 at the membrane of the receiving cell, followed by uptake via dynamin-dependent endocytosis, as reported previously[32].

We observed active signalling by Wnt5b–Ror2 complex in the target cells following transport. The dissemination of activated receptors on extracellular vesicles has been reported for the cytokine receptor CCR5, and the co-receptor for macrophage-tropic HIV-1[56]. Furthermore, in human glioma cells EGFRvIII is distributed by extracellular vesicle-based transfer to activate other membrane-associated oncogenic tyrosine kinases such as HER2, KIT and MET[57]. Our group recently demonstrated that ROR2 and WNT5A are significantly correlated on cytonemes of human cancer-associated fibroblasts; in the same study, ROR2 was identified in the cytoneme-receiving gastric tumour cells influencing migration of gastric cancer cells[58], suggesting a conserved signalling role in vertebrate embryogenesis and disease.

One of the most basic principles of embryogenesis involves chemical signalling between different groups of cells. The sending cells produce specific ligands, and the responding cells follow Spemann's principle of the 'ability to react' by presenting the specific signal transduction components, a suitable set of transcription factors, and chromatin accessibility. However, the presence of the appropriate set of ligand-specific receptors at the top of the pathway remains the essential prerequisite for signal activation. Here we propose a novel mechanism of cell–cell communication in embryogenesis whereby cytonemes carry active ligand–receptor complexes to the target cell. This mechanism may be more effective than diffusion-dependent mechanisms in regulating the behaviour of the receiving cell because the producing cell can target specific cells directly via cytonemes and control the signalling strength by adjusting the number of ligand–receptor pairs delivered by the cytonemes. However, the availability of appropriate receptors in the target cells may be less relevant in this context, as it still requires the presence of the appropriate downstream signalling cascade. Overall, this work expands our knowledge about modes of chemical signalling in embryogenesis.

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

## Methods

### Plasmids

Plasmids that were used for transfection, to generate mRNA for injecting into zebrafish embryos, and to generate probes for in situ hybridization experiments are as follows: mem-GFP in pCS2+, xRor2[3i] in pCS2+, xRor2-mCherry in pCS2+ and xRor2 in pCS2+[17], dynamin[K44A] in pCS2+, Wnt8a in pCS2+, cytosolic mCherry in PCS2+. mem-GFP was amplified and cloned via a BamHI and a XbaI site to generate mem-BFP in pCS2+. The open reading frame of xRor2 was amplified and ligated into pCS2+ mCherry using GeneArt Gibson Assembly HiFi Cloning Kit to make mCherry-xRor2. The open reading frame of zebrafish *Ror2* was amplified and cloned via ClaI and XbaI sites into pCS2+ mCherry to generate zfRor2-mCherry. To generate zfRor2-ΔCRD-mCherry, the amino acids from 170 to 304 were deleted from zfRor2 open reading frame and cloned into pCS2+. secVhh−mCherry was subcloned via a ClaI and a SnaBI site into pCS2+. GPI-anchored mCherry was cloned into pCS2+ (mem-mCherry)[59]. The open reading frame of zebrafish *Wnt5b* was amplified and cloned via a BamHI and an XbaI site to generate zfWnt5b-GFP in pCS2+. Irsp53[4K] (ref. 18), Cdc42[T17N] (ref. 15) and pPBbsr-JNK KTR-mCherry[60] were gifts from K. Aoki (Addgene plasmid #115493), and we subcloned them into pCS2+ via ClaI and SnaBI sites. The antisense probes against *lef1*, *ntl* and *pax6a* were used as previously described[17].

### Transfection and CRISPR–Cas9 knockout

PAC2 zebrafish fibroblasts were maintained at 28 °C without $CO_2$ in Leibovitz-15 medium (Gibco, 11415056). Stable secVhh−mCherry AGS cells were maintained at 37 °C with 5% $CO_2$ in RPMI medium (Gibco). All cell lines were tested regularly for mycoplasma by endpoint PCR testing every 3 months and broth tests every 12 months. PAC2 and AGS cells were trypsinized and seeded on glass-bottom 35-mm dishes for live imaging or on coverslips in 6-well plates for fixation. After 24 h, cells were transfected with relevant plasmids using Fugene HD Transfection Reagent (Promega, E2312) and incubated at 28 °C for 24 h. For co-culture, PAC2 and AGS cells were trypsinized 24 h after PAC2 transfection and reseeded together on 35-mm glass-bottom dishes for 24 h. Live cells were imaged on the Leica SP8 using the 63× water objective. To generate CRISPR-knockout PAC2 cells, 50 μM of *Ror2* gRNA was generated from 100 μM custom *Ror2* CRISPR RNA (crRNA) (custom sequence: TACAACTGGAGCTCATCTGG, IDT DNA) and 100 μM Alt-R CRISPR–Cas9 *trans*-activating crRNA (tracrRNA) (IDT DNA, 1072532) and heated to 95 °C for 5 min and cooled to room temperature. Fifty micromolar *Ror2* gRNA was then incubated for 10–20 min with 40 μl nucleofector solution (Lonza P2 Primary cell 4D X kit L, V4XP-2024) and 20 μM EnGen Cas9−NLS enzyme (NEB, M0646T) to form the RNP complex. Two-hundred thousand PAC2 cells were centrifuged for 10 min at 1,200 rpm and washed in PBS, followed by 10 min centrifugation at 1,200 rpm. Cells were resuspended in nucleofector solution and combined with the RNP complex, PBS, and 100 μM electroporator enhancer (1075915, IDT DNA) for 100 μl total volume. Next, 100 μl was transferred to a cuvette (Lonza P2 Primary cell 4D X kit L, V4XP-2024) and electroporated using a Lonza nucleofector. Next, 300 μl pre-warmed Leibovitz-15 medium was added to the cuvette and transferred to 2 ml pre-warmed medium in a 6-well plate and incubated at 28 °C for 48 h. For sequencing, DNA was extracted from cell pellets (GENEJET genomic DNA Purification kit, K0721, ThermoFisher Scientific), and the PCR product was amplified around the gRNA target site (forward primer: CACACTTGAGACTTTGGGGGA; reverse primer: GGTGTAAAATCCT TACCTGC, Eurofins; PCRBIO, PCR Bio taq mix red, PB10.13-02). PCR products were sent for Sanger sequencing (Eurofins, TubeSeq Service).

### Immunostaining of PAC2 fibroblasts

PAC2 zebrafish fibroblasts were seeded on coverslips in six-well plates and transfected as above. After 24 h, cells were fixed in 0.25%

Mem-Fix[20] (0.1 M Sorensen's phosphate buffer (pH 7.4), 4% formaldehyde, 0.25% glutaraldehyde) for 10 min at 4 °C. Cells were washed 2× 5 min in Sorensen's buffer and permeabilized in goat permeabilization buffer (0.1% Triton X-100, 5% goat serum, 0.2 M glycine, 1× PBS) for 1 h at room temperature. Appropriate primary antibodies were used at 1:50 dilution in goat incubation buffer (0.1% Tween-20, 5% goat serum, 1× PBS). Primary antibodies used were: WNT5A-B, rabbit polyclonal antibody, ProteinTech, 55184-1-AP; and rabbit monoclonal antibody, Cell Signaling Technology (CST), 88639S. Thirty microlitres of primary antibody in incubation buffer was placed on parafilm in a humidity chamber, and coverslips were placed cell-side down onto the buffer. Cells were incubated in primary antibody overnight at 4 °C. Coverslips were placed cell-side up in 6-well plates and washed 6× 5 min in PBS. Appropriate secondary antibodies (Goat anti-rabbit IgG H&L Alexa Fluor 488, ab150077, Abcam) were prepared at 1:1,000 dilution in goat incubation buffer. Thirty microlitres was placed on parafilm in a humidity chamber, and coverslips were placed cell-side down onto the buffer for 1 h at room temperature. Coverslips were then placed cell-side up in 6-well plates and washed 7× 10 min in PBS, then 1× in MilliQ and 1× in 1× PBS with 0.05% Tween-20. Coverslips were mounted on slides using ProLong Diamond (Invitrogen) and left in the dark for 24 h before imaging. Slides were imaged on the Leica SP8 using the 63× water objective.

### Zebrafish husbandry

Wild-type EZ9216B, Tg*(Rab5c:GFP)*, *ror2[t13]*, *vangl2[m209]* and Tg*(−6gsc:EGFP −CAAX)*[36] zebrafish (*Danio rerio*) were maintained at 28 °C on a 14 h light/10 h dark cycle[18]. All zebrafish husbandry and experimental procedures were followed and conducted under personal and project licences granted by the UK Home Office under the United Kingdom Animals Scientific Procedures Act (ASPA) and following ethical policies approved by the University of Exeter's Animal Welfare and Ethical Review Body (AWERB). All the work with zebrafish was carried out before animals became capable of independent feeding, here at 5 dpf or younger, per ASPA.

### Microinjection of mRNA

All the plasmids in this article were firstly linearized with corresponding New England Biolabs (NEB) restriction enzymes. Then, capped sense mRNA was generated by in vitro transcription from linearized plasmids using Invitrogen Ambion mMessage mMachine SP6 Transcription kit. For different experiment purposes, zebrafish embryos at the 1-cell (ubiquitous expression) to 16-cell stage (clonal expression) were injected with 1 μl of different concentrations of mRNA. To generate clonal expression, Invitrogen Dextran, Fluorescein, and Biotin, 10000 MW (mini-Emerald) was co-injected with mRNA to the label-producing or receiving cells.

### FCCS

FCCS was used to determine the binding affinity of molecules, quantified by the equilibrium dissociation coefficient, $K_d$. Based on the calibration measurements of the in vivo FCCS measuring system, fluctuation recordings were performed in a predefined volume of $0.65 \times 10^{-9}$ nm$^3$ with a recording time of 10 s (Extended Data Fig. 3f–k). Before the FCCS measurements, the GFP channel (excited at 488 nm) and the mCherry channel (excited at 587 nm) were calibrated to determine the effective volume ($V_{ef}$). ATTO 488 and ATTO 565 dyes (Sigma Aldrich) with a known diffusion coefficient[61,62] of 400 μm$^2$ s$^{-1}$ were used. ATTO 488 was diluted to 3 nM and 6 nM to measure the auto-correlation in the GFP channel. ATTO 565 was diluted to 4 nm and 8 nm to measure the auto-correlation in the mCherry channel. Finally, the effective volume for cross-correlation ($V_{cc}$) is determined as follows[63]:

$$V_{cc} = \frac{\pi^{3/2}}{\frac{\omega_g^2 + \omega_r^2}{2} \sqrt{\frac{z_g^2 + z_r^2}{2}}}$$

Embryos were injected in 1 out of 8- to 16-cell blastomeres with a low concentration of mRNA (50 to 100 ng μl$^{-1}$). For FCCS measurements, the expression level has to be as low as possible. When the embryos were at 50% epiboly (6 hpf), the live embryos were mounted in one of these cavities in 30 μl of 0.7% low-melting-temperature agarose and covered with a no. 1.5 coverslip. Tape was used on both sides to stabilize the coverslip. The cross-correlation was measured by a Leica Sp8 FCS module equipped with FALCON single-molecule detection unit. Each measurement lasted 10 s. The measurement procedure is illustrated in Extended Data Fig. 3.

For the measurement of the auto-correlation, the corresponding $V_{ef}$ in each channel was applied ($V_{ef}$ of GFP is 0.56 fl; $V_{ef}$ of mCherry is 0.75fl), and the LAS_X model of 'diffusion with triplet' was used for fitting because the tagged protein GFP and mCherry are at triplet state. Vcc 0.65fl was applied for the cross-correlation fitting, and the 'pure diffusion' model was selected. The LAS_X software determined this method for $V_{cc}$, as it included the triplet stage. All measurements were consistent with the calibration settings. The dissociation constant ($K_d$) was determined based on these fitting values for every measurement. To calculate the concentration of cross-correlation molecules: the number of molecules in the GFP focal volume is $N_1$; the number of molecules in the mCherry focal volume is $N_2$; the number of molecules in the cross-correlation channel is $N_{cc}$, the effective volume is $V_{cc}$, and Avogadro's constant is $N_A$. Based on these values from the fitting algorithm, the concentration in cross-correlation molecules was calculated as follows:

$$C_{cc} = \frac{N_1 \times N_2}{N_{cc}} \times \frac{1}{V_{cc} \times N_A}$$

The concentration of molecules in the GFP channel is $C_{green}$ and the concentration of molecules in the mCherry channel is $C_{red}$. The $K_d$ was calculated as

$$K_d = \frac{(C_{green} - C_{cc}) \times (C_{red} - C_{cc})}{C_{cc}}$$

We excluded measurements in which the molecular concentration for each channel was over 2,000 nM.

## FLIM–FRET

To describe the complex with a high spatial and temporal resolution with reference to the differentially tagged fluorescent components, FLIM–FRET was used, because it is independent of the fluorophore concentration, the excitation efficiency, and the effect of light scattering—a prerequisite for analysis in vivo. The energy transfer from GFP (the fluorescent donor) to mCherry (acceptor) was measured. High FRET efficiencies and short donor lifetimes indicate an energy transfer between the GFP donor and mCherry acceptor, and that tagged molecules are at distances <10 nm (Extended Data Fig. 3a–d). Consistently, FRET efficiency decreases, and the fluorescence lifetime of the donor (tagged GFP) increases when the tagged molecules are at distances >10 nm.

Embryos were injected in one blastomere with mRNA at the 8-cell to 16-cell stage. For FLIM–FRET experiments, we firstly injected the donor only to measure the donor's lifetime. Then the donor and acceptor were co-injected to perform the FRET analysis. Injected 50% epiboly embryos were mounted in a plastic 30-mm dish with 0.7% low-melting-temperature agarose. They were scanned with a Leica Sp8 FLIM module. The optical sections for all channels were of equal thickness, and for embryo scanning, each section was 2 μm. The FLIM–FRET data were acquired by excitation at 488 nm. Line repetition was set to 4 to collect sufficient photons.

The data were analysed by using the LAS_X_Single Molecule Detection unit. The fitting model 'multi-exponential donor' was selected for

FRET analysis. The mean value of donor-only lifetime was applied to the unquenched donor lifetime to do the analysis. The Förster distance[53] for the EGFP–mCherry pair is on the order of 52.4 Å, which was also applied to the system. Different regions of interest (ROIs) were selected and analysed. The software calculated the mean lifetime, FLIM–FRET efficiency, and donor–acceptor distance.

## Generation of knockout zebrafish lines

Preparation of gRNA: 1.2 μl *Ror2* gRNA from 100 μM custom *Ror2* crRNA (custom sequence: CATATATTGAGGATTACAAC, IDT DNA), 1.2 μl 100 μM Alt-R CRISPR–Cas9 tracrRNA (IDT DNA, 1072532) together with 7.6 μl duplex buffer (IDT DNA) were heated to 95 °C for 5 min and cooled to room temperature. 2.5 μl prepared gRNA, 1.25 μl EnGen Cas9–NLS enzyme (NEB, M0646T), 2 M KCl, and 0.5 μl phenol red were mixed to form the RNP complex. Wild-type zebrafish embryos were injected with the RNP complex at the one-cell stage. The injected F$_0$ embryos were maintained at 28 °C on a 14 h light/10 h dark cycle. When the F$_0$ zebrafish were 5 months old, the zebrafish were fin-clipped under the standard protocol. For genotyping, the DNA was extracted from individual clipped fins using 50 mM NaOH, heating at 95 °C for 15 min. The PCR product was amplified around the gRNA target site (forward primer (5′–3′: TTTTTGTTTTGCAAACGAA; reverse primer 5′–3′: CAGTGTTTAATTGTTACAGC), Eurofins; PCRBIO, PCR Bio taq mix red, PB10.13-02). PCR products were sent for Sanger sequencing (Eurofins, TubeSeq Service). The sequence data were analysed using Poly Peak Parser (http://yosttools.genetics.utah.edu/PolyPeakParser/) to identify the Ror2 mutant zebrafish (see Extended Data Fig. 4). The identified heterozygous zebrafish were out-crossed with wild-type zebrafish to obtain F$_1$ zebrafish embryos. When the F$_1$ zebrafish were five months old, the same genotyping method was carried out again to select for heterozygous zebrafish with same mutation site. Selected F$_1$ zebrafish were then in-crossed to obtain the F$_2$ zebrafish. During this process, a zebrafish line has been identified with a 2-bp deletion and 18-bp insertion leading to a premature stop codon in exon 2 (Extended Data Fig. 4a,b). Thus, the Ror2 protein sequence was changed at Threonine 13 (*t13*), leading to a frameshift and a new stop codon after position 38. Therefore, the mutant was termed *ror2$^{t13}$*, which lacks the extracellular Ig-like domain, the CRD domain, the transmembrane domain, and the tyrosine kinase domain. Finally, the F$_2$ homozygous zebrafish were in-crossed to obtain a stable and maternal zygotic Ror2-knockout fish line. The maternal-zygotic mutant (MZ*ror2$^{t13}$*) shows a slight widening of the notochord at 10 hpf (Extended Data Fig. 4c,d,n) and is viable, fertile, and shows a mild phenotype of a slight upwards bend at the tail tip (Fig. 3a and Extended Data Fig. 4e).

In parallel, the following F$_0$ crispant embryos (cr) were generated, *ror1$^{cr}$*, *ror2$^{cr}$*, *wnt5a$^{cr}$* and *wnt5b$^{cr}$*. Crispant larvae were generated by injecting, as above, the following gRNAs at 1-cell stage: *Ror1* (ror1-AE: CCGTGGCTCCTGAACCACAGGGG; ror1-AG: TATGGCACAGTGTCAACCACAGG), *Wnt5a* (wnt5a-AE: AGATCGTGGACGCAAACTCA; wnt5a-AF: CGTCGACAACTCCACAGTGC), and *Wnt5b* (wnt5b-AD: AGGTGGAAAGCTCACCCTCA; wnt5b-AE: GAACCAAGGACACCTACTTC). crRNAs were obtained from IDT. For example, *ror1$^{cr}$*/MZ*ror2$^{t13}$* mutant embryos were generated, which showed many typical features of a Wnt–PCP phenotype, including a wider and shorter axial mesoderm leading to a shorter body axis (Extended Data Fig. 4c,d, arrows), malformation of the trunk and tail, and heart defects (Fig. 3a and Extended Data Fig. 4e,j,k). In addition, the expression domain of the Wnt–β-catenin target gene *lef1* is also broader in the *ror1$^{cr}$*/MZ*ror2$^{t13}$* mutant embryos (Extended Data Fig. 4o,p). The phenotype of the *ror1$^{cr}$*/MZ*ror2$^{t13}$* embryo-larvae is similar to the phenotype observed in the zebrafish double-crispant *wnt5a/wnt5b$^{cr}$* (Extended Data Fig. 4g,h). The phenotype of the *ror1$^{cr}$*/MZ*ror2$^{t13}$* embryos is partially rescued by injection of *Ror2* mRNA (Fig. 3a and Extended Data Fig. 4f). Notably, a rescue of the phenotype of the double *wnt5a/wnt5b$^{cr}$* embryo by microinjection of *Ror2* mRNA was not

possible, suggesting that the receptor requires the ligand for signal activation (Extended Data Fig. 4i).

## Transplantation assay

Wild-type or $ror2^{t13}$ zebrafish embryos were ubiquitously injected with membrane marker mem-GFP; $ror2^{t13}$ embryos were ubiquitously injected with JNK reporter KTR–mCherry. At 3 hpf, the cells from the mem-GFP-positive donor embryo were transplanted into the $ror2^{t13}$ host embryos. At 5 hpf, the transplanted $ror2^{t13}$ host embryos were mounted, and the transplanted cell clones were imaged.

## Measurements of filopodia

Membrane-marked membrane protrusions were defined as filopodia as soon as they reached a length and width of 1 μm. Zebrafish embryos were injected mRNA together with a membrane marker in 1 blastomere at the 8-cell to 16-cell stage. Injected embryos thus generated fluorescently labelled cell clones for visualization of filopodia. Numbers of filopodia per cell were manually counted. Lengths of filopodia were measured from the base to the tips of protrusions in FIJI. At least ten different embryos, and in each embryo, at least ten isolated cells were measured.

## Inhibitor treatment

Zebrafish embryos at 6 hpf or 9–10 hpf were mounted in 0.7% low-melting-temperature agarose in 35-mm dishes. During the JNK inhibitor assay, mounted 6 hpf zebrafish embryos were treated with 40 μM JNK inhibitor (SP600125[18,64]). Embryos were imaged with a Leica TCS SP8 confocal microscope using a 63× dip-in objective. All the images were obtained from confocal z-stacks of living embryos. Stable secVhh–mCherry-expressing AGS cells and transfected PAC2 cells were co-cultured on glass-bottom dishes as described above. Dynasore (40 μM; abcam, ab120192) was added to the medium, and cells were imaged with a Leica TCS SP8 confocal microscope at 2 or 20 h after drug application.

## JNK reporter KTR–mCherry assay in vivo

The embryos were injected at the 1-cell stage with 250 ng μl$^{-1}$ mRNA KTR-mCherry and kept in a 28 °C incubator for 50 min. Later, we injected mRNA together with a membrane marker mem-GFP into 1 out of 8 or 16 cell blastomeres to generate clonal expression. When the embryos developed at 50% epiboly, the live sources were mounted and imaged. Cells that expressed GFP were considered producing cells. We drew a border to distinguish between producing and receiving cells. Cells without nuclear KTR–mCherry expression were considered active for JNK signalling. Paracrine JNK signalling activation was analysed up to a distance of five cells from the clone.

## Zebrafish in situ hybridization

*lef1*, *ntl* and *pax6a* digoxigenin probes were generated using a Roche RNA labelling and detection kit and then purified through ProbeQuant G50 Micro Columns[3].

Microinjected embryos with mRNA and mini-Emerald were collected at 5 hpf and 30 hpf and fixed with 4% PFA. In situ hybridization experiments were carried out as described[17]. In addition, we carried out double staining in these experiments to identify the producing cells. After *Lef1* and *Pax6a* were labelled, the embryos were fixed again and incubated at 70 °C for 1 h to deactivate the first probe. Subsequently, we treated these embryos with anti-FITC to bind the mini-Emerald. Finally, Fast-Red was used to label the clones a different colour. Double-stained embryos were kept in 70% glycerol for further analysis.

In situ stained embryos were imaged with an Olympus SZX16 stereomicroscope equipped with a DP71 digital camera. Images were taken using the Cell D imaging software. Embryos stained for *lef1* were imaged with the animal pole up. To analyse the Wnt5b–Ror2 function, we categorized the embryos according to normal expression, mild expression, or no

expression. Embryos stained for *pax6* expression were deyolked and flat-mounted in a dorsal view. The length of the forebrain and the width of the hindbrain were measured in FIJI. Based on the midline position and expression of *pax6a* in the hindbrain, we grouped them into three categories: normal phenotype, mild phenotype, and severe phenotype. The length and width of *ntl* expression domains were measured in FIJI and the width/length ratio was determined.

## Fluorescence Intensity of antibody

Images of fixed PAC2 cells were processed for α-Ror2 antibody or α-Wnt5a/b antibody and imaged on a Leica TCS SP8 confocal microscope with the same saved settings for each condition. In FIJI, the ROI of a cell from a 'no primary' sample or wild-type non-transfected and transfected cells from the same sample slide were taken, and fluorescent intensity was measured.

## Cell extension and convergence assay

Wild-type zebrafish embryos were injected with mRNA and mini-Emerald in 1 out of 8 to 16 cell blastomeres to generate a clonal expression. Tg(−6gsc:EGFP-CAAX) zebrafish[36] embryos were injected with mRNA in 1 out of 8 to 16 cell blastomeres to generate a clonal expression. Then, at 9 hpf, the embryos were mounted in a dorsal view for fluorescence imaging. First, the maximum width and length of a cell were measured in FIJI. In the meantime, the width of the notochord was also measured. Finally, the circularity was calculated to indicate the shape of the individual cell in the notochord. For the in situ hybridization analysis, the embryos were collected and fixed with 4% PFA for further analysis after 30 h.

## Statistics and reproducibility

Statistical analysis was carried out using GraphPad Prism 9.0. Independent Student's *t*-tests were used to test for differences between two groups for normal data, and Mann–Whitney U tests were used for non-parametric data. One-way or two-way ANOVA with appropriate multiple comparison tests were used for normally distributed data. The Kruskal–Wallis test with a Bonferroni correction for multiple comparisons was used for non-parametric data. Each experiment was repeated three times independently.

## Reporting summary

Further information on research design is available in the Nature Portfolio Reporting Summary linked to this article.

## Data availability

Microscopy data reported in this paper and any information required to re-analyse the data reported in this paper are presented within the paper or are available from the corresponding author upon reasonable request. Source data are provided with this paper.

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

**Acknowledgements** C.Z. is supported by a Chinese Scholarship Council (CSC) studentship. Research in the Scholpp laboratory (including L.B. and Y.O.) is supported by the BBSRC (research grants BB/S016295/1 and BB/X008401/1 and an equipment grant, BB/R013764/1) and by the Living Systems Institute, University of Exeter. S.R. was supported by the MRC (MR/S007970/1). The authors thank Z. Jiang, C. Liddle, G. Paull, A. Bell and F. Carlisle for their technical support;

and T. Wohland, G. Jékely, A. Smith and the entire Scholpp lab for their critical comments on the manuscript. The schematic drawing in Fig. 1l was created with BioRender.com.

**Author contributions** C.Z., L.B., S.R. and S.S. designed the experimental strategy. C.Z., L.B. and Y.O. performed and analysed the experiments. S.S., L.B. and C.Z. wrote the manuscript. The project was conceived and supervised by S.S.

**Competing interests** The authors declare no competing interests.

**Additional information**
**Correspondence and requests for materials** should be addressed to Steffen Scholpp.

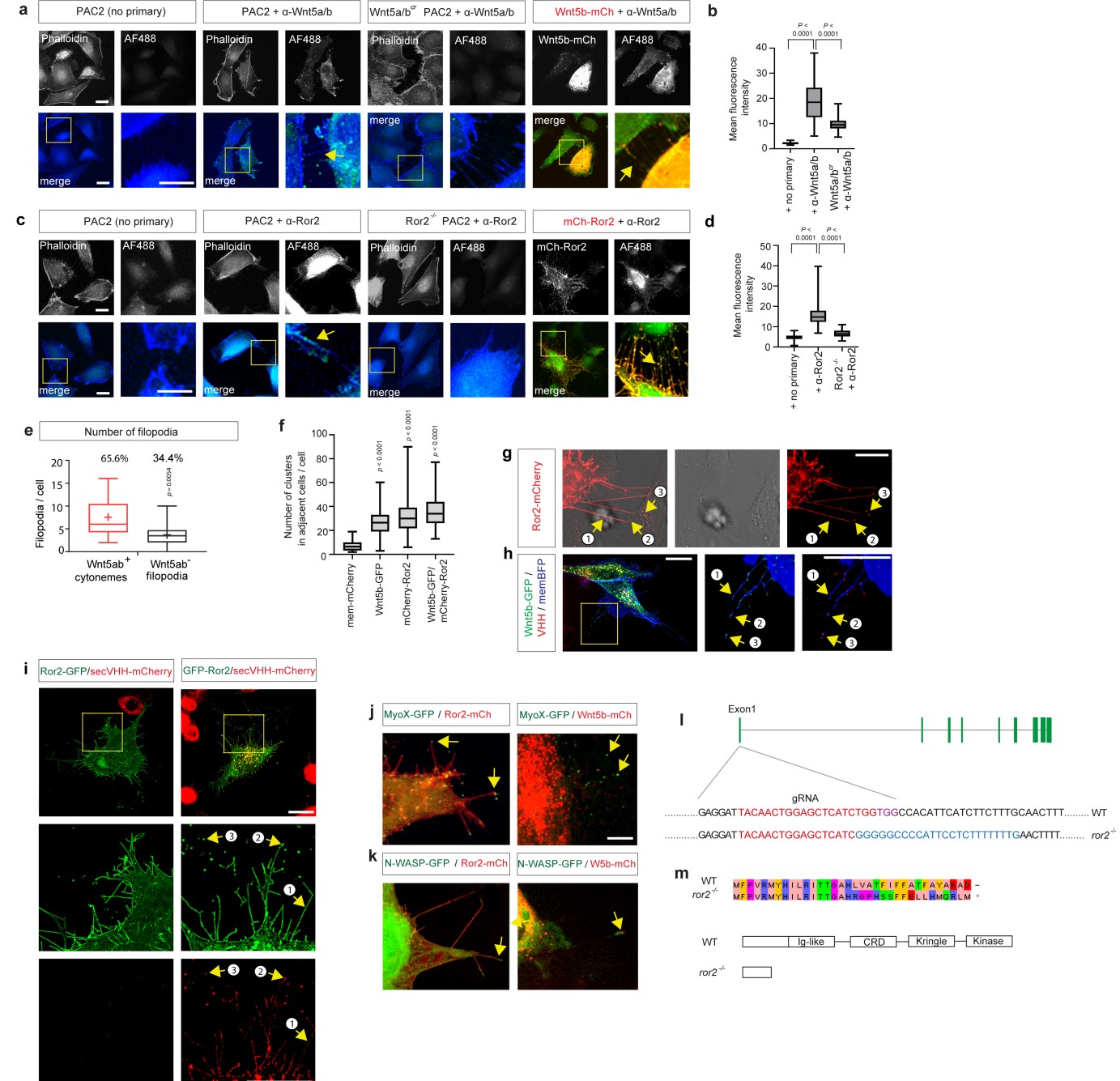

**Extended Data Fig. 1 | Wnt5b/Ror2 transport in vitro and in vivo. a,c.** Wildtype; double crispants *wnt5a/b; ror2⁻/⁻* & Wnt5b-mCh or Ror2-mCh transfected zebrafish fibroblasts processed as described in Fig. 1a,c with/without primary antibodies. A high-magnification image indicates filopodia. Scale bar 10 μm. **b,d.** Fluorescence intensity of secondary antibody in 'no primary' control; Pac2 WT cells, *wnt5a/b^cr^; ror2⁻/⁻* PAC2 cells. **b.** (n = 42,52,107 cells, 3 biological repeats). Significance is calculated by one-way ANOVA together with Tukey posthoc test. p-values as indicated, **d.** (n = 41,73,63 cells, 3 biological repeats). Significance is calculated by Kruskal-Wallis non-parametric test with Bonferroni correction for multiple comparisons. **e.** Quantification of the number of filopodia versus Wnt5a/b-positive cytonemes of PAC2 cells (n = 17, number of cells, three biological repeats). Two-tailed *p*-values were determined by a Mann-Whitney test. **f.** Number of membrane-mCh, Wnt5b-GFP, mCh-Ror2 & Wnt5-GFP/mCh-Ror2 puncta in adjacent non-transfected (brightfield only) cells from Fig. 1f (n = 44, 44, 46, 33, numbers of adjacent cells, three biological repeats). Significance is calculated by one-way ANOVA together with Dunnett multiple comparisons test. p-values as indicated. Box and whisker plots (b,d-f) show median, upper and lower quartile range with whiskers for minimum and maximum values. **g.** PAC2 fibroblasts transfected with Ror2-mCherry contacting receiving cell (brightfield). 1–3 illustrate stages of Ror2-mCherry transport (see Fig. 1l), from (1) cargo transport along cytonemes and (2) cargo-transporting vesicle buds off from a cytoneme to (3) internalization of the cargo-loaded vesicle. Scale bar 10 μm **h-i.** Super-resolution image (based on Lattice SIM² technology) of PAC2 fibroblasts expressing indicated constructs and exposed to secVHH-mCherry to mark extracellular accessible cargo. The numbering describes the steps of cargo transport as described in g. **j,k.** PAC2 zebrafish fibroblasts transfected with indicated constructs and imaged live at 24 h of post-transfection. Yellow arrows co-localization of indicated markers with Wnt5b or Ror2 on cytonemes. Scale bar 5 μm. **l,m.** Verification of CRISPR/Cas9-based knock-out of Ror2 by Sanger sequencing. Binding sites for gRNA and PAM are indicated.

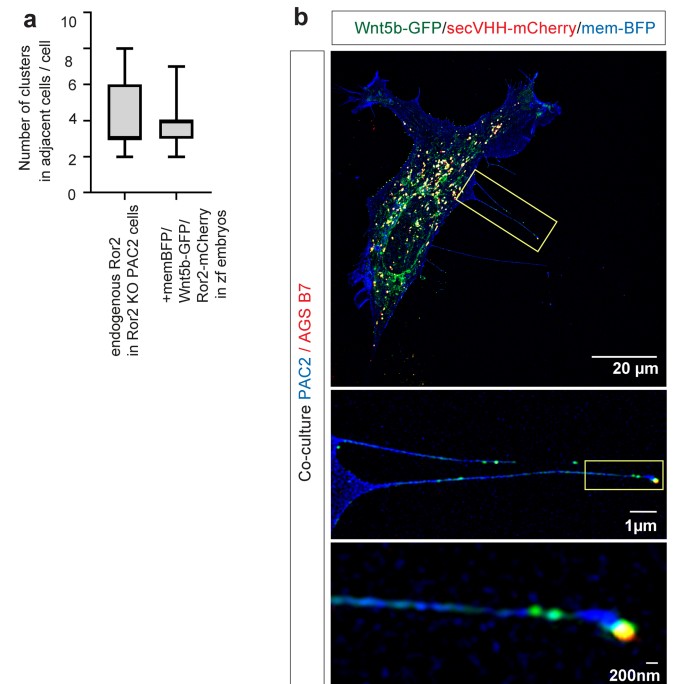

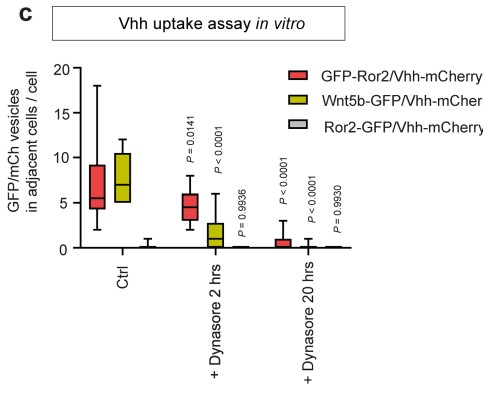

**a** Number of clusters in adjacent cells / cell

endogenous Ror2 in Ror2 KO PAC2 cells

+memBFP/ Wnt5b-GFP/ Ror2-mCherry in zf embryos

**b** Wnt5b-GFP/secVHH-mCherry/mem-BFP

Co-culture PAC2 / AGS B7

20 μm

1μm

200nm

**c** Vhh uptake assay *in vitro*

GFP/mCh vesicles in adjacent cells / cell

- GFP-Ror2/Vhh-mCherry
- Wnt5b-GFP/Vhh-mCherry
- Ror2-GFP/Vhh-mCherry

Ctrl
+ Dynasore 2 hrs
+ Dynasore 20 hrs

$P = 0.0141$
$P < 0.0001$
$P = 0.9936$
$P < 0.0001$
$P < 0.0001$
$P = 0.9930$

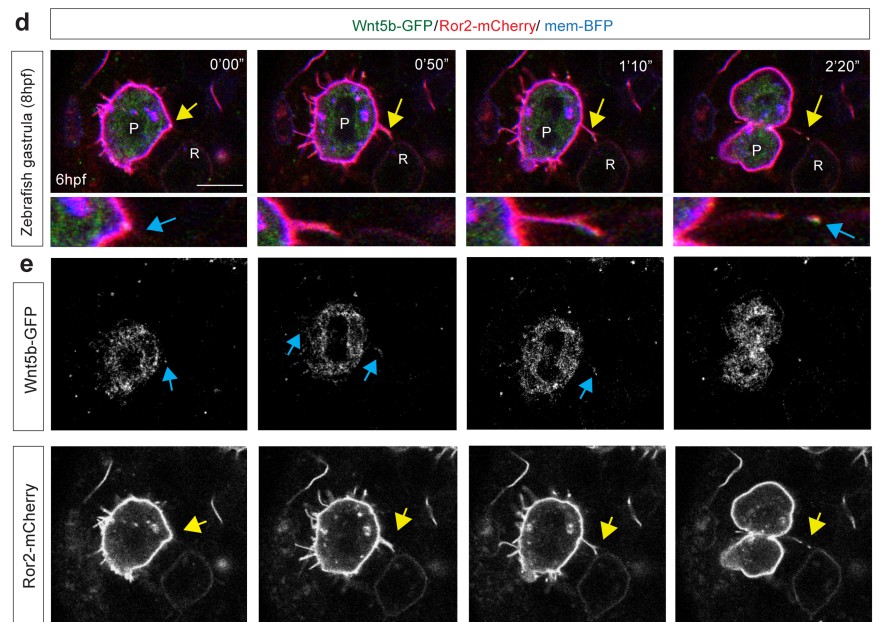

**d** Wnt5b-GFP/Ror2-mCherry/ mem-BFP

Zebrafish gastrula (8hpf)

0'00"  0'50"  1'10"  2'20"

6hpf

P  R

**e** Wnt5b-GFP

Ror2-mCherry

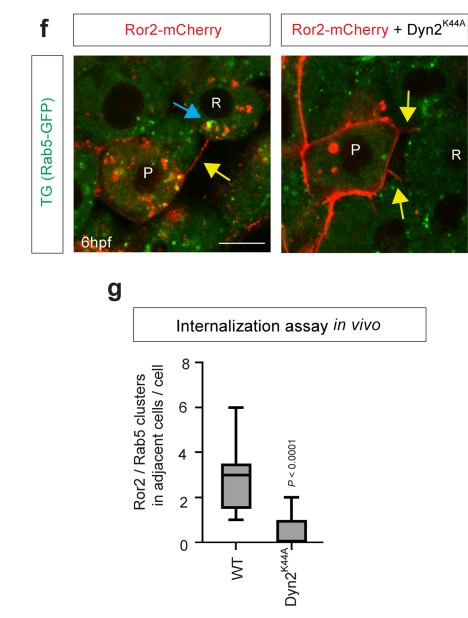

**f** Ror2-mCherry | Ror2-mCherry + Dyn2$^{K44A}$

TG (Rab5-GFP)

6hpf  P  R

**g** Internalization assay *in vivo*

Ror2 / Rab5 clusters in adjacent cells / cell

WT  Dyn2$^{K44A}$

$P < 0.0001$

**Extended Data Fig. 2 | Wnt5b/Ror2 transport and orientation in vitro and in vivo. a**. Quantification of the number of puncta in adjacent *ror2$^{-/-}$* cells from wild-type PAC2 cells and in adjacent cells from clone in embryos from Fig. 1g, Extended Data Fig. 2d (n = 11,11, numbers of adjacent cells). **b**. Super-resolution Structured Illumination Microscopy (SIM) image of a PAC2 treated with SecVHH-mCherry expressing indicated constructs. High-magnification images are shown indicated by yellow boxes. **c**. VHH uptake assay in vitro. Number of GFP/secVHH-mCh positive vesicles in adjacent cells after treatment with secVHH-mCh. PAC2 cells were transfected with Wnt5b-GFP, GFP-Ror2 or Ror2-GFP, with and without 2 hr, 20 hr treatment of 40 μM Dynasore. (n = 10,18, 11;10, 24, 11;10, 9,11, respectively, n = number of adjacent cells, 3 biological repeats). Significance is calculated by two-way ANOVA together with Tukey multiple comparisons test. p-values as indicated. **d**. Time series of cytoneme-based transportation of Wnt5b/Ror2 in zebrafish embryos from 0'00" to 2'20". Scale bar 10 μm (n = 4 embryos). Yellow arrows, Wnt5b/Ror2 cytonemes. **e**. Time series of cytoneme-based transportation

of Wnt5b-GFP and Ror2-mCherry in wild-type zebrafish embryos from 0'00" to 2'20". Embryos were injected with mRNA for Wnt5b-GFP and Ror2-mCherry at the 8-cell stage in one blastomere to generate local clones and imaged live at 6 hpf. Yellow and blue arrows indicate Wnt5b-GFP and Ror2-mCherry localization and cytonemes, respectively. Scale bar 10 μm. **f**. TG (Rab5c:GFP) injected expressing Ror2-mCherry and Ror2-mCherry/Dynamin$^{K44A}$. Yellow arrows, cytonemes from the Producing cell (P), blue arrows, Rab5c:GFP/Ror2-mCh vesicles in the Receiving cell (R), (n = 5, 4 embryos). Scale bar 10 μm. **g**. Early endosome assay in vivo. 1-cell *Tg(Rab5c:GFP)* embryos injected with Dynamin$^{K44A}$ mRNA and then Ror2-mCh mRNA injected at 8-cell to generate clones. Quantification of Ror2-mCh/ Rab5c:GFP vesicles in adjacent receiving cells in Ror2-mCh only or ubiquitously expressing Dyn$^{K44A}$ embryos. (n = 5, 4, n = numbers of embryos). Significance is calculated by unpaired Student T- test and *p* value is shown in a two-tailed way. Box and whisker plots (a,c,g) show median, upper and lower quartile range with whiskers for minimum and maximum values.

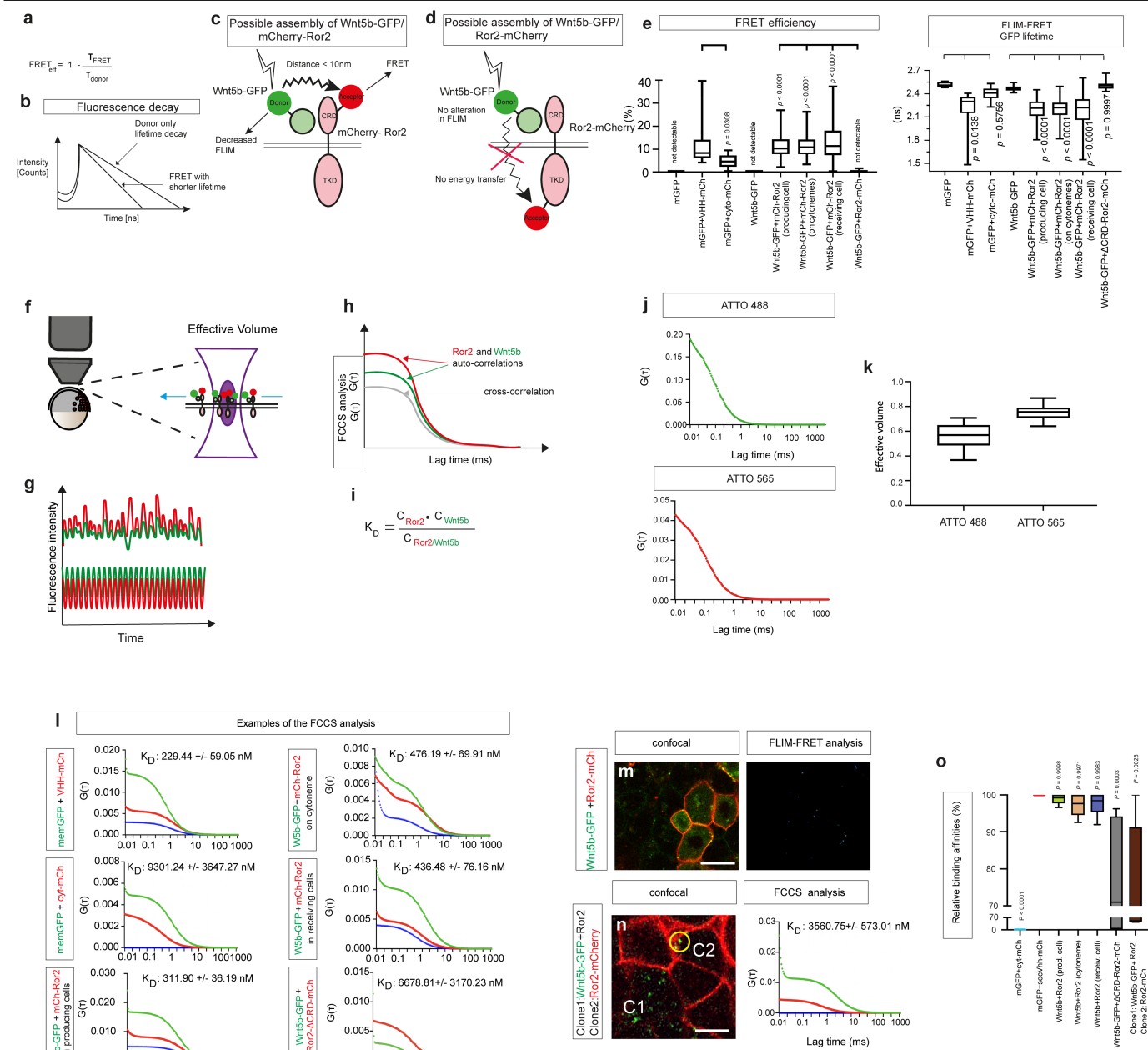

**Extended Data Fig. 3 | Principles of FLIM-FRET analysis of Wnt5b/Ror2 complexes. a.** FRET efficiency, FRET$_{eff}$, is defined as the fraction of photons that is transferred from a donor to an acceptor. $\tau_{FRET}$, a lifetime of the donor in the presence of the acceptor; $\tau_{donor}$, lifetime if the donor is without the acceptor. **b.** Bi-exponential fitting of the fluorescence decay demonstrates that the lifetime of a donor is decreased in the presence of a FRET acceptor. **c.** Close packaging in the Wnt5b-GFP/mCherry-Ror2 complex allows FRET to the acceptor and, thus, reduces donor FLIM. **d.** A C-terminally fused Ror2-mCherry can be used as a negative control like the distance between donor and acceptor is too large, and the plasma membrane acts like as insulator. **e.** Quantification of FRET efficiency (n = 0, 17, 24, 0, 36, 22, 30, 11, n = regions of interests in three biological repeats). Significance is calculated by a one-way ANOVA test plus Dunnett multiple comparisons test. p-values shown on the graph are compared to the corresponding negative controls, indicated on the graph. Quantification of fluorescent GFP lifetime (n = 8, 47, 25, 21, 36, 23, 33, 11, n = regions of interest in three biological repeats). Significance is calculated by a one-way ANOVA test plus Tukey multiple comparisons test. *p*-values shown on the graph are compared to the corresponding negative controls indicated on the graph. **f.** Schematic representation of the confocal detection volume positioned on the plasma membrane in a living zebrafish embryo to measure ligand-receptor interactions. **g.** Fluctuations in the detected fluorescence intensities with time. **h.** Auto- and cross-correlation analysis of fluorescence fluctuations over time. The Grey curve represents the cross-correlation between Wnt5b-GFP and Ror2-mCherry. **i.** The dissociation constant K$_D$ is defined as the fraction of the concentration of the unbound ligands and unbound receptors to the concentration of the ligand-receptor complexes. **j.** Autocorrelation curves of ATTO 488 and ATTO 565 in solution to determine the FCCS volume. **k.** Determination of the FCS volume obtained from n = 10 independent samples of labeled protein stock solutions. **l.** For FCCS analysis, the auto-correlation curves for Wnt5b-GFP and Ror2-mCherry are shown in red and green, respectively. The cross-correlation curve is shown in blue. The corresponding average dissociation constant (K$_D$) is indicated on the graph. **m-n.** Representative fluorescence images for FCCS measurements and example measuring spots (yellow circle) in the zebrafish embryo at 6hpf, injected with indicated mRNA. C1 indicates the Wnt5b-GFP/ Ror2 positive clone, and C2 indicates a clone expressing Ror2-mCherry. Scale bar 10 µm. **o.** Relative binding affinities measured by FCCS for indicated fluorescent protein combinations from Fig. 2h. Significance is calculated by one-way ANOVA together with Dunnett multiple comparisons test. Box and whisker plots (e,k,o) show median, upper and lower quartile range with whiskers for minimum and maximum values.

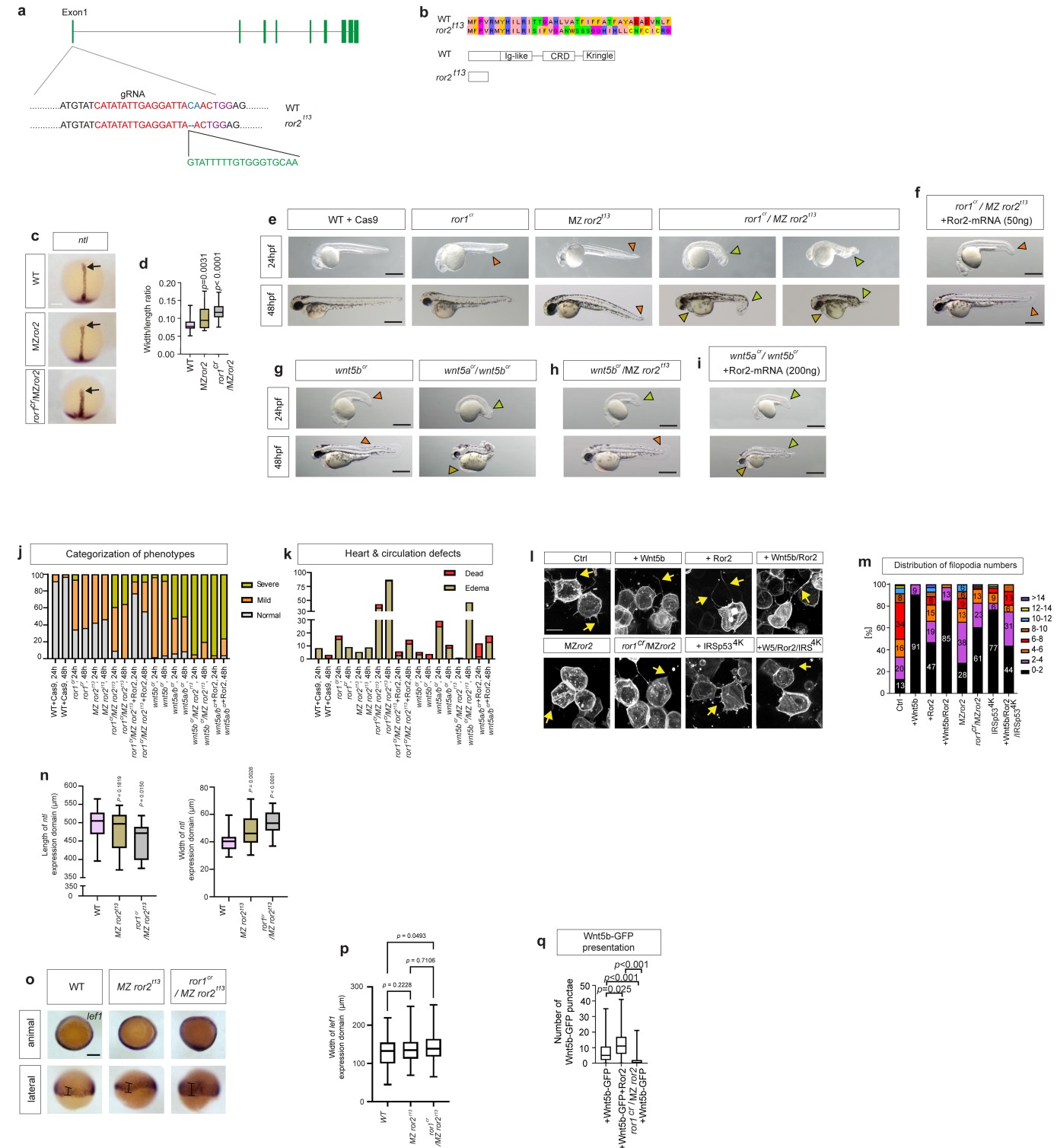

**Extended Data Fig. 4** | See next page for caption.

**Extended Data Fig. 4 | Generation of the zebrafish mutant line *ror2$^{t13}$*. a.** By using the CRISPR/Cas9 technology, a deletion/insertion mutant for Ror2 was generated. An insertion of 25 amino acids led to a new termination site of the protein at position 38. **b.** CRISPR/Cas9-based deletion/insertion generated a truncated Ror2 protein. **c.** *ntl* expression in indicated mutant lines. **d.** Width/length ratio of *ntl* expression (n = 38, 25, 13 embryos, 3 biological repeats) including a one-way ANOVA test with a Dunnett multiple comparisons test. *p*-values as indicated. Scale bar = 100 μm. **e.** Morphology phenotypes for WT +Cas9, *ror1$^{cr}$*, *ror2$^{t13}$*, *ror2$^{t13}$ & ror1$^{cr}$/ror2$^{t13}$* 24 and 48hpf larvae. **f.** Morphology phenotypes of *ror1$^{cr}$/ror2$^{t13}$* 24 and 48 hpf larvae with 50 ng Ror2 mRNA injected at 1-cell. **g.** Morphology phenotypes of Wnt5b$^{cr}$ & Wnt5a/Wnt5b$^{cr}$, **h.** *Wnt5b$^{cr}$/ror2$^{t13}$*. Wnt5a/b gRNA sequences (IDT DNA): (wnt5a-AE: AGATCGTGGACGCAA ACTCATGG, wnt5a-AF: CGTCGACAACTCCACAGTGCTGG, wnt5b-AD: AGGTGG AAAGCTCACCCTCACGG, wnt5b-AE: GAACCAAGGACACCTACTTCTGG), **i.** Double crispants *wnt5a/wnt5b$^{cr}$* with 200 ng Ror2 mRNA injected at 1-cell at 24 and 48 hpf. Orange and yellow arrowheads show mild and severe tail morphology, light orange arrowheads show heart oedema. (e-i), Scale bar = 500 μm. **j.** Total percentage of normal, mild and severe phenotypes. **k.** Quantification of heart and circulation defects (percentage dead or with oedema). (n = 12, 127, 78, 11, 75, 45, 128, 93, 35, 34, 56, 52, 48, 46, 52, 52, 40, 40 embryos, three biological repeats). **l.** Confocal images of zebrafish embryos injected with indicated constructs or gRNA and imaged at 6hpf. Scale bar 10 μm. **m.** Quantitation of filopodia numbers per cell in vivo, categories: 0-2, 2–4, 4–8, 8–10, 10–12, 12–14, >14. Percentage is shown on the bar (n = 98, 43, 119, 40, 32, 71, 65, 16 cells, three biological repeats). **n.** Quantification of length and width of *ntl* expression domain in wild type, *ror2$^{t13}$ & ror1$^{cr}$/ror2$^{t13}$* embryos from Extended Data Fig. 4c (n = 38, 25, 13 embryos, 3 biological repeats). Significance is calculated by one-way ANOVA together with Dunnett multiple comparisons test. *p*-values as indicated. **o.** in situ hybridization of *lef1* expression of wild type, *ror2$^{t13}$ & ror1$^{cr}$/ror2$^{t13}$* 50% epiboly embryos (animal pole and lateral view). Black line shows width of domain. Scale bar 200 μm. **p.** Quantification of width of *lef1* expression domain. (n = 38,40,31, n = number of embryos). Significance is calculated by one-way ANOVA together with Tukey multiple comparisons test. *p*-values as indicated. **q.** Wnt5b-GFP puncta on plasma membrane of producing cells (n = 9, 6,9 embryos, three biological repeats). A Kruskall-Wallis non-parametric multiple comparisons test, with a Bonferroni correction for multiple comparisons was used. Box and whisker plots (d,n,p,q) show median, upper and lower quartile range with whiskers for minimum and maximum values.

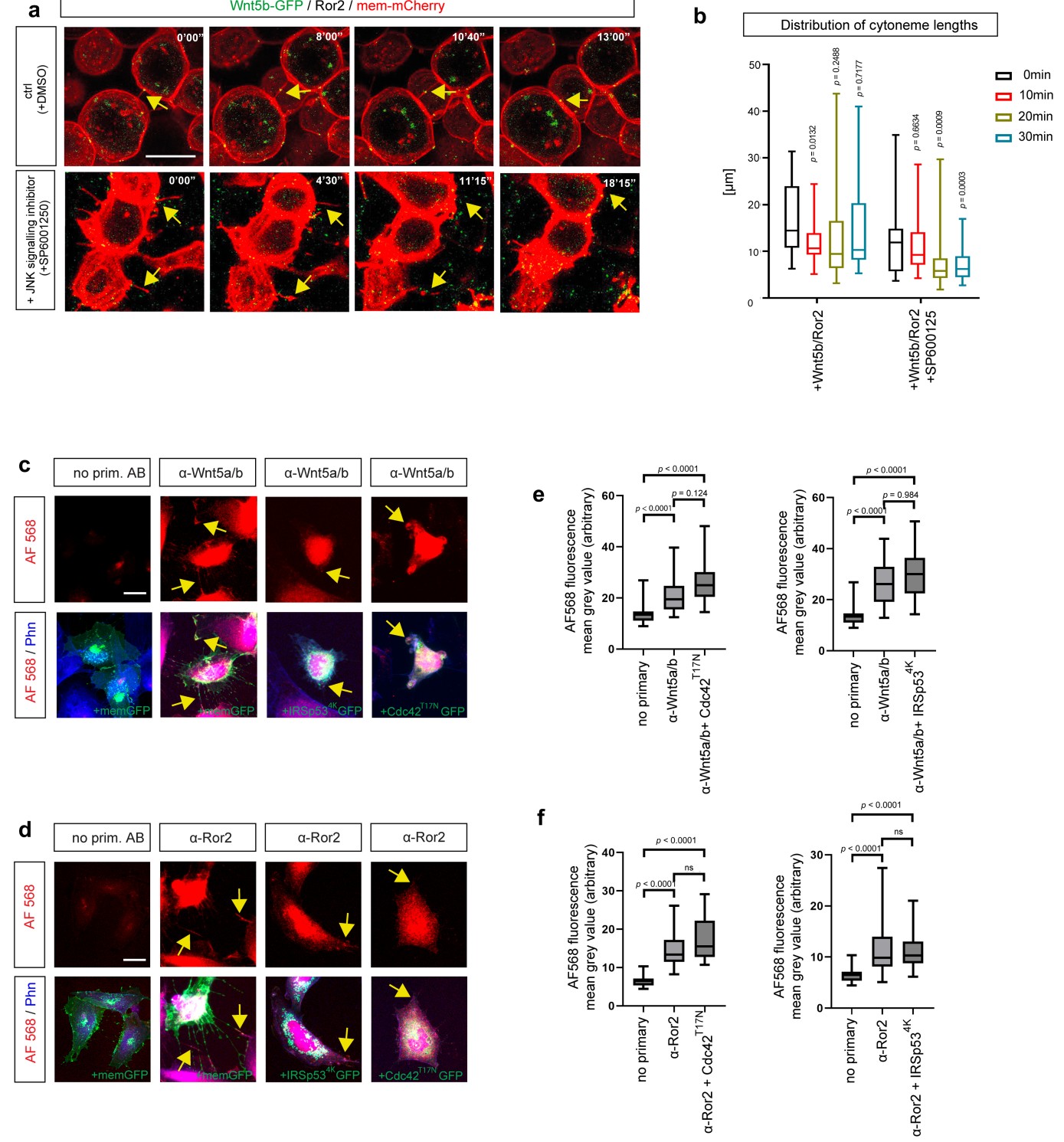

**Extended Data Fig. 5 | Cytoneme modulators for Wnt5b/Ror2 transport.** **a**. Time series of cytonemes in wild-type zebrafish embryos. Embryos were injected with Wnt5b-GFP/Ror2/mem-mCherry in the clone. At 6hpf, the embryos were treated with DMSO or JNK inhibitor SP600125 from 0′00″ to 35′00″ and imaged live. Yellow arrows indicate the retraction of cytonemes at the corresponding time points. **b**. The quantitative analysis of cytoneme length at the different time points in zebrafish embryos was treated as indicated in (a). (Wnt5b/Ror2: n = 25, 23, 16, 13; Wnt5b/Ror2 + SP600125: n = 35, 34, 38, 54, n = numbers of cytonemes in three biological repeats). Significance is calculated by two-way ANOVA together with Dunnett multiple comparisons test. *p*-values as indicated. **c**,**d**. PAC2 cells were transfected with indicated constructs and processed with a Wnt5a/b (**c**) and a Ror2 (**d**) antibody and phalloidin 405.

Scale bar 20 μm. **e**. Fluorescence intensity of secondary antibody in IRSp53⁴ᴷ-GFP and CDC42^T17N-GFP transfected PAC2 cells processed with Wnt5a/b antibody. (n = 28,23, 26, n = 28,31,22, n = number of cells in three biological repeats). Significance is calculated by the Kruskal-Wallis non-parametric test with a Bonferroni correction for multiple comparisons. **f**. Fluorescence intensity of secondary antibody in IRSp53⁴ᴷ-GFP and CDC42^T17N-GFP transfected PAC2 cells processed with Ror2 antibody. (n = 23,26,23, n = 23,32,29, n = number of cells in three biological repeats). Significance is calculated by the Kruskal-Wallis non-parametric test with a Bonferroni correction for multiple comparisons. Box and whisker plots (b,e,f) show median, upper and lower quartile range with whiskers for minimum and maximum values.

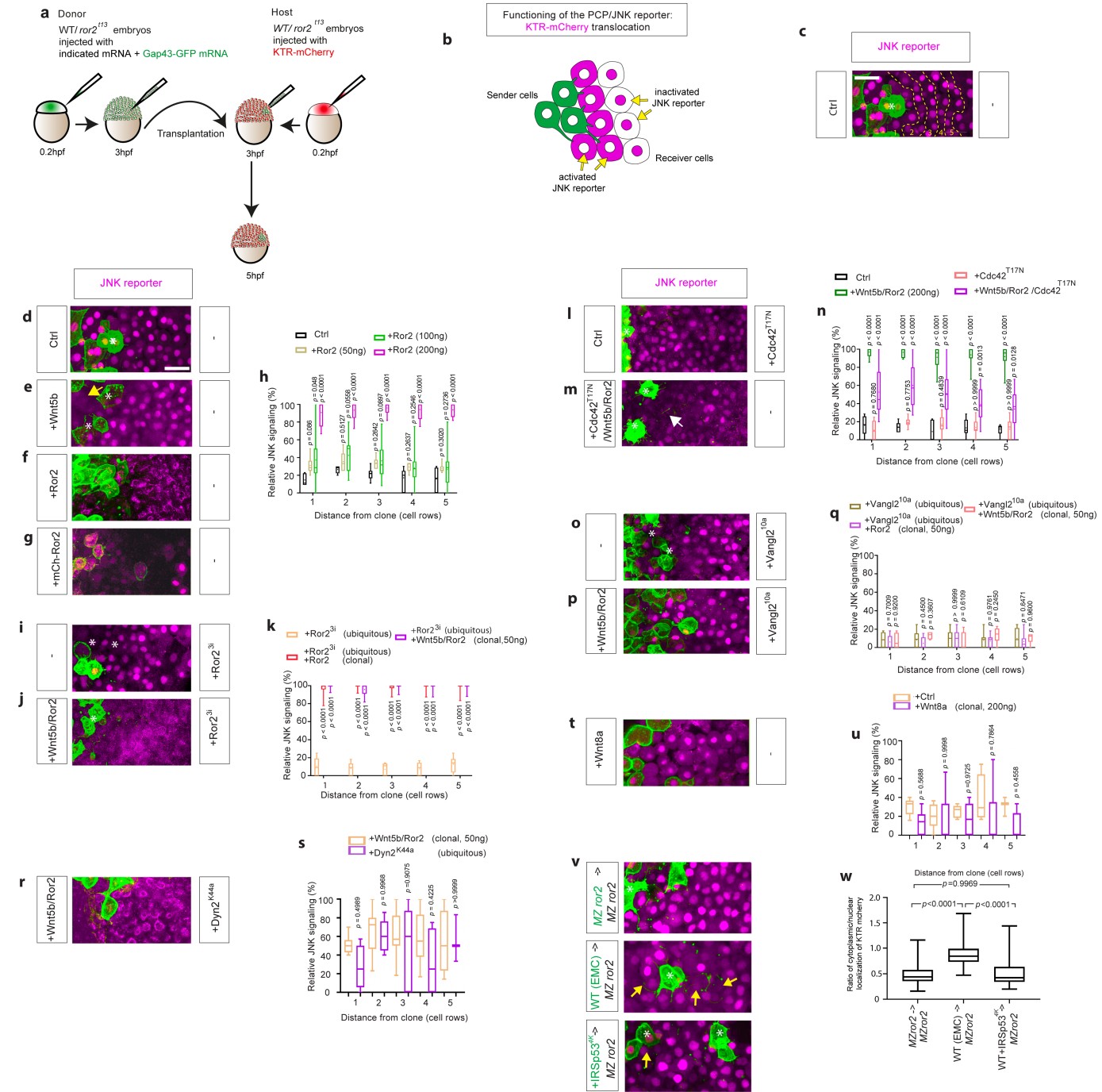

**Extended Data Fig. 6 | Wnt5b/Ror2 expressing clones activate autocrine and paracrine JNK signaling. a.** Schematic drawing illustrates the transplantation strategy from a WT or *ror2^t13* donor to a *ror2^t13* mutant host (for Extended Data Fig. 6v,w). **b.** Schematic drawing of principle of the JNK signaling KTR-reporter. **c.** Principle of measurements of cell rows (1–5), around the Wnt5b/Ror2-expressing clones. Scale Bar 20 μm. **d,e,f,g,i,j,l,m,o,p,r,t.** Wild-type zebrafish embryos were ubiquitously injected with KTR-mCherry and indicated constructs (ubiquitous, in the right box). At the 8-cell stage, one blastomere was injected with the indicated constructs (clonal, in the left box) to generate clones at the embryonic margin and imaged live at 6 hpf. White stars indicate the signaling-producing clonal cells expressing memGFP. Scale Bar 20 μm. **g.** N-terminal fusion of mCherry with Ror2 activates JNK signaling around the clone, similar to the C-terminal fusion. **h,k,n,q,s,u.** Relative JNK signaling within 5 cell rows distance from clone when injected with different amounts of Ror2 (n = 7, 7, 11, 10, n = numbers per embryo, 3 biological repeats), Cdc42^T17N (n = 6, 12, 11, 14, n = numbers per embryo, 3 biological repeats), with ubiquitous

Ror2^3i (n = 9, 10, 11, n = numbers of embryos, 3 biological repeats), ubiquitous Vangl2^10a (n = 11, 11, 6, n = numbers of embryos, 3 biological repeats), ubiquitous Dynamin2^K44A (n = 8, 5, n = number of embryos, 3 biological repeats), with clonal Wnt8a (n = 5, 7, n = number of embryos, three biological repeats). Significance is calculated by two-way ANOVA with Dunnett (**h,n,k,q**) and Sidak's (**s,u**) multiple comparisons test. *p*-values as indicated. **v,w.** Transplantation assay. *ror2^t13* cells or WT cells or WT cells expressing IRSp53^4K were labeled with memGFP (white asterisks) and grafted into a *ror2^t13* mutant embryo microinjected with the KTR JNK reporter. Yellow arrows highlight cells targeted by cytonemes in which KTR mCherry is shifted from the nucleus to the cytoplasm, indicating JNK activation, asterisks represent the location of grafted cells. **w.** Quantification of the cytoplasmic/nuclear ratio of KTR mCherry in the neighboring cells of 2–5 grafted cells from 7 different embryos per treatment. Significance is calculated by a one-way ANOVA together with a Dunnett multiple comparisons test. *p*-values as indicated, Scale bars 20 μm.

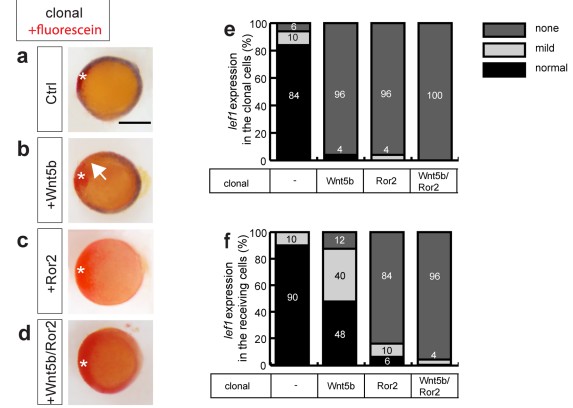

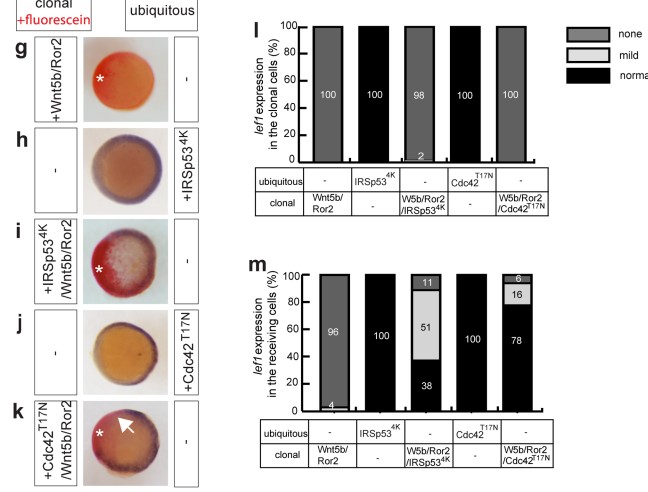

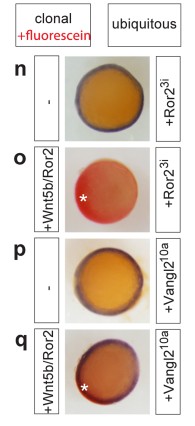

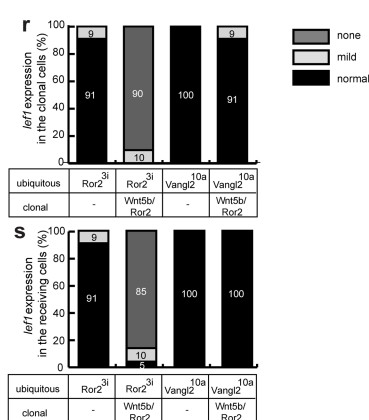

**Extended Data Fig. 7 | Wnt5b/Ror2 expressing clones suppress clonal and adjacent *lef1* expression. a-d, g-k, n-q.** Embryos injected with the mRNA for the indicated constructs at the 8-cell stage in one blastomere to generate local clones were subjected to an in situ hybridization analysis against *lef1* at 6 hpf. The embryos were stained and mounted with the animal pole up. Purple staining around the margin indicates the expression of *lef1*, and red staining (white star) indicates the site of the injected clonal cells. Clonal cells were co-injected with Mini-Emerald as a lineage tracer. Scale bar 200 μm.

**e,f,l,m,r,s.** Classification of *lef1* expression in the clonal cells and the receiving cells. The expression is classified according to the severity of the downregulation of the *lef1* expression into not detectable (e.g., d), reduced (e.g., b), and normal (e.g., a). Numbers in the bars represent the percentage of total embryos in each group. (n = 31, 52, 79, 49, 24, 47, 24, 32, 91, 89, 16, 24, 24, n = numbers of embryos).

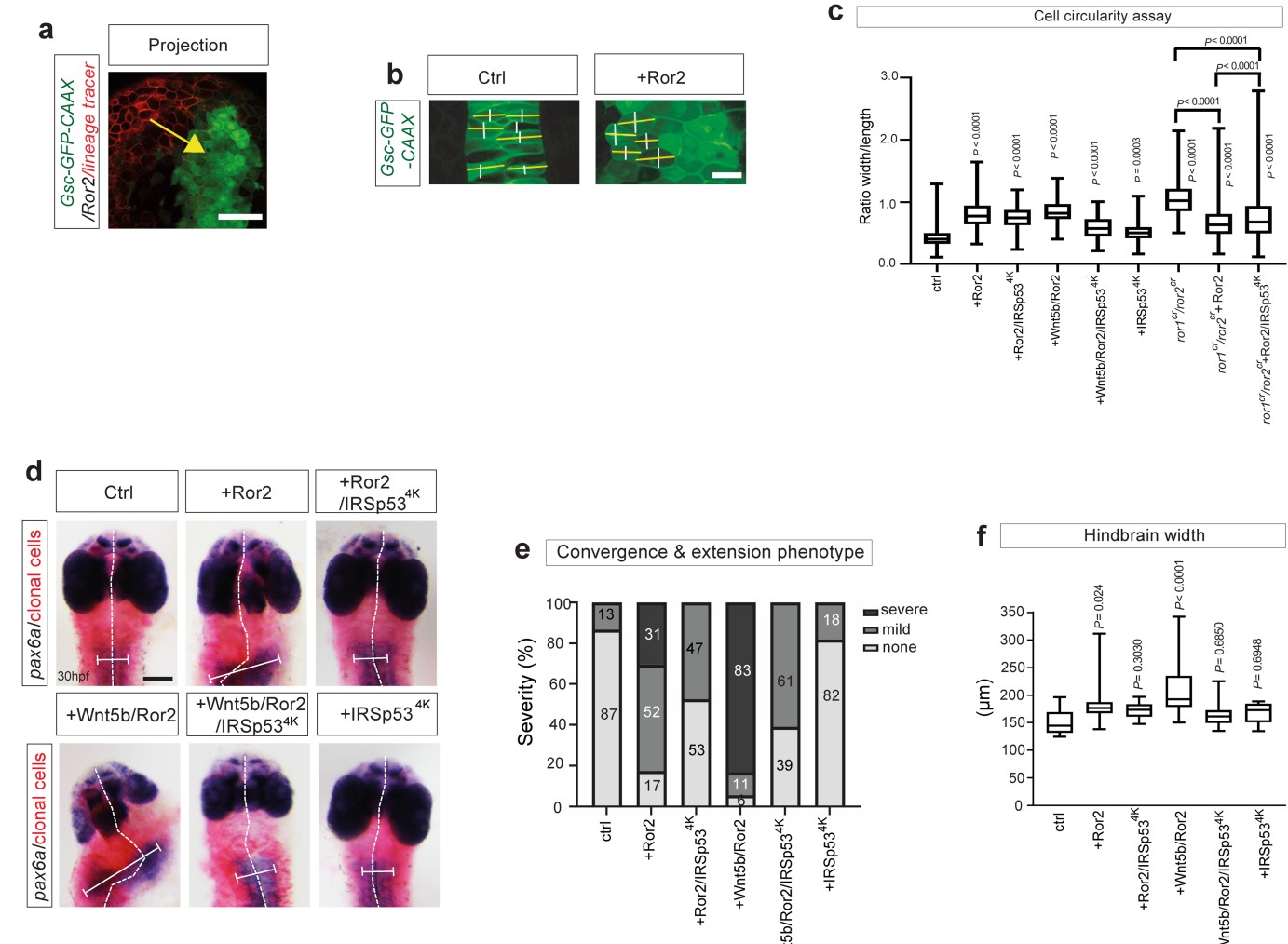

**Extended Data Fig. 8 | Principles of convergence and extension assay. a.** An example maximum projection image to illustrate the proximity of the lineage tracer labeled clones (red) to Tg(Gsc-GFP-CAAX) expressing cells in the embryo shown in Fig. 5a–i. The yellow arrow shows the proximity. Scale bar is 50 μm **b.** Circularity assay. Some example cells illustrate how the width of a cell in parallel to the body axis (white lines) or perpendicular to the body axis (yellow line) was measured in a WT embryo or an embryo with a Ror2 expressing clone at 24 h. The ratio between the yellow line and the white line was calculated. A perfect circular cell has a circularity of 1.0, while below or above 1.0 indicates a noncircular, elongated shape. Scale bar is 20 μm. **c.** Cell circularity was measured within each embryo (n = 173, 292, 194, 200, 149, 176, 305, 458, 607 n = numbers per cell, 3 biological repeats). A perfect circular cell has a circularity of 1.0, while below or above 1.0 indicates a noncircular, elongated shape. **d.** Wild-type zebrafish embryos injected with mRNA for the indicated constructs plus Mini-Emerald (lineage tracer), subsequent in situ hybridization against *pax6a*. White dashed lines indicate the course of the midline, and the white bars indicate the width of the hindbrain domain. Scale bar 200 μm. **e.** The quantitative analysis of phenotype severity in zebrafish embryos. The phenotype severity is classified into none (e.g. ctrl), mild (e.g., +Ror2), and severe (e.g., +Wnt5b/Ror2). Ctrl, +Ror2 and +Wnt5b/Ror2 are representative examples of the different severity classifications, respectively. Numbers in bars represent the percentage of total embryos in each group. **f.** Hindbrain width was measured in embryos. n = 13, 23, 19, 18, 23, 11, n = numbers per embryo, 3 biological repeats. Significance is calculated by a one-way ANOVA test plus a Tukey multiple comparisons test. Box and whisker plots (c,f) show median, upper and lower quartile ranges with whiskers for minimum and maximum values.

# Reporting Summary

## Statistics

For all statistical analyses, confirm that the following items are present in the figure legend, table legend, main text, or Methods section.

| n/a | Confirmed | |
|---|---|---|
| ☐ | ☒ | The exact sample size (*n*) for each experimental group/condition, given as a discrete number and unit of measurement |
| ☐ | ☒ | A statement on whether measurements were taken from distinct samples or whether the same sample was measured repeatedly |
| ☐ | ☒ | The statistical test(s) used AND whether they are one- or two-sided *Only common tests should be described solely by name; describe more complex techniques in the Methods section.* |
| ☐ | ☒ | A description of all covariates tested |
| ☐ | ☒ | A description of any assumptions or corrections, such as tests of normality and adjustment for multiple comparisons |
| ☐ | ☒ | A full description of the statistical parameters including central tendency (e.g. means) or other basic estimates (e.g. regression coefficient) AND variation (e.g. standard deviation) or associated estimates of uncertainty (e.g. confidence intervals) |
| ☐ | ☒ | For null hypothesis testing, the test statistic (e.g. *F*, *t*, *r*) with confidence intervals, effect sizes, degrees of freedom and *P* value noted *Give P values as exact values whenever suitable.* |
| ☒ | ☐ | For Bayesian analysis, information on the choice of priors and Markov chain Monte Carlo settings |
| ☒ | ☐ | For hierarchical and complex designs, identification of the appropriate level for tests and full reporting of outcomes |
| ☒ | ☐ | Estimates of effect sizes (e.g. Cohen's *d*, Pearson's *r*), indicating how they were calculated |

*Our web collection on statistics for biologists contains articles on many of the points above.*

## Software and code

Policy information about availability of computer code

| Data collection | Image data were visualised and analysed in Leica LAS X software (Version 3.7.2.22383) and FIJI (ImageJ 2.00-rc-59/1.53c). Movies were produced in Imaris 9.0.0 (Bitplane AG). |
|---|---|
| Data analysis | Statistical analysis was carried out using GraphPad Prism 9.0. Depending on different experiments, ordinary one-way ANOVA and Tukey's multiple comparisons test, and two-way ANOVA together with Dunnett's multiple comparisons test were used. |

For manuscripts utilizing custom algorithms or software that are central to the research but not yet described in published literature, software must be made available to editors and reviewers. We strongly encourage code deposition in a community repository (e.g. GitHub). See the Nature Portfolio guidelines for submitting code & software for further information.

## Data

Policy information about availability of data

All manuscripts must include a data availability statement. This statement should provide the following information, where applicable:
- Accession codes, unique identifiers, or web links for publicly available datasets
- A description of any restrictions on data availability
- For clinical datasets or third party data, please ensure that the statement adheres to our policy

Microscopy data reported in this paper and any information required to re-analyse the data reported in this paper are presented in the Extended Figures or are available from the lead contact upon reasonable request.

# Human research participants

Policy information about studies involving human research participants and Sex and Gender in Research.

| | |
|---|---|
| Reporting on sex and gender | N/A |
| Population characteristics | N/A |
| Recruitment | N/A |
| Ethics oversight | N/A |

Note that full information on the approval of the study protocol must also be provided in the manuscript.

# Field-specific reporting

Please select the one below that is the best fit for your research. If you are not sure, read the appropriate sections before making your selection.

☒ Life sciences  ☐ Behavioural & social sciences  ☐ Ecological, evolutionary & environmental sciences

For a reference copy of the document with all sections, see nature.com/documents/nr-reporting-summary-flat.pdf

# Life sciences study design

All studies must disclose on these points even when the disclosure is negative.

| | |
|---|---|
| Sample size | Sample sizes may be chosen by using a target of power of a statistical test to be applied once the sample is collected and described in the figure legends. |
| Data exclusions | No data were excluded from the analyses. |
| Replication | All experiments were done as three independent experiments with three biological replicates analysed. All attempts correctly executed were included. All details were described in the figure legends. |
| Randomization | Randomization involved selection of random samples within a experimental /treatment group by the investigators. Usually 10 randomly selected samples (i.e. cells, zebrafish embryos) were chosen and the parameters analysed. The investigators choosing the samples did not know about the particular treatment. |
| Blinding | Samples were chosen randomly from a group without giving the investigators access to the treatment protocol. |

# Reporting for specific materials, systems and methods

We require information from authors about some types of materials, experimental systems and methods used in many studies. Here, indicate whether each material, system or method listed is relevant to your study. If you are not sure if a list item applies to your research, read the appropriate section before selecting a response.

## Materials & experimental systems

| n/a | Involved in the study |
|---|---|
| ☐ | ☒ Antibodies |
| ☐ | ☒ Eukaryotic cell lines |
| ☒ | ☐ Palaeontology and archaeology |
| ☐ | ☒ Animals and other organisms |
| ☒ | ☐ Clinical data |
| ☒ | ☐ Dual use research of concern |

## Methods

| n/a | Involved in the study |
|---|---|
| ☒ | ☐ ChIP-seq |
| ☒ | ☐ Flow cytometry |
| ☒ | ☐ MRI-based neuroimaging |

# Antibodies

| | |
|---|---|
| Antibodies used | WNT5A-B, rabbit PolyAb, ProteinTech, 55184-1-AP, 1:50; ROR2 (D3B6F), rabbit mAb, Cell Signalling Technology, 88639S, 1:50; anti-rabbit antibody Alexafluor 488, ab150077, Abcam, 1:1000; donkey anti-goat antibody AlexaFluor 647, ab150135, Abcam, 1:1000 |
| Validation | All primary antibodies have been validates by IHC and WB as stated in the manuscript. |

# Eukaryotic cell lines

Policy information about cell lines and Sex and Gender in Research

| | |
|---|---|
| Cell line source(s) | PAC2 zebrafish fibroblasts provided by Nicholas Foulkes (KIT, Germany) and Reinhard Koester (TU Braunschweig, Germany) - No commercial source. AGS (Gastric cancer cells)- American Tissue Culture Collection, ATTC, Wesel, Germany |
| Authentication | None of the cell lines were authenticated. |
| Mycoplasma contamination | The cell line was tested for Mycoplasma contamination every three months and confirmed mycoplasma-free - overseen by experimental officer Dr Francesca Carlie, LSI Tissue Culture Facility. |
| Commonly misidentified lines (See ICLAC register) | No commonly misidentified cell lines were used in this study. |

# Animals and other research organisms

Policy information about studies involving animals; ARRIVE guidelines recommended for reporting animal research, and Sex and Gender in Research

| | |
|---|---|
| Laboratory animals | zebrafish, strain: WIK, sex: male/female, age: 8-18months |
| Wild animals | No wild animals were used in the study. |
| Reporting on sex | N/A |
| Field-collected samples | No field collected samples were used in the study. |
| Ethics oversight | WIK wild-type, Tg(-6gsc: EGFP –CAAX), and Ror2 (T13fs38X) zebrafish (Danio rerio) were maintained as previously described at 28°C and on a 14hr light/10hr dark cycle. Zebrafish care and all experimental procedures were carried out in accordance with the European Communities Council Directive (2010/63/EU) and Animals Scientific Procedures Act (ASPA) 1986. In detail, adult zebrafish for breeding were kept and handled according to the ASPA animal care regulations and all embryo experiments were performed before 120h post fertilization. Zebrafish experimental procedures were carried out under personal and project licenses granted by the UK Home Office under ASPA. The project has been ethically approved by the Animal Welfare and Ethical Review Body at the University of Exeter. |

Note that full information on the approval of the study protocol must also be provided in the manuscript.

