## [Peer Review File · Nature]

Manuscript Title: Cytoneme-mediated transport of active Wnt5b/Ror2 complexes in zebrafish

Reviewer Comments & Author Rebuttals

Reviewer Reports on the Initial Version:

Referee main expertise:

Referee #1: Wnt signalling

Referee #2: Cytonemes

Referee #3: Ligand-receptor interactions

Referee #4: Zebrafish gastrulation, Wnt signalling

Referees' comments:

Referee #1 (Remarks to the Author):

Zhang et al report on work that is built on an earlier paper from the same group, (Mattes, B. et al. Wnt/PCP controls spreading of Wnt/ β -catenin signals by cytonemes in vertebrates. *Elife* 7, e36953 (2018). Purportedly, the group now shows that the Wnt5b protein binds to a receptor, Ror2, to activate the Wnt/planar cell polarity signaling pathway in neighboring cells to regulate tissue polarity and cell migration. Based on imaging approaches, the authors suggest that the Wnt5b growth factor is loaded on long protrusions called cytonemes in a complex with its Ror2 receptor, thereby providing a mechanism for long range signaling by Wnts. They also propose that Wnt5b/Ror2 complexes are formed in the producing cell and handed over from cytonemes to the receiving cell. This would then activate Wnt/PCP signaling even without the need for the expression of functional receptors. In an overall conclusion the authors suggest that cytoneme-mediated transfer of ligand-receptor complexes is a mechanism for paracrine signaling and challenge the existing knowledge that tissues need receptors to respond to signals.

These conclusions are certainly innovative and if held up by strong experimental results, worthy of considering for publication in *Nature*. Indeed, to claim that cells can respond to a signal without a receptor, and mediated by a protrusion (cytoneme) from another cell would call for unequivocal experimental support. However, there are a number of questionable data in this manuscript that need to be addressed.

The phenotype of the genes involved. It is stated that Wnt5b and Ror 2 are implicated in planar cell polarity but that poorly supported and not at all by classical loss of function genetics. In their previous paper (*eLife*, 2018), a Morpholino-based Ror2 knock-down experiment is shown. Otherwise, the Ror receptors are implicated in convergence extension in *Xenopus*, but without

genetic data in that organism. Surprisingly, in the current submission on line 220, the authors actually describe that they made a mutant Zebrafish line with a loss of function mutation in Ror2 made by Crispr technology. They do experiments with cells obtained from these animals but don't describe an organism phenotype in developing fish that are homozygous or heteroallelic mutant for the gene. Why not, this is such an important piece of information. Are there planar polarity phenotypes.

Similarly, there is no information on possible mutant phenotypes of Wnt5b. It would not be hard to generate Crispr alleles of Wnt5b, a gene that the group has been working on since at least 2018. Having loss of function alleles of these two main players, Wnt5b and Ror2, will also allow for testing the specificity of the antibodies used. This is always a concern, but if the staining is absent in a mutant, this can be resolved. In this context, it should be mentioned that the authors use antibodies made against Wnt5a/b (line 80) , and in the paper sometimes refer to Wnt5 (not Wnt5b) as being involved in PCP signaling (line 75), sometimes to the Wnt5a/b protein (line 90). This tends to be confusing.

A second major experiment that is missing is a biochemical and cell biological characterization of the Wnt5b/Ror 2 complex. It is a mystery how a complex of a ligand and its receptor would be taken up by a cell, without any cognate receptor-like molecule. How does a receptor protein complex get into a cell from the outside, adopt a conformation to signal, which needs to happen over a membrane? As this claim is the centerpiece of the manuscript, and at the same time without any precedent, it is reasonable to ask for more experiments. Can the complex be purified and added to cells? Is the complex attached to membranes produced by the cytonemes? If it is membrane bound, the complex should be sensitive to detergents. The authors are using a cultured cell system, PAC2 cells and this might be adapted to study the cell biology of the uptake of the Wnt5b/Ror 2 complex.

As an argument that active signaling occurs, evidence is shown that the Ror2 molecule ends up in the nucleus of the target cells. This is puzzling. Ror2 is a receptor tyrosine kinase, and these molecules in general do not end up in the nucleus. The authors refer to a relatively obscure publication that would support this observation (Carbone, C. et al. Adipocytes sustain pancreatic cancer progression through a non- canonical WNT paracrine network inducing ROR2 nuclear shuttling. *Int. J. Obesity* 42, 334-343 (2018)). However the data in that paper showing that Ror2 becomes nuclear are not strong and the case becomes yet weaker reading that Carbone et al seem to think that Ror2 is "a 7 trans- membrane receptor". It is not as we all know, Ror2 is a single transmembrane tyrosine kinase equipped receptor.

As a means to specifically inactivate cytonemes, the authors use two proteins in a mutant form, IRSp53 and Cdc42 (line 196). How is it known that these do not affect production and functions of Wnt5b and/or Ror2? Cdc42 for example is a small GTPase of the Rho family, which regulates signaling pathways that control many cellular functions.

Line 105, "in less than 4 min, we observed". After what, what is the zero time point?

Neither the pages nor the figures are numbered, which doesn't help in presenting a transparent view of this complicated manuscript. The discussion makes references to Spemann's legacy that are

uncalled for.

Referee #2 (Remarks to the Author):

This paper provides strong evidence for a previously unknown mechanism that can spatially organize non-canonical Wnt5B signaling in the zebrafish embryo. Authors performed many challenging experiments for both in vitro and in vivo systems to demonstrate that Ror2 (one of the Wnt5b receptors) and Wnt5b can interact in the signal-producing cells, and the Wnt5a/b-Ror2 complex is then delivered from the source to neighboring cells via cytonemes. Apparently, the Ror2-Wnt5b complex's exchange via cytonemes can induce autocrine and paracrine JNK signaling. Genetic experiments convincingly show that the Ror2-Wnt5b distribution via cytonemes is required for the paracrine JNK signaling downstream to the non-canonical Wnt pathway. Induction of non-canonical paracrine signaling via Wnt-carrying cytonemes was further supported by the suppression of the canonical Wnt pathway, commonly known to be inhibited by the activation of the non-canonical signaling. Genetic perturbation of cytonemes provides evidence that the cytoneme-mediated Wnt5B/Ror communication might be required for the early embryonic C&E, a morphogenetic process induced by the Wnt5a/b non-canonical signaling. This is an important finding, for the first time showing a mechanism by which non-canonical Wnt signaling can be distributed and organized in space and time. The manuscript is well-written, results are original and significant, but addressing a few additional gaps would improve the paper.

1) Authors suggested that the work presents a mechanism that challenges the idea of the long-standing concept of characterizing responsive and non-responsive tissues based solely on the expression of the receptors and that a revision of the long-standing concept is required. However, the activity of WNT ligands depends on their binding to various receptors and co-receptors, including 10 members of the Fzd receptor, required for both the canonical and non-canonical signaling pathways, and coreceptors/receptors like LRP5/6, ROR1/2, and RYK. To my understanding, receptor(s) and/or co-receptor(s) to which Wnt5b binds to induce responses is still controversial/not well characterized. Since the current investigation is limited to only Ror2, it is better to be cautious in suggesting "a revision of the long-standing concept". Decades of study have provided strong experimental evidence, showing that the long-distance interactions between the canonical receptors and ligands, expressed in discrete groups of cells, are required to induce spatiotemporal coordination of cells during morphogenesis. In contrast, non-canonical signaling has been shown to involve ubiquitously expressed cell junctional proteins/cell-recognition molecules, intracellular polarity complexes, and Ca²⁺ signaling, in addition to canonical ligands and receptors. How the spatial/temporal coordination/organization/activation of these non-canonical signaling components, which are expressed by all cells, is achieved is a mystery. I think this paper might contribute to this conceptual gap. So, instead of a claim for a paradigm-changing finding, the authors should clearly state and discuss that the paper investigates Wnt5b-induced non-canonical signaling induced via Wnt5b-Ror interactions and discovers one of the ways by which this non-canonical pathway can be organized in space.

2) In the paper, cells that overexpressed Ror2 and/or Wnt5B could deliver the Ror-Wnt complex to

Ror2-mutant cells and can activate Wnt/PCP signaling. This brings along a question about the WT expression and distribution patterns of Wnt5b and Ror2. The normal WT distribution patterns of Ror2 and Wnt5b relative to their sources could not be assessed from the text/results. Does the native Ror2 move from a producing region to its non-producing region in the WT embryo to shuttle Wnt5b? Do we see a defect in Wnt5b distribution profile in the absence of Ror2?

3) Wnt5b-Ror2 interactions and delivery: Due to the technical challenges, the authors had to rely on the over-expression of Wnt5b and Ror2 in cells. Under these experimental conditions, tiny puncta constituted of Wnt5b-GFP and Ror2-Cherry complex are visible on source cytonemes and on the recipient cell membrane. In recipient cells, the puncta are at or near the contact sites of source cytonemes and recipient cells. Since there are only a few puncta in the recipient cell and are observed only to be at or near the contact sites of Wnt-Ror-delivering cytonemes, it is important to clarify what is meant by the signal delivery. To be specific, It is unclear from only a few images shown if the Wnt5b-Ror complex is endocytosed in the recipient cells or is simply displayed on the recipient membrane as remnants of the cytoneme member that had retracted from the contact sites.

From Fig 1 and movies, it is clear that the cytoneme membrane is also delivered together with Wnt5b and Ror2 to the recipient cells. If so, a possibility is that any membrane-tethered signal overexpressed in the same cytoneme-forming cell can simply be delivered randomly to the non-overexpressing cells during the cytoneme membrane exchange. Based on Fig.1a, mCherry-marked cytonemes show intimate adhesion to recipient cells and can exchange membrane via cytoneme contact sites, irrespective of Wnt signaling. Do we observe similar membrane exchange in WT scenario without Wnt/Ror overexpression?

Do we need endocytosis of the Ror-Wnt complex to induce signaling? Any background information related to endocytosis/membrane display of the complex and correlation of this event with the induction of signaling in cells, if available, would be helpful.

4) About colocalization in Fig 1 d,e. Either Ror-mCherry or Wnt5b-GFP, when overexpressed in cells, mark the entire cytoneme membrane and can appear to be colocalized with any signal puncta that happen to be loaded on the cytonemes. Therefore, do we see similar source-membrane remnants/endocytosed puncta in WT cytoneme-recipient cells when Wnt5b-GFP/Ror-Cherry are not overexpressed? The specificity of the signaling process is then defined by events that control the cytoneme contact sites. It is also important to verify if the native Ror-2/Wnt 5b (e.g. by using antibody co-staining) can be dispersed from producing cells to non-producing cells, showing a similar profile.

5) Fig 2&3: The FLIM-FRET analyses are indicative that the mem-localized Wnt and Ror can interact with each other. These results are clear. However, it is appropriate to mention in conclusion that these proteins are overexpressed in the same cell and they interact under the specific experimental conditions. Where and when within a cell native Wnt and Ror2 meet for actions are probably unknown. For the lifetime-FRET assay, the interaction between overexpressed Wnt-Ror shows similar values to that of the non-specific mem-GFP-memRFP. Therefore, a better control FLIM-FRET

expt might be a mutant Wnt/Ror that does not interact or, alternatively, WT interacting partners that can not be sorted together in rafts/vesicles. Authors have utilized a mutant Ror-2 in experiments with FCCS in Fig 3, and results suggest an interaction between Ror-Cherry and Wnt5b-GFP, when overexpressed.

6) Line 190- Authors mentioned the "activation" of Wnt5b and Ror, but these experiments involved overexpression of these proteins in cells without any measurement of their downstream signaling activity. Does overexpression of these proteins in cells always induce signaling in cells?

Secondly, activation of Wnt/Ror was proposed to induce long cytonemes. A concern could be that the production of long cytonemes can simply be an effect of overexpression of these proteins, and removal of the basic cytoneme-modulators (eg, IRSP53 or CDC42 function) would simply block cytoneme formation. My guess is that the molecular consequences of Wnt5b/Ror overexpression in cells are not fully understood. Therefore, in addition to the overexpression, Wnt5b or Ror2-knockdown results are required to strongly conclude that Wnt-Ror2 interaction is required for long cytonemes.

7) Fig 4m-r. Does Wnt5a/b localize in Ror-2 mutant cytonemes? Secondly, it is surprising to observe many long cytonemes from Ror-2 mutant cells (4m), connecting to neighboring cells. This is in sharp contrast to the Wt cells shown. The formation of so many long cytonemes in mutant cells is not consistent with the claim that the activation/expression/interaction of Ror and Wnt is required to produce cytonemes as a signaling response in the producing cells. The authors should clarify or revisit this data.

8) line 230. Authors suggest that mutant Ror2 clones do not change the status of the JNK-reporter expression/distribution in the neighboring cells, unlike Wt Ror2 clones. However, a concern is that the JNK reporter is introduced to the embryonic cells via microinjection, which might distribute variable numbers of constructs to cells. Clearly, there is variation in reporter fluorescence detection among the neighboring cells where there are no Wnt-clones introduced. Moreover, the interpretation of the JNK activation is difficult as the only thing clearly visible to me is the nuclear stain. It is hard to understand the cytoplasmic stain as the recipient cell outlines are unmarked, and there are so many cells crowded within the same place. Although there are variations in nuclear signal among cells, a corresponding change in the cytoplasmic signal cannot be assessed from these pictures. It is unclear how the change in signaling can be assessed from these experiments unless pre- and post-transplantation levels of the reporter were measured.

8) Fig 5 and 6. Although the change in cytoplasmic localization of the JNK reporter is unclear from the images, a nuclear-localized reporter is clearly removed in recipient cells under the specified conditions that removed cytonemes, indicating cytoneme-dependent paracrine function of Wnt5b/Ror2 complex. Similarly, suppression of canonical Wnt signaling by the Wnt5b/Ror-induced PCP pathway is very clear. These results are exciting. While the spatial distribution of *lef* expression is used to report the effect of non-canonical Wnt/Ror signaling on the canonical Wnt signaling, it might be more direct to visualize *b-cat* staining under the conditions tested. In Fig 6, in situ Figure panels would be easier to interpret with the split color display.

9) Since the paper claims cell-cell movement of the preassembled receptor-ligand complex based on only Ror2 and Wnt5b, the Introduction should indicate the requirement of various other receptors/co-receptors for canonical and non-canonical Wnt pathways. Secondly, signal dispersion and signaling outcomes were known to be spatially coordinated, target-specific, and robust. It is unclear how the dispersion of the preassembled receptor-ligands might determine the specificity and spatial coordination during signaling and morphogenesis? A discussion point would be helpful.

Few minor points:

1) Fig.1 Clear image of Wnt 5a/b on cytonemes. 1b. I suggest not distinguishing between filopodia and cytonemes at this point. The non-Wnt5a/b cytonemes could be for another signal or could be the precursor of the Wnt5a/b cytonemes prior to ligand loading. It is clear that the Wnt5 or Ror cytonemes are longer, and they connect between cells.

2) Line 194: "To block cytoneme formation in zebrafish without interfering with Wnt signaling, we used the mutated protein IRSp534K and the dominant negative Cdc42T17N 14,17." It might be critical to mention non-interference with the canonical signaling here, especially since cytoneme induction appears to be caused by the non-canonical Wnt5b-PCP signaling but not by the canonical pathway.

3) Line 258-260 (Fig 4i needs to be 5i). Citation of Fig 5G,L panels in the text needs to be verified.

Referee #3 (Remarks to the Author):

Review Nature WNT-5b-GFP/ROR2 in cytonemes

The authors present a highly provocative concept of cytonemal WNT signaling where the major and most exciting hypothesis is the transfer of a signaling WNT5B/ROR2 complex from the cytoneme of one cell to a target cell. The authors have used advanced microscopy techniques to make the original observations and employ complicated but sound transplantation paradigms in an attempt to support their ideas. I appreciate first of all the concept, which is not only provocative but also interesting. Cytonemal transfer of active signaling complexes is at the border of detection limits for current technology.

Some of the technologies are difficult to quantify and the authors compensate that in part by presenting movies in the extended data files. Some of the data in the first part of the manuscript appear as circumstantial observations because the information of number of experiments performed is not provided, which indicates that the reporting routines on number of independent observations must be improved. Furthermore, I feel that the cell biological fundament of the novel signaling concept is not clarified. Where is the complex formed (in the golgi/ER; at the plasma membrane), how is it transported (in intracellular transport vesicles or at the cytonemal membrane) and what process is employed by the cell to transfer the complex (some kind of transcytosis). These basal questions remain unanswered and the authors instead investigate potential signaling crosstalk in vivo, data that are very difficult to interpret with relevance to the original hypothesis of complex transfer.

While I remain excited about the original finding, I do not see that the data and especially the

controls that the authors present really form a firm foundation for their conclusions.

Detailed comments:

1. The authors need to validate the biological activity of the WNT-5b-GFP fusion protein thoroughly.
2. Overexpression of ROR2 in HEK293 cells induced a “spiky” phenotype in our hands (unpublished). I wonder therefore what the hen and what the egg is in the observation of cytonemes as signal transducers. Is the proposed widely applicable or a special and potentially artificial consequence of ROR overexpression?
3. The WNT field generally refers to RORs as WNT “coreceptors” suggesting that they act in concert with FZDs. While, I am not necessarily convinced that RORs do not act independently of FZDs, I suggest to at least mention FZDs, which are the main WNT receptors also mediating PCP signals (especially in the introduction). This becomes even more obvious when convergent extension movements are introduced as a consequence of PCP signaling – a complex phenomenon that surely involves FZDs signaling and very likely WNT5B-FZD signaling.
4. Already in the abstract, the authors refer to “active WNT5B/ROR2 complexes”. How do the authors validate the complex activity justifying this statement?
5. What is a “WNT cytoneme”? Please reword! The manuscript would benefit from a clear and early definition of cytoneme vs filopodium.
6. Please avoid the term WNT ligands. While WNTs are ligand of e.g. FZDs and RORs, the combined term WNT ligand is pharmacologically confusing.
7. The authors must present some kind of validation of antibody specificity for the main target antibodies WNT5A-B, rabbit PolyAb, ProteinTech, 55184-1-AP; & ROR2 629 (D3B6F). The vendor validation is not sufficient for long ranging statements such as the one put forward in this manuscript. For example the anti-ROR produces substantial nuclear immunoreactivity in Fig. 1C, which in my eyes seems unspecific but on the other hand, it is absent in the ROR KO cells. Please comment and explain.
8. The authors call ROR2 the cognate receptor of WNT5B. This is an overstatement given that many of the FZDs also bind WNT5B.
9. The colocalization data shown in Fig. 1D can by no means be interpreted as a confirmation of physical interaction between ROR2 and WNT5A. Colocalization is not interaction! In fact it remains elusive if the WNT is intra- or extracellular. It could be localized in a transporting vesicle, potentially in the same vesicle as ROR. The term WNT/ROR cluster is in my eyes an overstatement.
10. The central observations in Fig. 1 e, g, f, h are not accompanied by any form of quantification, which renders them circumstantial. It remains unclear how many independent experiments were performed in Fig. 1. E.g. n=17 (cells) do not represent independent observations that would justify statistics or satisfy general rules of reporting.
11. The settings for confocal imaging must be reported. Are the optical sections for the different channels equally thick? If not colocalization or lack thereof does not allow too many conclusions.
12. The term “Wnt/PCP cytonemes” is an unfortunate construction to name a cell organelle.
13. The ligand binding domain of ROR2 is extracellular. The authors need to address (experimentally) where the receptor-ligand complex is formed and how it is transported. I have difficulties to envision that the receptor-ligand complex is formed in the ER/Golgi and that ligand binding would be maintained when the receptor is exposed to the cell membrane with the ligand on the outside of the cell (infinite dilution of the ligand should/could lead to its dissociation). The other alternative is that

WNT binds ROR on the cell surface and that the ligand receptor-complex is transported along the cytoneme. In this case, however, the ROR/WNT-positive dots in the cytoneme are difficult to explain. These remind more of transport vesicles. These aspects pose a major shortcoming of the study and require clarification.

14. Fig.2 c-f: The WNT5B construct is most likely a GFP fusion, that should be indicated in the figure everywhere (now it appears that the legends state WNT5B and WNT5B-GFP). Abbreviations in Fig. 2g, f, I are not consistent. Is mGFP the same as memGFP? Or is mGFP monomeric GFP? The nomenclature/abbreviations must match between figure and graph. It is unclear whether the data originate from one or several independent experiments. Different cells are not independent experiments, rather technical replicates. The authors need to confirm that statistics are suitable when the groups contain different independent observations. inhibitor.

15. For the FLIM FRET approach the authors report GFP lifetime, FRET efficiency and donor-acceptor distance. First of all the authors should explain to the general audience what these values mean in terms of protein-protein proximity and orientation. Second, I understand that the distance calculations are based on the Förster distances of the eGFP-mCherry FRET pair (54.2 Å). However, this is valid for the fluorescent proteins, not for fusion proteins fused to these fluorescent proteins. Thus, the donor-receptor distance calculations are at best estimates. Furthermore, FRET efficiency, which the distance calculations are based on, is dependent on both distance of the FRET pair and the orientation of the two resonating fluorophores. This orientation is surely different, when investigating GFP vs mCherry or ROR-mCherry vs WNT-GFP.

16. The experiments shown in Figure 2 are poorly controlled. This statement is based on the fact that the maximum FLIM FRET between memGFP and mem-mCh is assessed between proteins that are not showing any specific interaction. The resonance energy transfer appears between randomly colliding fluorescent proteins in a totally overcrowded membrane. This random interaction by no means sets the maximum value for specific interaction. One could for example create an artificial construct containing both eGFP and mCherry for optimal energy transfer. Alternatively, two proteins with a known interaction should be used. Similarly, the negative control of memGFP and cyto-mCh assesses random collision of proteins expressed in two different compartments. Here again proteins that are not known to interact fused to eGFP and mCherry should be used instead.

In relation to the design of these control experiments, the authors need to match the system under investigation. This is a single transmembrane protein (ROR) and a secreted lipoglycoprotein potentially bound to a carrier such as afamin (WNT).

In this context, I would also like to get back to one of my previous points above, emphasizing the location of the receptor and the ligand in the cell. This needs to be reflected in the control experiments. Is the ligand-receptor complex embedded in the plasma/cytoneme membrane or is the complex transported along the cytoskeleton in transport vesicles.

17. The Figure 3a presents a very nice positive control in my eyes, where the membrane-embedded receptor is bound by a fluorescently-tagged Vhh construct. Splendid! The control in Fig. 3b, however, suffers from the same shortcomings as mentioned above (reflected by the absence of cross correlation). Furthermore, the control with the Delta-CRD-ROR construct shown in Fig. 3f is highly relevant. In that particular micrograph, however, the WNT-GFP fluorescence is hardly visible. This leads back to my question about the number of independent experiments that have been performed for every observation in the paper. This information must be provided in the figure legends.

18. Figure 3/FCCS: The number of independent experiments is not reported. The authors must justify and explain, why the ROI for FCCS is placed across the membrane vs on top or the bottom of the cell.

How large is the ROI/effective volume for FCCS? Is the ROI ($0.65 \times 10^{-9} \text{ nm}^3$) relevant in relation to the diameter of a cytoneme?

19. The membrane compartment recorded in a cytoneme vs the plasma membrane is completely different. I fear that the diffusion behavior in the restricted cytoneme area vs large plasma membrane are different and make the FCCS values difficult to compare.

20. The calculation of a K_d value from FCCS measurements must be clarified. Especially the “diffusion with triplet”! and “pure diffusion” models. What does “The dissociation constant (K_d) was analysed based on these fitting values for every measurement.” mean? I would like to see the raw data for the K_d calculation in the supplement/extended data. Are binding curves hyperbolic or linear? The authors must be clear that FCCS in 3 dimensional diffusion can be applied to the apparent 2 dimensional diffusion in membrane protein complexes. For inspiration, I suggest to look at <https://journals.asm.org/doi/10.1128/MCB.00087-14>. References for the analysis of receptor-ligand K_d s in 2 D membrane diffusion by FCCS should be given. How do the K_d values match those previously reported for WNT-ROR interactions (e.g. recent work from the Lemmon lab?).

21. Residence time: The presented concept requires a long residence time of the ligand at the receptor. While binding dynamics of WNTs to CRDs of membrane receptors has not been systematically addressed there are indications in the literature that the residence time is long. However, given the low K_d s presented in this study (300-500 nM, which corresponds to round about 16 $\mu\text{g/ml}$ WNT!!! This is a concentration that is not reachable in cell culture with addition of recombinant WNTs!), it appears unlikely that the residence time is particularly long. In fact, residence time is in part defining affinity since it substantially affects the ligand off rate. The calculated K_d values in the range of hundreds of nM of WNT will NEVER be reached in vivo and thus do most likely not reflect a biologically meaningful affinity. As an example, binding affinities of WNTs to FZDs in receptor overexpression systems have been calculated to be in the range of low nM concentrations, more like 2-5 nM employing different methods.

22. Lane 133 – the authors claim that their mCherry-ROR2 is biologically active, referring to Ext Fig 4c. How was biological activity of the construct validated?

23. Lane 145-147: The authors argue that their FLIM-FRET data argues for ROR-WNT5B interaction. Given the criticism of controls, I do not see that the experiments support the claim. A comparison of FLIM FRET with the Vhh construct (max FRET) and the Delta-CRD-ROR2 (min FRET) should be done similar to what has been done for the FCCS set up.

24. Fig. 4: In the quantification of filopodia length it appears that ROR is active but that addition of WNT5B does not substantially elevate the effect. When it comes to filopodia number WNT5B in combination with ROR inhibits the effect. This discrepancy is not reflected in the text of the result section.

25. The transfected constructs in Fig. 4 are untagged? How is it controlled that cells express both ROR and WNT? Or are the constructs tagged with the same fluorescent proteins as in Fig. 3? If so that needs to be indicated in the figures and legends.

26. SP600125 is a JNK inhibitor, not a JNK signaling. According to <https://www.pnas.org/doi/full/10.1073/pnas.251194298>, the JNK inhibitor has a K_i value of 190 nM with a 20 fold selectivity over other kinases. Tocris provides values of $IC_{50}=40-90$ nM. However, the authors use the inhibitor at a concentration of more than 200 fold the K_i (40 μM), which basically renders it unspecific or at least non selective. I would argue that it is not surprising that delicate structure such as cytonemes collapse at this concentration and that has relatively little to do with WNT-induced and ROR-mediated signaling. Also, the authors should check the size bar in Fig. 4k. The

cells in the DMSO control look much larger than in the JNK inhibitor-treated conditions. This could either be a different magnification or inhibitor-induced cell shrinkage. concentration response curve could assist to distinguish between target specific and unspecific effects.

27. I could not find the reference Brunt et al 2021 in the reference list. Is that a reference to the JNK activity reporter? If not, a reference for the reporter should be included.

28. Fig. 4: How do the authors relate te observed effects on the JNK reporter with the transplantation paradigm to ROR and WNT signaling. There is no experimental aspect (blockade of ROR) that argues for an involvement of WNT/ROR signaling in the activation of JNK in surrounding cells. This argument is based on pure and relatively weak correlation, so far.

29. Figure 5: Are the images for Fig. c, d, g, o taken with identical settings? Background seems higher, but this could obviously be due to a wide-spread cytosolic expression of the reporter. How is the ratio determined in this case, when the nucleus is difficult to identify?

30. The scale bar in Fig. 5 is 20 μm . It surprises me a bit that the whole field of view (about 80 μm) is equally and evenly affected with regard to the JNK reporter. If cytonemal signaling (cytonemes are not visible despite the memGFP in the transplanted cells) would be responsible for the effect, I would expect a gradient effect or cells that are in contact with memGFP-positive cytonemes showing high JNK signaling. In this scenario it appears more likely that a released and diffusible factor X is mediating the effect over long relatively distance. In addition, this model is not suitable to resolve the implications of the original concept of transfer of a functional WNT/ROR complex. In lane 249-253, the authors argue that ROR expression and ROR/WNT expression show differences over distance referring to the graphs Fig. 5e, f. The p values presented refer most likely to comparisons to baseline, not between the distances. By eye, the effects over distance in these graphs look very similar. Thus, I do not feel that this argument holds.

31. Lane 263-262: Functional CDC42 is most likely relevant for the formation of any cytoneme, not only those that are assessed in the context of WNT signaling

32. The remaining figures do not – in my eyes – address the interesting concept of ligand-receptor complex transfer, which in fact is the provocative and exciting aspect of this work. I would rather suggest the authors to focus on that aspect and to find a biochemical or biophysical way to detect complex transfer that investigating signaling crosstalk in vivo that deviates from the central message.

Referee #4 (Remarks to the Author):

In this manuscript Zhang and collaborators are reporting on their studies of cytoneme-mediated Wnt5b/Ror2 complexes intercellular Wnt/PCP signaling in zebrafish embryos. Whereas cytoneme mediated intercellular signaling has been documented in *Drosophila* and other systems, including the recent work by these authors showing cytoneme-mediated Wnt8 transport to mediate canonical/ β -catenin activity in zebrafish (Brunt et al., *Nature Communications*, 2021), the novel aspect of the signaling mechanism proposed here is that the transported complexes retain activity and a receiving cell activates Wnt5b/PCP signaling without expressing the cognate Ror2 receptor. However, these conclusions are not fully supported by the data presented in the current manuscript. There are concerns with the conducted experiments and the interpretation of the results.

My overarching concern is the Wntb/Ror2 signaling complexes proposed to be loaded on cytonemes and transported on them to a receiving cell are not physiological, as the key colocalization, FLIM-FRET and FCCS experiments are performed using overexpression of the component proteins. Second, the role of Ror2 is studied using dominant negative form of the receptor and transplanted mutant clones, while the relevant phenotypes of the zebrafish mutants are not analyzed/reported. Specific questions and comments follow.

1. Figure 1 shows colocalization of Wnt5b/Ror2 puncta. The number of cells and cytonemes analyzed should be provided on panel "b". The co-localization of a-Wnt5a/b and Ror2 shown in figure 1d is not compelling without seeing individual color channels and needs to be quantified. Moreover, one of the components is overexpressed. Additional evidence of co-localization at endogenous protein levels is needed such as proximity ligation assay.
2. The detection of Wnt5a/b needs to be performed on relevant cells and tissues. In Fig. 1 fibroblasts and epiblast cells are used, while the authors wonder "how these lipid-modified ligands are precisely disseminated in embryonic tissue to regulate complex tissue movements like C&E". Much work is being done on epiblast cells at 8 hpf, but in which embryonic position? How does this position relate to the normal morphogenetic processes?
3. Detection of Wnt5/Ror2 puncta in *ror2* KO zebrafish fibroblasts. A more complete characterization of the *ror2* mutant cell line should be performed including a western blot.
 - a. Only one mutant cell is shown with α -Ror2 Ab positive puncta. Quantifications from larger number of cells from more than one experiments should be provided.
 - b. Additional controls should be provided:
 - i. α -Ror2 Ab staining of *ror2* mutant cells not co-cultured with WT cells.
 - ii. α -Ror2 Ab staining of *ror2* mutant cells cultured in conditioned media from WT cells, to ensure that Ror2 or Ror2/Wnt5 vesicles are not transferred as secreted vesicles rather than "handed over" to adjacent cells.
4. The conclusion about the Wnt/PCP filopodia is premature. In line 189 the authors write "... we overexpressed the indicated PCP constructs (i.e. Wnt5b and Ror2) in clones in the zebrafish embryo and quantified the length and the number of filopodia per cell". At best, these data support the conclusion "Wnt5b and Ror2 can be loaded on signalling filopodia", but the following conclusions "referring to them as Wnt/PCP cytonemes" and "the Wnt5b/Ror2 positive cytoneme tip is delivered to the neighbouring cell during zebrafish gastrulation" cannot be inferred from the current data.
5. In subsequent lines, the authors state "We found that activation of Wnt5b, Ror2, and Wnt5b/Ror2 led to the formation of fewer but much longer filopodia" and later (line 193) "We asked if the Wnt/PCP-induced filopodia are dependent...". And Lines 201-202 "Therefore, we conclude that Wnt/PCP is required to induce long cytonemes in the source cell." Here the authors move from Wnt5b and Ror2 overexpression to "Wnt-PCP pathway activation without analyzing any Wnt-PCP pathway activity readouts. Later JNK activation is used as the only Wnt/PCP pathway activity readout, but the key hallmarks of Wnt/PCP signaling during gastrulation: cell shape, cell body orientation are not analyzed here.
6. The specificity of Wnt5b as a Wnt/PCP ligand is not clear. Other Wnt ligands should serve as controls. First, Wnt8 ligand that can also bind to Ror2 but stimulate canonical Wnt signaling should be used and tested for the ability to activate JNK. Second, another Wnt ligand that cannot bind Ror2 should be used as a negative control. It would be particularly interesting to analyze Wnt11, an essential Wnt/PCP ligand during zebrafish gastrulation.

7. To further probe the conclusion that “Wnt/PCP is required to induce long cytonemes in the source cell” and considering the earlier studies of the authors that Vangl2 (the conserved PCP pathway component from *Drosophila* to zebrafish to mammals) promotes the formation of long cytonemes to enable Wnt/b-catenin signaling (Nat. Comm., 2021), the experiments with Wnt5b/Ror2 cytoneme formation should be performed in *vangl2* zebrafish mutants lacking both maternal and zygotic Vangl2 expression.

8. Another key thesis of this manuscript is that Ror2 signaling mediated by the cytoneme-transferred Wnt5b/Ror2 complexes activate paracrine JNK signaling. This is tested by employing a fluorescent JNK kinase translocation reporter, JNK-KTR-mCherry that relocates from the nucleus to cytoplasm upon JNK activation and a newly generated *ror2* zebrafish mutant line. The observation that *ror2* mutant cells transplanted into WT embryos expressing JNK-KTR-mCherry reporter do not alter the reporter intracellular distribution in the neighboring cells whereas WT cells induced JNK-KTR-mCherry activation when transplanted into WT host, is interpreted that that endogenous Ror2 signalling in the grafted cells is required for paracrine JNK activation via the cytoneme-mediated transfer. Similar experiments with transplanted Wnt5b expressing clones in which cytoneme formation is experimentally impaired using the dominant-negative mutants IRSp534K and Cdc42T17N leads the authors to conclude that paracrine Wnt/PCP signalling is strongly dependent on cytoneme appearance in zebrafish development. There are several concerns with these experiments and their interpretation. The *ror2* mutant line harboring a premature stop codon (truncated Ror2 protein) used in these experiments is not phenotypically characterized. What is the phenotype of *ror2*-deficient mutant embryos with respect gastrulation, in particular convergence and extension phenotypes ?

a. Morphometric analyses should be performed to monitor anteroposterior axis extension and mediolateral narrowing of embryonic tissues.

b. The hallmarks of Wnt/PCP signaling during zebrafish gastrulation, i.e. mediolateral cell elongation and asymmetric Vangl2/Pk localization should be assessed in these mutants.

c. How does loss of *ror2* function affect JNK signaling? The JNK-KTR-mCherry reporter activity should be compared between WT and *ror2* mutant embryos throughout gastrulation to understand the requirement for the endogenous *ror2* activity in JNK signaling and also to identify appropriate stages of development to perform functional experiments.

d. In the experiments quantifying JNK activity, nuclei should be labeled with Histone-FP and the ratio of nuclear JNK versus Histone could be used as an additional measure of JNK activity. E.g. how do we know that there is a cell where the arrow points in Fig. 5b? And how the authors could measure the nuclear versus cytoplasmic level of JNK activity in a cell in which nucleus is not visible?

9. Expression of the dominant-negative mutants IRSp534K and Cdc42T17N is used to impair formation of cytonemes in the transplanted cell clusters expressing Wnt5b what leads to the conclusion that cytonemes are needed/mediate Wnt5b/Ror2s paracrine signaling and JNK activation. Against this notion, the authors report that ubiquitous expression of IRSp534K in the zebrafish embryos does not interfere with JNK signaling. This raises the concern that the phenomena observed with transplanted cell clusters expressing Wnt5b/Ror2 do not reflect physiological processes but rather the effects of overexpressed proteins. Impairment of cytonemes in entire embryos should lead to altered JNK activity and cell polarity if cytoneme-mediated Wnt5b/Ror2 paracrine signaling is required for Wnt/PCP regulated gastrulation morphogenesis. Indeed, in these experiments a control clone expressing memGFP does not interfere with JNK signal activation and only clones overexpressing Wnt5b/Ror2 do. A more appropriate or additional experiment would be to

transplant cells from embryonic regions that do or do not express these proteins. In zebrafish, like in *Drosophila* Wnt/PCP signaling and the processes it regulates are exquisitely sensitive to both loss but also gain of function.

10. The finding that challenges the classical notion of intercellular signaling whereby a cell competent to respond to a signal produced by a different cell expresses the cognate receptor is that cell receiving cytoneme-mediated Wnt5b/Ror2 complexes do not require Ror2 activity. This conclusion is reached based on the experiments in which a dominant negative form of the Ror2 receptor, the kinase-dead mutant Ror23i (Hikasa et al., 2002) is expressed in receiving cells what blocks JNK signal activation. Yet such cells can activate JNK signaling when cells expressing Wnt5b/Ror2 are transplanted in their environment. It is puzzling why to inactivate Ror2 in these experiments a dominant negative form of the receptor is used given the ror2 mutant line described in the manuscript? Such experiments should be also performed using the ror2 mutant line and JNK reporter.

11. The ror2 mutant line provides additional opportunities to perform experiments that would alleviate the concern about overexpressing fluorescent Ror2 fusion proteins. When ror2 phenotypes are analyzed, the dose of mCh-Ror2 that rescues these phenotypes (i.e. physiological dose) would be identified allowing to perform some of the colocalization studies in more physiological setting. Alternatively, although admittedly more difficult, knock-in technologies now afford generating fluorescent fusion proteins in endogenous genes.

12. When concluding that a receiving cell does not require Wnt5a/b receptor one needs to consider that Ror2 is only one of the receptors binding Wnt5 ligands during zebrafish gastrulation. In fact Ror2 receptors are thought to work with Frizzled receptors. Is the reception of Wnt5b/Ror2 independent of any receptors or could it be mediated by Frizzled, Ptk7, Ryk or other membrane receptors implicated in Wnt/PCP signaling?

13. To probe the role of Vangl2, which is known to be required for mediating JNK signaling, the authors deploy expression of the dominant-negative mutants Vangl210A rather than well-characterized vangl2 zebrafish mutant lines. Such orthogonal experiments should be performed.

14. The requirement for the role of the membrane-localized Vangl2 in transducing Wnt5b/Ror2-cytoneme-mediated signalling in the receiving cell is puzzling given that Ror2 is not required in the receiving cells? This amplifies the point #11 that the more typical Wnt/PCP membrane complexes composed of Fz/Vangl/Celsr could be required for Wnt5b/Ror2-cytoneme mediated transfer?

15. Another key conclusion (line 288) "Cytonemal-delivered Wnt5b/Ror2 complexes can block Wnt/b-catenin signaling" is also questionable. Here, small cell clusters expressing Wnt5b, Ror2 or both are transplanted into the zebrafish margin and Wnt/b-catenin pathway activity is assessed using lef1 expression in the transplanted cells and their neighbors. These experiments show that overexpressed Wnt5b/Ror2 CAN interference with b-catenin activity in autocrine and paracrine manner, and the latter is dependent on cytonemes. Surprisingly to address whether ENDOGENOUS genes are involved, Ror23i and Vangl210a mutant protein overexpression is deployed. As indicated above, a requirement for the endogenous components should be tested using the mutant lines: is lef1 expression altered in ror2 mutants the authors generated? Would cell clusters transplanted from the equivalent embryonic positions from ror2 mutants to the margin of wild-type embryos have such effects.

16. It is stated in the summary that "cytoneme-dependent spreading of active Wnt5b/Ror2 affects convergence and extension in the zebrafish gastrula". These experiments are also problematic as they test the ability of cell clusters overexpressing Wnt5 and Ror2 rather than using mutant cells.

The ability of many overexpressed proteins to interfere with convergence and extension is known

Additional points:

- Line 284 "This data suggests " data is a plural noun.
- The figure numbers should be provided on the figures – their absence made reviewing the manuscript rather cumbersome.

Author Rebuttals to Initial Comments:

Referees' comments

All referees' comments and suggestions have been addressed. We have listed the referees' points, including our responses in *italics*.

Referee #1 (Remarks to the Author)

1.1 The phenotype of the genes involved. It is stated that Wnt5b and Ror2 are implicated in planar cell polarity but that poorly supported and not at all by classical loss of function genetics. In their previous paper (eLife, 2018), a Morpholino-based Ror2 knock-down experiment is shown. Otherwise, the Ror receptors are implicated in convergence extension in *Xenopus*, but without genetic data in that organism.

Surprisingly, in the current submission on line 220, the authors actually describe that they made a mutant Zebrafish line with a loss of function mutation in Ror2 made by Crispr technology. They do experiments with cells obtained from these animals but don't describe an organism phenotype in developing fish that are homozygous or heteroallelic mutant for the gene. Why not, this is such an important piece of information. Are there planar polarity phenotypes.

Similarly, there is no information on possible mutant phenotypes of Wnt5b. It would not be hard to generate Crispr alleles of Wnt5b, a gene that the group has been working on since at least 2018.

The ror2 mutant zebrafish: We agree with the reviewer that a detailed analysis of the Ror2 mutant zebrafish line was not provided in the original manuscript. We, therefore, added a detailed description of the ror2 mutant (ror2^{t13}, Fig. 3, Extended Fig. 4). As the observed phenotype of the ror2^{t13} mutant fish is rather mild, we generated a maternal-zygotic Ror2 mutant (MZ ror2^{t13}). Also, here, the phenotype was subtle. We speculated that Ror2 is functionally compensated by Ror1, the second Wnt-binding receptor tyrosine pseudokinase. Therefore, we generated F0 ror1^{crispant} embryos in the MZ ror2^{t13} background. The ror1^{crispant}/ror2^{t13} mutants show a shorter anteroposterior axis of the body, a rippled notochord, tail defects, and defects in the cardiovascular system, which we described and quantified in detail (Fig. 3, Extended Fig. 4). The phenotype of the ror1^{crispant}/ror2^{t13} mutants resembles defects described in zebrafish wnt5b/pipetail mutants (Kilian et al., 2002, Westfal et al., 2003) and the Wnt5a/Wnt5b^{crispant} mutants, which we generated (Extended Fig. 4).

To provide further evidence for specificity, we co-injected 200ng of mRNA for Ror2, which rescued the phenotype to a substantial extent. In parallel, overexpression of wnt5b mRNA (200ng) could not rescue the phenotype. Furthermore, this suggests that the following experiments, in which we used a similar amount of mRNA (100ng – 200ng mRNA), were conducted under a roughly endogenous level.

*The wnt5a and wnt5b fish mutant zebrafish: In the last 20 years, there have been numerous reports published describing the zebrafish pipetail (ppt) mutant phenotype in detail. The ppt mutant was first described in the Tübingen screen in 1996 (Hammerschmidt et al., 1996, Development) and linked to a mutation in wnt5b (Rauch et al., 1996; Cold Spring Harbour Symp Quant Biol). A detailed phenotypic analysis was provided by the Heisenberg lab showing that ppt/wnt5b is required for regulating convergence and extension (C&E) movements in mesendodermal as well as ectodermal regions. Furthermore, ppt/wnt5b influences cell elongation and C&E movements in posterior parts of the gastrula (Kilian et al., 2003, MoD). In the same year, the Sularski lab demonstrated that maternal-zygotic ppt/wnt5b mutant show an extremely shortened axis, undulating notochord, and tail defects (Westfall et al., 2003, JCB). The detailed analysis also showed a dorsalized mutant phenotype with partial axis duplication, including an expansion of somites, shortened and twisted tail. The observed phenotype of ppt/wnt5b mutants can be explained by essential functions on cell polarity, C&E, and, in addition, by a repressive function of Wnt5b on Wnt/β-catenin signaling. Further aspects of the mutant have been characterized by the labs of Tada, Wilson, Solnica-Krezcel and others and reviewed e.g. by our lab (Rogers and Scholpp, 2022, Semin Cell Dev Biol). These phenotypes match well with the phenotypes observed in the Wnt5a/Wnt5b double crispants (Extended Fig. 4). Although most of the embryos show a shorter anteroposterior body axis, tail defects and defects in the cardiovascular system, we did not observe a fusion of the eyes, which would indicate an alteration of Wnt11 signaling as seen in the wnt11/silberblick mutants. Furthermore, we were not able to rescue the Wnt5a/Wnt5b *crispant* mutants with micro-injection of 200ng Ror2 mRNA (Extended figure 4f). Wnt5^{crispant}/MZ ror2^{t13} mutants show a stronger phenotype compared to Wnt5b^{crispant} and MZror2^{t13} mutants, suggesting an additive effect of Wnt5b and Ror2. In summary, these results further suggest that Wnt11, together with an excess of Ror2, is not able to rescue the lack of Wnt5a/Wnt5b regarding C&E. This notion is supported by our in vivo FCCS analysis suggesting the direct interaction of Wnt5b to Ror2.*

1.2 Having loss of function alleles of these two main players, Wnt5b and Ror2, will also allow for testing the specificity of the antibodies used. This is always a concern, but if the staining is absent in a mutant, this can be resolved.

We agree with the reviewer that such zebrafish mutants would be beneficial to characterize the specificity of the Ror2 and the Wnt5a/b antibody used in this work. Therefore, we used our ror2^{-/-} PAC2 cells and wnt5a/b^{crispant} cells and performed an IHC analysis (Extended Data Fig. 1). In comparison to the control cells, we found no detectable signal for Ror2 in the KO cells and a significant reduction in the Wnt5a/b^{crispant} cells, suggesting high specificity of the antibodies used in this work. We also used overexpression of the respective genes in PAC2 cells to further demonstrate the specificity of the antibodies (Extended Data Fig. 1).

1.3 In this context, it should be mentioned that the authors use antibodies made against Wnt5a/b (line 80), and in the paper sometimes refer to Wnt5 (not Wnt5b) as being involved in PCP signaling (line 75), sometimes to the Wnt5a/b protein (line 90). This tends to be confusing.

We replaced “Wnt5” by “Wnt5a/Wnt5b” in the text.

1.4 A second major experiment that is missing is a biochemical and cell biological characterization of the Wnt5b/Ror 2 complex. It is a mystery how a complex of a ligand and its receptor would be taken up by a cell without any cognate receptor-like molecule. How does a receptor protein complex get into a cell from the outside, adopt a conformation to signal, which needs to happen over a membrane? As this claim is the centerpiece of the manuscript, and at the same time, without any precedent, it is reasonable to ask for more experiments. Can the complex be purified and added to cells? Is the complex attached to membranes produced by the cytonemes? If it is membrane-bound, the complex should be sensitive to detergents. The authors are using a cultured cell system, PAC2 cells and this might be adapted to study the cell biology of the uptake of the Wnt5b/Ror 2 complex.

To address the transport of the ligand-receptor complex during cytoneme-mediated trafficking, we expressed Wnt5b-GFP, GFP-Ror2, and Ror2-GFP in PAC2 fibroblasts and co-cultured these with cells stably expressing a secreted nanobody coupled to mCherry (Fig. 1h-l, Extended Fig. 2). After 24h of co-culture, we find Wnt5b-GFP signal co-localizing with secVhh-mCherry on cytonemes, similar to GFP-Ror2. However, Ror2-GFP does not co-localize with secVhh (Fig. 1j), suggesting that the ligand and the N-terminal part of the receptor face the extracellular side, which is accessible for secVhh. The C-terminal Ror2-GFP fusion shows no co-localization with secVHH-mCherry, suggesting that the secVHH cannot penetrate the plasma membrane on its own. We further provide evidence that the uptake of the Wnt5b/Ror2 ligand complex is Dynamin dependent in PAC2 fibroblasts and in the zebrafish embryo (Fig. 1l, n, Extended Fig. 2). Finally, we conclude that Wnt5b/Ror2 complexes can be loaded on and are transported along the cytonemes towards the tip. After contact formation of the Wnt/PCP cytoneme, the Wnt5b/Ror2 positive cytoneme tip is handed over to the neighboring cell and endocytosed in a Dynamin-dependent way (Fig. 1k).

1.5 As an argument that active signaling occurs, evidence is shown that the Ror2 molecule ends up in the nucleus of the target cells. This is puzzling. Ror2 is a receptor tyrosine kinase, and these molecules in general do not end up in the nucleus. The authors refer to a relatively obscure publication that would support this observation (Carbone, C. et al. Adipocytes sustain pancreatic cancer progression through a non- canonical WNT paracrine network inducing ROR2 nuclear shuttling. Int. J. Obesity 42, 334-343 (2018)). However, the data in that paper showing that Ror2 becomes nuclear are not strong and the case becomes yet weaker reading that Carbone et al seem to think that Ror2 is “a 7 transmembrane receptor”. It is not as we all know, Ror2 is a single transmembrane tyrosine kinase equipped receptor.

*In our experiments, we detect a signal for Ror2 at the plasma membrane and in the nucleus of PAC2 fibroblasts (Fig. 1 c, g). We then compared the localization of the Ror2 antibody (D36BF) in PAC2 fibroblasts with a second antibody against human Ror2 (H1 - sc374174) in human fibroblasts and find a similar signal at the plasma membrane and in the nucleus (data not shown). In contrast, in the embryo, we find that only some cells show a nuclear localization of Ror2, whereas others do not. Therefore, we agree with the reviewer and focus on the presence of Ror2 protein at the membrane to identify the *ror2* KO cells. We observe that *ror2*^{-/-} PAC2 cells show no signal for Ror2 protein at the membrane compared to a detectable signal in WT cells (Extended Fig. 1). We have changed that in the text and in the figures.*

1.6 As a means to specifically inactivate cytonemes, the authors use two proteins in a mutant form, IRSp53, and Cdc42 (line 196). How is it known that these do not affect the production and functions of Wnt5b and/or Ror2? Cdc42, for example, is a small GTPase of the Rho family, which regulates signaling pathways that control many cellular functions.

To visualise the cell morphology and check for alteration in the production of Wnt5b and Ror2, we transfected PAC2 cells with Cdc42^{T17N} and IRSp53^{4K} and stained for Wnt5a/b and Ror2, respectively (Extended data Fig. 5c-f). Although we find a strong reduction of filopodia (as described in our previous publications (Stanganello et al., 2015; Mattes et al., 2028; Brunt et al 2021), we find no detectable difference in the expression levels for Wnt5a/b and Ror2. In contrast, we find a stronger signal in the cells suggesting a lack of secretion due to impaired cytoneme formation (Extended data Fig. 5e,f).

1.7 Line 105, “in less than 4 min, we observed”. After what, what is the zero time point?

We have changed the text accordingly.

1.8 Neither the pages nor the figures are numbered, which doesn't help in presenting a transparent view of this complicated manuscript.

We have added a numbering to the figures and pages.

The discussion makes references to Spemann's legacy that are uncalled for.

We have added the appropriate reference and expanded this point in the introduction.

Referee #2 (Remarks to the Author)

2.1a Authors suggested that the work presents a mechanism that challenges the idea of the long-standing concept of characterizing responsive and non-responsive tissues based solely on the expression of the receptors and that a revision of the long-standing concept is required. However, the activity of WNT ligands depends on their binding to various receptors and co-receptors, including 10 members of the Fzd receptor, required for both the canonical and non-canonical signaling pathways, and coreceptors/receptors like LRP5/6, ROR1/2, and RYK. To my understanding, receptor(s) and/or co-receptor(s) to which Wnt5b binds to induce responses is still controversial/not well characterized. Since the current investigation is limited to only Ror2, it is better to be cautious in suggesting "a revision of the long-standing concept".

We have changed the text accordingly.

2.1b Decades of study have provided strong experimental evidence, showing that the long-distance interactions between the canonical receptors and ligands, expressed in discrete groups of cells, are required to induce spatiotemporal coordination of cells during morphogenesis. In contrast, non-canonical signaling has been shown to involve ubiquitously expressed cell junctional proteins/cell-recognition molecules, intracellular polarity complexes, and Ca²⁺ signaling, in addition to canonical ligands and receptors. How the spatial/temporal coordination/organization/activation of these non-canonical signaling components, which are expressed by all cells, is achieved is a mystery. I think this paper might contribute to this conceptual gap. So, instead of a claim for a paradigm-changing finding, the authors should clearly state and discuss that the paper investigates Wnt5b-induced non-canonical signaling induced via Wnt5b-Ror interactions and discovers one of the ways by which this non-canonical pathway can be organized in space.

We clarified this in the text and focused now on Wnt5b-mediated PCP signaling.

2.2 In the paper, cells that overexpressed Ror2 and/or Wnt5B could deliver the Ror-Wnt complex to Ror2-mutant cells and can activate Wnt/PCP signaling. This brings along a question about the WT expression and distribution patterns of Wnt5b and Ror2. The normal WT distribution patterns of Ror2 and Wnt5 relative to their sources could not be assessed from the text/results. Does the native Ror2 move from a producing region to its non-producing region in the WT embryo to shuttle Wnt5b? Do we see a defect in Wnt5b distribution profile in the absence of Ror2?

We investigated the transport of endogenous Wnt5b/Ror2 in PAC2 fibroblasts and in zebrafish embryos. We could show that the Ror2 signal can be found in Ror2^{-/-} cells co-cultured with wildtype

cells, suggesting that Ror2 protein is handed over to the KO cells (Fig. 1g). We further compared the spreading of Wnt5b and Ror2 to the spreading of mem-mCherry labeled membrane and show that the spreading of Wnt5b and Ror2 is significantly increased compared to the distribution of membrane fragments, suggesting a specific role for Wnt5b/Ror2 dissemination in embryonic tissue (Fig. 1f, Extended Fig. 1F). We further added quantification of Ror2 clusters in the PAC2 cells and in zebrafish embryos, indicating that 3-4 clusters of Wnt5b/Ror2 in the neighboring cells (Extended fig. 2a). Based on the suggestion from the reviewer, we performed an experiment investigating if Wnt5b spreading is dependent on the presence of Ror2 in the source cells. Therefore, we compared the dissemination of Wnt5b-GFP in zebrafish regarding different Ror2 expression levels in the source cells (Fig. 3j-l). To our delight, we find that overexpression of Ror2 in the Wnt5b source cells leads to an increase in the spreading from Wnt5b-GFP. Consistently, Wnt5b spreading is strongly reduced in the *ror1^{crispant}/MZ ror2^{t13}* mutant (Fig. 3j-l). Together with the data showing that Wnt5b/Ror2 increases the length of cytonemes (Fig. 3g-i), these data suggest that the Ror2 function is required for the spreading of Wnt5b on cytonemes.

2.3a Wnt5b-Ror2 interactions and delivery: Due to the technical challenges, the authors had to rely on the over-expression of Wnt5b and Ror2 in cells. Under these experimental conditions, tiny puncta constituted of Wnt5b-GFP and Ror2-Cherry complex are visible on source cytonemes and on the recipient cell membrane. In recipient cells, the puncta are at or near the contact sites of source cytonemes and recipient cells. Since there are only a few puncta in the recipient cell and are observed only to be at or near the contact sites of Wnt-Ror-delivering cytonemes, it is important to clarify what is meant by the signal delivery. To be specific, It is unclear from only a few images shown if the Wnt5b-Ror complex is endocytosed in the recipient cells or is simply displayed on the recipient membrane as remnants of the cytoneme member that had retracted from the contact sites.

We agree with reviewer 2 (and similarly with reviewer 1) that we could not conclude from the existing data that Wnt5b/Ror2 clusters are taken up by endocytosis. Therefore, we provide fresh data sets showing that the uptake of Ror2 in the receiving cells is Dynamin dependent. In two experiments, in PAC2 cells and in the zebrafish embryo, we reduce Dynamin function by Dynasore treatment and Dyn2^{K44A}, respectively (Fig. 1l, n). We found that the number of Ror2 clusters is significantly reduced, although the source cells form Ror2-positive cytonemes, which contact the receivers (Extended Fig. 2c). Furthermore, we use the Rab5-GFP TG line and show that the number of ror2-positive early endosomes in the receiving cells is significantly reduced if Dynamin function is perturbed (Extended Data Fig. 2e). This data set suggests that Ror2 can be taken up into the receiving cells by endocytosis into early endosomes.

2.3b From Fig 1 and movies, it is clear that the cytoneme membrane is also delivered together with Wnt5b and Ror2 to the recipient cells. If so, a possibility is that any membrane-tethered signal overexpressed in the same cytoneme-forming cell can simply be delivered randomly to the non-overexpressing cells during the cytoneme membrane exchange. Based on Fig.1a, mCherry-marked cytonemes show intimate adhesion to recipient cells and can exchange membranes via cytoneme

contact sites, irrespective of Wnt signaling. Do we observe similar membrane exchange in the WT scenario without Wnt/Ror overexpression?

We do understand the concern of this reviewer that membranes from filopodia can be spread. We, therefore, transfected mem-mCherry, Wnt5b-GFP, mCherry-Ror2, and Wnt5b-GFP/mCherry-Ror2 in PAC2 cells (Fig. 1f) and quantified the number of clusters in the adjacent cells (Extended Data Fig. 1f). We found that the number of Wnt5b-GFP, mCherry-Ror2, and Wnt5b-GFP/mCherry-Ror2 is significantly higher than mem-mCherry, suggesting that Wnt5b and Ror2 facilitate the transfer of cytoneme tip vesicles. In addition, we cannot exclude that also the delivered vesicles we observe, after mem-mCherry transfection, contained unlabeled, endogenous Wnt signaling components.

2.3c Do we need endocytosis of the Ror-Wnt complex to induce signaling? Any background information related to endocytosis/membrane display of the complex and correlation of this event with the induction of signaling in cells, if available, would be helpful.

To address this question, we performed two sets of experiments. First, we blocked endocytosis and analyzed the uptake of Wnt5b/Ror2 (Fig. 1n; Fig. 1l; Extended data Fig. 2c, e) and then analyzed the consequences on signaling (Extended Data Fig. 6r,s). In detail, we expressed Dyn2^{K44A} in zebrafish and used Dynasore in PAC2 cells and found that the uptake of Wnt5b/Ror2 is significantly reduced. We then introduced a clone of Wnt5b/Ror2 expressing as described in Fig. 4a. The data suggest that Dyn2-dependent endocytosis is dispensable for signal activation by transferred Wnt5b/Ror2 complexes. Based on these data sets, we conclude that endocytosis is required for the uptake of Wnt5b/Ror2 into the receiving cells; however, endocytosis is not required for signal activation.

2.4a About colocalization in Fig 1 d,e. Either Ror-mCherry or Wnt5b-GFP, when overexpressed in cells, mark the entire cytoneme membrane and can appear to be colocalized with any signal puncta that happen to be loaded on the cytonemes. Therefore, do we see similar source-membrane remnants/endocytosed puncta in WT cytoneme-recipient cells when Wnt5b-GFP/Ror-Cherry are not overexpressed?

To address this question, we repeated the experiment Fig. 1f. Here, we overexpressed a membrane marker, Wnt5b-GFP, mCh-Ror2, and Wnt5b-GFP and mCh-Ror2. We then obtained high-resolution images of the producing and receiving cells and overlaid these with the corresponding bright field images. This analysis indicates that mCh-Ror2 and Wnt5b-GFP are localized in puncta at the plasma membrane and on cytonemes, which is different from the membrane marker. Furthermore, quantification revealed that the transfer of Wnt5b, Ror2, and Wnt 5b/Ror2 into the receiving cells is significantly increased compared to the transfer of mCherry marked membrane, suggesting specificity in the transfer (Extended data Fig. 1f). In zebrafish, we find that Wnt5b, Ror2-mCherry, and fluorescent-

tagged membrane is handed over (Fig. 1m, single channels displayed in Extended Data Fig. 2d). In conclusion, we propose that the cytoneme-dependent spreading is specific to Wnt5b/Ror2.

2.4b The specificity of the signaling process is then defined by events that control the cytoneme contact sites. It is also important to verify if the native Ror-2/Wnt 5b (e.g. by using antibody co-staining) can be dispersed from producing cells to non-producing cells, showing a similar profile.

*To provide evidence that endogenous Ror2 protein can be handed over, we use an IHC approach in Pac2 cells and show that endogenous Ror2 is transmitted from WT PAC2 cells to the *ror2*^{-/-} receiving cells in PAC2 cells in cell culture (Fig 1g). Now, we provide a quantification of transfer of endogenous Ror2 protein (Extended Data Fig. 2a). Interestingly, we find that the amount of endogenous Ror2 clusters in the adjacent Pac2 cells is similar to the number of Wnt5b-GFP/Ror2-mCh clusters in the zebrafish embryo. Regarding signaling of the endogenous protein, we further show that endogenous Wnt5b/Ror2 in WT embryonic margin cells are able to induce JNK signaling in a halo in MZ *ror2*^{t13} mutant embryos, whereas MZ *ror2*^{t13} source cells cannot activate JNK signaling (Fig. 4p-s), indicating that endogenous Ror2 is transferred to the neighboring cells and subsequently, signaling can activate in the receiving cells.*

2.5a Fig 2&3: The FLIM-FRET analyses are indicative that the mem-localized Wnt and Ror can interact with each other. These results are clear. However, it is appropriate to mention in conclusion that these proteins are overexpressed in the same cell and they interact under the specific experimental conditions. Where and when within a cell native Wnt and Ror2 meet for actions are probably unknown.

We agree with the reviewer, and we have added this information to the text.

2.5b For the lifetime-FRET assay, the interaction between overexpressed Wnt-Ror shows similar values to that of the non-specific mem-GFP-memRFP. Therefore, a better control FLIM-FRET expt might be a mutant Wnt/Ror that does not interact or, alternatively, WT interacting partners that can not be sorted together in rafts/vesicles. Authors have utilized a mutant Ror-2 in experiments with FCCS in Fig 3, and results suggest an interaction between Ror-Cherry and Wnt5b-GFP, when overexpressed.

We provide better positive and negative control for the FLIM-FRET experiments. Therefore, we repeated the experiments with overexpressed memGFP and GFP-nanobody Vhh-mCherry. We find a

strong FLIM-FRET signal in the zebrafish embryo (Fig. 2a). As further suggested, we added a new set of experiments with a Ror2 construct lacking the Wnt interacting CRD domain. Overexpression of the Δ CRD-Ror2-mCherry together with Wnt5b-GFP shows a strongly reduced FLIM-FRET signal (Fig.2k). Furthermore, we have amended the text accordingly.

2.6a Line 190- Authors mentioned the "activation" of Wnt5b and Ror, but these experiments involved overexpression of these proteins in cells without any measurement of their downstream signaling activity. Does overexpression of these proteins in cells always induce signaling in cells?

We agree with the reviewer's suggestion and, thus, replaced the word "activation" with "expression". In addition, we backed up our observation with the analysis of the MZ $ror2^{t13}$ mutant and the $ror1^{crispant}/MZ\ ror2^{t13}$. For example, we added a data set showing that Wnt5b-Ror2 overexpression clones can activate JNK signaling (Fig. 4h-k), whereas the same clones fail to activate JNK signaling if they are induced in the $vangl2^{m209}$ mutant background (Fig. 4m-o). These results suggest that – although Wnt5b/Ror2 is overexpressed in the clonal cells – downstream signaling components such as Vangl2 are still essential to trigger the Wnt/PCP signaling cascade in the receiving cells. Finally, we show the potential of transplanted wild-type embryonic margin cells to activate JNK signaling in an MZ $ror2^{t13}$ mutant background (Fig. 4p-s).

2.6b Secondly, activation of Wnt/Ror was proposed to induce long cytonemes. A concern could be that the production of long cytonemes can simply be an effect of overexpression of these proteins, and removal of the basic cytoneme modulators (eg, IRSP53 or CDC42 function) would simply block cytoneme formation. My guess is that the molecular consequences of Wnt5b/Ror overexpression in cells are not fully understood. Therefore, in addition to the overexpression, Wnt5b or Ror2-knockdown results are required to strongly conclude that Wnt-Ror2 interaction is required for long cytonemes.

We agree with the reviewer on this point. Therefore, we quantified cytoneme emergence in MZ $ror2^{t13}$ mutants and in $Ror1^{Crispant}/MZ\ ror2^{t13}$ mutants (Fig. 3g-i). The new data set suggests that loss of $ror1/ror2$ function can lead to a reduced number of filopodia with a limited length. Interestingly, overexpression of Ror2 leads similarly to a reduction of filopodia number; however, it also leads to the formation of much longer filopodia (Fig. 3h,i). This is in accordance with our recently published data suggesting that a cross-regulation of Vangl2 and Ror2 is required to form the correct number and length of cytonemes (Mattes et al., 2018; Brunt et al., 2021).

2.7a Fig 4m-r. Does Wnt5a/b localize in Ror-2 mutant cytonemes?

To address this question, we have over-expressed Wnt5b-GFP in $ror1^{crispant}/MZ\ ror2^{t13}$ mutant (Fig. 3j-l). We find Wnt5b spreading is strongly reduced in the $ror1^{crispant}/MZ\ ror2^{t13}$ mutant (Fig. 3j-l). The number and length of cytonemes are reduced in the $ror1^{crispant}/MZ\ ror2^{t13}$ mutant, and we can find Wnt5b localization on cytonemes in rare exceptions (Fig. 3j).

2.7b Secondly, it is surprising to observe many long cytonemes from Ror-2 mutant cells (4m), connecting to neighboring cells. This is in sharp contrast to the Wt cells shown. The formation of so many long cytonemes in mutant cells is not consistent with the claim that the activation/expression/interaction of Ror and Wnt is required to produce cytonemes as a signaling response in the producing cells. The authors should clarify or revisit this data.

We have replaced the image with more representative ones (Extended Data Fig.6i). We further added data showing the strong reduction of cytonemes in the $ror1^{crispant}/MZ\ ror2^{t13}$ mutant (Fig. 3g-i; Fig. 4h).

2.8 line 230. Authors suggest that mutant Ror2 clones do not change the status of the JNK-reporter expression/distribution in the neighboring cells, unlike WT Ror2 clones. However, a concern is that the JNK reporter is introduced to the embryonic cells via microinjection, which might distribute variable numbers of constructs to cells. Clearly, there is variation in reporter fluorescence detection among the neighboring cells where there are no Wnt-clones introduced. Moreover, the interpretation of the JNK activation is difficult as the only thing clearly visible to me is the nuclear stain. It is hard to understand the cytoplasmic stain as the recipient cell outlines are unmarked, and there are so many cells crowded within the same place. Although there are variations in nuclear signal among cells, a corresponding change in the cytoplasmic signal cannot be assessed from these pictures. It is unclear how the change in signaling can be assessed from these experiments unless pre- and post-transplantation levels of the reporter were measured.

In this experiment, we overexpress the JNK reporter by mRNA injection. This procedure leads to a ubiquitous and even distribution of mRNA to all progenitor cells. However, we cannot exclude slight variations in the expression level. Therefore, we measure the ratio between cytoplasmic to nuclear localisation of KTR-mCherry to determine JNK signal activation. This KTR-mCherry reporter has been established as a valid procedure to measure PCP/JNK activation previously (Routledge et al., 2022; Brunt et al., 2021). We further added a better explanation about the functioning of the reporter (Fig. 4a and Extended data Fig. 6a-c). We further expanded our analysis and investigated the JNK reporter also in the $ror1^{crispant}/MZ\ ror2^{t13}$ mutant as well as in the $vangl2^{m298}$.

2.8b Fig 5 and 6. Although the change in cytoplasmic localization of the JNK reporter is unclear from the images, a nuclear-localized reporter is clearly removed in recipient cells under the specified conditions that removed cytonemes, indicating cytoneme-dependent paracrine function of Wnt5b/Ror2 complex. Similarly, suppression of canonical Wnt signaling by the Wnt5b/Ror-induced PCP pathway is very clear. These results are exciting.

We thank the reviewer for the encouraging words.

While the spatial distribution of *lef* expression is used to report the effect of non-canonical Wnt/Ror signaling on the canonical Wnt signaling, it might be more direct to visualize b-cat staining under the conditions tested.

*We respectfully disagree with the reviewer at this point. *Lef1* is a direct target gene of β -catenin. As discussed previously, an IHC for analyzing nuclear β -catenin in whole zebrafish embryos is extremely difficult. Therefore, we believe the alteration of *lef1* expression is sufficient to demonstrate Wnt/ β -catenin downregulation. To allow a better structure of the manuscript and based on the encouraging words from the reviewer, we expanded our analysis of the JNK reporter in Fig. 4 and Extended Fig. 6; and consequently, we moved all *Lef1* results to the accompanying Extended Data Fig. 7.*

In Fig 6, in situ Figure panels would be easier to interpret with the split color display.

*Unfortunately, this is not possible as in this experiment, the expression of *Lef1* is displayed in a purple, non-fluorescent color, and the clone in red. Therefore, the images were taken with a light microscope with a color camera.*

2.9 Since the paper claims cell-cell movement of the preassembled receptor-ligand complex based on only Ror2 and Wnt5b, the Introduction should indicate the requirement of various other receptors/co-receptors for canonical and non-canonical Wnt pathways.

We have added this information to the introduction and discussion.

Secondly, signal dispersion and signaling outcomes were known to be spatially coordinated, target-specific, and robust. It is unclear how the dispersion of the preassembled receptor-ligands might determine the specificity and spatial coordination during signaling and morphogenesis? A discussion point would be helpful.

We agree with the reviewer that signal dispersion needs to be spatially coordinated, target-specific, and robust. However, our current understanding of paracrine signaling is often based on randomly diffusible signaling molecules in the extracellular space. This stands in stark contrast to the need for

precise spatial coordination and target specificity. To our knowledge, direct cell-to-cell-based transport via cytonemes is the only mechanism allowing such a high degree of precision. However, the underlying mechanism directing the emergence of cytonemes is not fully understood. We speculate that the signal activation in the neighboring cells can be controlled by the direction, number and length of the protrusion. Furthermore, the competence status of the target cell can also influence the response. For example, we show that the target cell requires crucial downstream factors, such as Vangl2, to respond to the Wnt5b/Ror2 complex (Fig. 4m-o, Extended Data Fig. 6o-q). To clarify the knowledge gaps, we have further added a short paragraph to the discussion.

Few minor points:

Fig.1 Clear image of Wnt 5a/b on cytonemes. 1b. I suggest not distinguishing between filopodia and cytonemes at this point. The non-Wnt5a/b cytonemes could be for another signal or could be the precursor of the Wnt5a/b cytonemes prior to ligand loading. It is clear that the Wnt5 or Ror cytonemes are longer, and they connect between cells.

We agree with this point and removed the word cytonemes from 1b. In the following, we use the term 'cytoneme' only to describe 'signaling filopodia'.

2) Line 194: "To block cytoneme formation in zebrafish without interfering with Wnt signaling, we used the mutated protein IRSp534K and the dominant negative Cdc42T17N 14,17." It might be critical to mention non-interference with the canonical signaling here, especially since cytoneme induction appears to be caused by the non-canonical Wnt5b-PCP signaling but not by the canonical pathway.

We add "Wnt/beta-catenin" to the text. We further added a data set indicating that the expression of Wnt5b and Ror2 is unaltered by the co-expression of IRSp534K or Cdc42T17N (Extended Data Fig. 5c-f).

3) Line 258-260 (Fig 4i needs to be 5i). Citation of Fig 5G,L panels in the text needs to be verified.

We corrected the information for the figure.

Referee #3 (Remarks to the Author):

3.1 The authors need to validate the biological activity of the WNT-5b-GFP fusion protein thoroughly.

We agree with the reviewer that tagging of signaling proteins such as Wnt can alter their function regarding the activation of downstream signaling cascades. Therefore, we planned the experiments carefully and divided them into two groups: In the first one, we used Wnt5b-GFP to determine the subcellular localization (Fig. 1-3). These data are complemented by antibody staining against the endogenous protein. However, we would like to mention that – due to the limited number of good antibodies in zebrafish – we use an antibody against Wnt5a/Wnt5b. In the second group, we use non-tagged Wnt5b to determine its function in autocrine and paracrine signaling (Fig. 3-5). We added this information to the text and to all figures e.g., Wnt5b-GFP, Wnt5b, or α -Wnt5b.

3.2 Overexpression of ROR2 in HEK293 cells induced a “spiky” phenotype in our hands (unpublished). I wonder therefore what the hen and what the egg is in the observation of cytonemes as signal transducers. Is the proposed widely applicable or a special and potentially artificial consequence of ROR overexpression?

*We are grateful for these comments as the observation about the Wnt co-receptor Ror2 from the reviewer fits our described results – also in previous publications, e.g. Mattes et al., 2018. Furthermore, our data suggest that Ror2/Wnt5b can orchestrate their own transport route, namely the emergence of the cytonemes. We quantify the length and the number of filopodia in various Ror1 and Ror2 gain- and loss-of-function situations (Fig. 3g-i). We find that Ror2 (and Wnt5b) can induce long filopodia, whereas, in the *ror1^{crispant}/MZ ror2^{t13}* mutant, the number is significantly reduced. We find a comparable situation in controlling the formation of Hh cytonemes in *Drosophila*. In flies, the Hh co-receptor Interference hedgehog (*Ihog*), can activate long cytonemes, which are stabilized – a prerequisite for signaling (Bischoff et al., 2013; Gonzalez-Mendez et al., 2017). We added this information to the discussion.*

3.3 The WNT field generally refers to RORs as WNT “coreceptors” suggesting that they act in concert with FZDs. While, I am not necessarily convinced that RORs do not act independently of FZDs, I suggest to at least mention FZDs, which are the main WNT receptors also mediating PCP signals (especially in the introduction). This becomes even more obvious when convergent extension movements are introduced as a consequence of PCP signaling – a complex phenomenon that surely involves FZDs signaling and very likely WNT5B-FZD signaling.

*We agree with the reviewer. We do not exclude an Fzd-Wnt5b signal. Indeed, our results show that the single mutants for *ror1* and *ror2*, as well as the *MZror2^{t13}* have subtle phenotypes. In addition, ca. 50% of the *ror1^{crispant}/MZror2^{t13}* display a weak phenotype. This can be explained by redundancy or by additional receptors like the Fzds, which can partially compensate for the lack of Ror protein. Therefore, we have added this information to the introduction as well as to the discussion.*

3.4 Already in the abstract, the authors refer to “active WNT5B/ROR2 complexes”. How do the authors validate the complex activity justifying this statement?

We investigate the activity of the receptors in the producing cells as well as in the receiving cells. We show that Wnt5b/Ror2 signaling is important for the formation of long cytonemes in the producing cell. Furthermore, we show that the transferred Wnt5b/Ror2 complexes induce JNK signaling and repress Lef1 expression in the receiving cells. By using advanced, in-vivo imaging techniques such as in vivo FLIM-FRET and in vivo FCCS, we characterize the Wnt5b/Ror2 complex during transport, and the data suggest that after the formation of the complex in the producing cells, the complex maintains its integrity during transport. Therefore, we believe that the Wnt5b/Ror2 complex, after its assembly at the membrane of the producing cell, remains active, which would justify our statement in the abstract.

3.5 What is a “WNT cytoneme”? Please reword! The manuscript would benefit from a clear and early definition of cytoneme vs filopodium.

We changed the term “Wnt cytonemes” to “Wnt-bearing cytonemes” and added a definition of cytonemes and filopodia.

3.6 Please avoid the term WNT ligands. While WNTs are ligand of e.g. FZDs and RORs, the combined term WNT ligand is pharmacologically confusing.

We changed the term “Wnt ligands” to “Wnt proteins” or “ligands”.

3.7 The authors must present some kind of validation of antibody specificity for the main target antibodies WNT5A-B, rabbit PolyAb, ProteinTech, 55184-1-AP; & ROR2 629 (D3B6F). The vendor validation is not sufficient for long ranging statements such as the one put forward in this manuscript. For example the anti-ROR produces substantial nuclear immunoreactivity in Fig. 1C, which in my eyes seems unspecific but on the other hand, it is absent in the ROR KO cells. Please comment and explain.

We agree with the reviewer that the nuclear localization of the antibody against Ror2 is mysterious and require further examination. Therefore, we removed the reference to the nuclear staining and focus on the presence of Ror2 at the membrane (or its absence in the Ror2^{-/-} cells). We show the specificity of the Ror2 staining at the membrane (Extended Data Fig. 1c,d).

3.8 The authors call ROR2 the cognate receptor of WNT5B. This is an overstatement given that many of the FZDs also bind WNT5B.

We agree with the reviewer and removed the expression “cognate”. We further suggest a potential signaling complex of Wnt5b-Fzd without the presence of Ror2 in the discussion.

3.9. The colocalization data shown in Fig. 1D can by no means be interpreted as a confirmation of physical interaction between ROR2 and WNT5A. Colocalization is not interaction! In fact it remains elusive if the WNT is intra- or extracellular. It could be localized in a transporting vesicle, potentially in the same vesicle as ROR. The term WNT/ROR cluster is in my eyes an overstatement.

We agree with the reviewer that the co-localization shown in Fig. 1 is not direct proof of binding/interaction. Therefore, we provide a substantial amount of data in Fig. 2 showing that Wnt5b binds directly to Ror2 in the zebrafish embryo by using in vivo FCCS and FLIM-FRET. In detail, we provide data that the overexpressed Wnt5b-GFP can bind to mCh-Ror2. Consistently, we find no binding of Wnt5b-GFP with Ror2 lacking the CRD domain (Δ CRDRor2-mCh). We further carefully reworded the text and removed the word “cluster” where appropriate. Finally, to provide evidence of how Wnt5b/Ror2 is transported and handed over to the receiving cells, we added an experiment to Fig. 1. Here, we show that Wnt5b and the N-terminal part of Ror2 is extracellular and, thus, accessible for a secreted GFP nanobody (secVhh-mCh). In addition, co-localization of secVHH-mCherry with Ror2-GFP was not observed when the GFP tag was at the C-terminus of Ror2 (Fig. 1h-l). We further added a super-resolution image showing Wnt5b-GFP/secVHH-mCherry on cytonemes (Extended Fig. 2b). Finally, we block the uptake of Wnt5b/Ror2 in the receiving cells by Dyn2(K44A) and Dynasore (Fig. 1l, n). These data sets and their accompanying quantifications suggest how Wnt5b/Ror2 is taken up into the receiving cells (Fig. 1k).

3.10 The central observations in Fig. 1 e, g, f, h are not accompanied by any form of quantification, which renders them circumstantial. It remains unclear how many independent experiments were performed in Fig. 1. E g n=17 (cells) do not represent independent observations that would justify statistics or satisfy general rules of reporting.

We agree with the reviewer and added all quantifications to Extended Data Fig. 1 and 2, and explain these in the text and the figure legends.

3.11 The settings for confocal imaging must be reported. Are the optical sections for the different channels equally thick? If not, co-localization or lack thereof does not allow too many conclusions.

We use in all images sequential scanning to avoid cross-talk between the different channels. All optical sections for all channels are equally thick to allow a comparison of the localization. Furthermore, all technical details have been added to the Material and Methods section.

3.12 The term “Wnt/PCP cytonemes” is an unfortunate construction to name a cell organelle.

We removed this term and replaced it with Wnt5b/Ror2-bearing cytonemes.

3.13 The ligand binding domain of ROR2 is extracellular. The authors need to address (experimentally) where the receptor-ligand complex is formed and how it is transported. I have difficulties to envision that the receptor-ligand complex is formed in the ER/Golgi and that ligand binding would be maintained when the receptor is exposed to the cell membrane with the ligand on the outside of the cell (infinite dilution of the ligand should/could lead to its dissociation). The other alternative is that WNT binds ROR on the cell surface and that the ligand receptor-complex is transported along the cytoneme. In this case, however, the ROR/WNT-positive dots in the cytoneme are difficult to explain. These remind more of transport vesicles. These aspects pose a major shortcoming of the study and require clarification.

We agree with the reviewer and performed several sets of experiments to experimentally determine the orientation of Wnt5b/Ror2 complexes during transport. First, we revisited our in vivo, FCCS and FLIM-FRET data. These data suggest that a FLIM-FRET signal and cross-correlation of Wnt5b-GFP/mCh-Ror2 is detectable at the plasma membrane of the producing cell.

Next, we addressed the orientation of the ligand-receptor complex during cytoneme-mediated trafficking. Therefore, we expressed Wnt5b-GFP, GFP-Ror2, and Ror2-GFP in PAC2 fibroblasts and co-cultured these with cells stably expressing a secreted nanobody (secVhh) coupled to mCherry (Fig. 1h-j). After 24h of co-culture, we find Wnt5b-GFP signal co-localizing with secVhh-mCherry on cytonemes, similar to GFP-Ror2. However, Ror2-GFP does not co-localize with secVhh (Fig. 1j), suggesting that the ligand and the N-terminal part of the receptor face the extracellular side, which is accessible for secVhh.

Finally, we further provide evidence that the uptake of the Wnt5b/Ror2 ligand complex is Dynamin dependent in PAC2 fibroblasts and in the zebrafish embryo (Fig. 1l, n).

Finally, we conclude that Wnt5b/Ror2 complexes can be loaded on cytonemes and are transported along these protrusions towards the tip (Extended Data Fig. 2b). After contact formation of the Wnt/PCP cytoneme, the Wnt5b/Ror2 positive cytoneme tip is handed over to the neighboring cell and endocytosed in a Dynamin-dependent way (Fig. 1k).

3.14. Fig.2 c-f: The WNT5B construct is most likely a GFP fusion, that should be indicated in the figure everywhere (now it appears that the legends state WNT5B and WNT5B-GFP). Abbreviations in Fig. 2g,

f, I are not consistent. Is mGFP the same as memGFP? Or is mGFP monomeric GFP? The nomenclature/abbreviations must match between figure and graph.

We have changed the description in the text and in the figures accordingly.

It is unclear whether the data originate from one or several independent experiments. Different cells are not independent experiments, rather technical replicates. The authors need to confirm that statistics are suitable when the groups contain different independent observations. inhibitor.

We added the requested information to the figure legends.

3.15 For the FLIM FRET approach the authors report GFP lifetime, FRET efficiency and donor-acceptor distance. First of all the authors should explain to the general audience what these values mean in terms of protein-protein proximity and orientation.

We expanded the text and the Extended Data Fig. 3 to better describe the reasoning and the procedure of this approach.

Second, I understand that the distance calculations are based on the Förster distances of the eGFP-mCherry FRET pair (54.2 Å). However, this is valid for the fluorescent proteins, not for fusion proteins fused to these fluorescent proteins. Thus, the donor-receptor distance calculations are at best estimates.

We agree with the reviewer that the distance measured is the distance between the fluorophores tagged to Wnt5b and Ror2. Therefore, we amended the text and expanded on the information in (Extended Data Fig. 3).

Furthermore, FRET efficiency, which the distance calculations are based on, is dependent on both distance of the FRET pair and the orientation of the two resonating fluorophores. This orientation is surely different, when investigating GFP vs mCherry or ROR-mCherry vs WNT-GFP.

In this experiment, we compare the differences of the interaction of Wnt5b-GFP and mCh-Ror2 on the plasma membrane of the producing cell, on cytonemes, and in the receiving cells. We agree with the reviewer that the tagging and the distance influence the FRET efficiency. Therefore, we use the same

constructs to explore relative differences between Wnt5b and Ror2 during transport. Surprisingly, we do not find a significant difference, suggesting that the Wnt5b/Ror2 complex remains stable. We changed the positive control to mem-GFP + Vhh-mCherry to make it comparable to the FCCS approach.

3.16a The experiments shown in Figure 2 are poorly controlled. This statement is based on the fact that the maximum FLIM FRET between memGFP and mem-mCh is assessed between proteins that are not showing any specific interaction. The resonance energy transfer appears between randomly colliding fluorescent proteins in a totally overcrowded membrane. This random interaction by no means sets the maximum value for specific interaction. One could for example create an artificial construct containing both eGFP and mCherry for optimal energy transfer.

We have changed the positive control and now use a mem-bound form of GFP and an anti-GFP Vhh nanobody tagged with mCherry for both the in vivo FLIM-FRET experiments as well as in the in vivo FCCS experiments. We further added a negative control with a Ror2 construct lacking the Wnt binding domain CRD (Δ CRD-Ror2-mCherry). These data sets are now displayed in Fig. 2.

Similarly, the negative control of memGFP and cyto-mCh assesses random collision of proteins expressed in two different compartments. Here again proteins that are not known to interact fused to eGFP and mCherry should be used instead.

We respectfully disagree with the reviewer memGFP and cyto-mCherry are localized in two different compartments leading to a minimal FRET signal and no cross-correlation (Fig. 2). Therefore, we believe that this experimental setting can serve as a negative control. However, we added a further negative control by analyzing FLIM-FRET and cross-correlation of Wnt5b-GFP with a Ror2 construct lacking the Wnt binding domain CRD (Δ CRD-Ror2-mCherry). These data sets are shown in Fig. 2.

3.16b In relation to the design of these control experiments, the authors need to match the system under investigation. This is a single transmembrane protein (ROR) and a secreted lipoglycoprotein potentially bound to a carrier such as afamin (WNT).

In this context, I would also like to get back to one of my previous points above, emphasizing the location of the receptor and the ligand in the cell. This needs to be reflected in the control experiments. Is the ligand-receptor complex embedded in the plasma/cytosol membrane or is the complex transported along the cytoskeleton in transport vesicles.

We agree with this suggestion and therefore, we present new data suggesting that the Ror2 receptor is embedded into the cytosol membrane (Fig. 1h-j). Furthermore, we added an experiment using Wnt5b-GFP and a Ror2-mCherry construct which is tagged at the C-terminus. We observe no signal as the plasma membrane act as a FRET insulator and does not allow FRET (Extended Data Figure 3m).

3.17 The Figure 3a presents a very nice positive control in my eyes, where the membrane-embedded receptor is bound by a fluorescently-tagged Vhh construct. Splendid! The control in Fig. 3b, however, suffers from the same shortcomings as mentioned above (reflected by the absence of cross correlation). Furthermore, the control with the Delta-CRD-ROR construct shown in Fig. 3f is highly relevant. In that particular micrograph, however, the WNT-GFP fluorescence is hardly visible. This leads back to my question about the number of independent experiments that have been performed for every observation in the paper. This information must be provided in the figure legends.

We thank the reviewer for the encouraging statement, and we have added the requested experiments and information to Fig. 2 and the accompanying Extended Data Fig. 3

3.18. Figure 3/FCCS: The number of independent experiments is not reported. The authors must justify and explain, why the ROI for FCCS is placed across the membrane vs on top or the bottom of the cell. How large is the ROI/effective volume for FCCS? Is the ROI ($0.65 \cdot 10^{-9} \text{ nm}^3$) relevant in relation to the diameter of a cytoneme?

We added the information to the figure legend. For auto-correlation, the effective volume for GFP and mCherry channels are calibrated with low concentrations of dyes, which are diluted in water. For FCCS measurements, all the ROIs are equal in each channel ($V_{ef} \text{ GFP } 0.56 \text{ fl}$; $V_{ef} \text{ mCherry } 0.75 \text{ fl}$). The cross-correlation effective volume ($0.65 \cdot 10^{-9} \text{ nm}^3$) is calibrated based on the Schwille formula. Furthermore, the volume analyzed is extremely small, and therefore, the ROI can be placed on a membrane of a cell or of a cytoneme.

3.19 The membrane compartment recorded in a cytoneme vs the plasma membrane is completely different. I fear that the diffusion behavior in the restricted cytoneme area vs large plasma membrane are different and make the FCCS values difficult to compare.

It is unclear why the reviewer assumes that the cytoneme membrane is “completely” different to the plasma membrane. There are few studies elucidating the composition of the filopodia membrane versus the plasma membrane, and the reports conclude that the filopodia membrane is less fluid compared to other regions of the plasma membrane (Gaus et al., 2003; PNAS). In line with the fact that membrane fluidity decreases when Cholesterol is depleted, the enrichment of Cholesterol in filopodia was indirectly demonstrated (Chierico et al., 2014, Sci Rep). However, inconsistent with these observations, a number of papers reported that Cholesterol depletion activates Cdc42 and filopodia formation (Xie et al., 2011, JCB; Brachet et al., 2015, JCB). Similarly, we observe a similar diffusion behavior (D2) for Wnt5b-GFP and Ror2-mCh at the plasma membrane compared to the cytoneme membrane.

3.20a The calculation of a K_d value from FCCS measurements must be clarified. Especially the “diffusion with triplet”! and “pure diffusion” models. What does “The dissociation constant (K_d) was analysed based on these fitting values for every measurement.” mean? I would like to see the raw data for the K_d calculation in the supplement/extended data.

For absolute K_D measurement, the system for each channel's effective volume (in our assay: red channel and green channel) needed to be calibrated. For the calibration assay, we used Atto 488 and Atto 565 (Extended Data Fig. 3j). These two dyes are rhodamine-based fluorophores. In the FCCS measurements in the embryo, tagged GFP and mCherry fluorophores were used. The rhodamine dyes and the tagged proteins are all in a triplet state. So, for the auto-correlation, the “diffusion with triplet” model was used. However, for cross-correlation measurements, the V_{cc} is determined by specific equations set in the LAS-X software from Leica, which had excluded the triplet already. In this case, the “pure diffusion” is the proper fitting model for cross-correlation. We added this information to the text.

3.20b Are binding curves hyperbolic or linear? The authors must be clear that FCCS in 3 dimensional diffusion can be applied to the apparent 2 dimensional diffusion in membrane protein complexes. For inspiration, I suggest to look at <https://journals.asm.org/doi/10.1128/MCB.00087-14>. References for the analysis of receptor-ligand K_ds in 2 D membrane diffusion by FCCS should be given. How do the K_d values match those previously reported for WNT-ROR interactions (e.g. recent work from the Lemmon lab?).

We are grateful for this comment from the reviewer, and we have added this information to the discussion. Unfortunately, in the study in Drosophila from the Lemmon lab only the K_d for Drl (the homolog of Ryk) and DWnt5 were measured. In this study, surface plasmon resonance studies were used to determine the binding affinity of Drl family to DWnt-5 immobilized on a sensor chip. The authors report that the mean K_D for binding of the extracellular region of Drl to immobilized DWnt-5 across multiple repeats was 720 ± 160 nM. The authors conclude that the Drosophila RYK/Drl family receptors bind DWnt-5 with affinities typical for RTK ligand binding. Interestingly, we found that the zebrafish RTK Ror2 binds to ZFWnt5b with a similar affinity, namely around 440 ± 80nM.

3.21. Residence time: The presented concept requires a long residence time of the ligand at the receptor. While binding dynamics of WNTs to CRDs of membrane receptors has not been systematically addressed there are indications in the literature that the residence time is long. However, given the low K_ds presented in this study (300-500 nM, which corresponds to round about 16 µg/ml WNT!!! This is a concentration that is not reachable in cell culture with addition of recombinant WNTs!), it appears unlikely that the residence time is particularly long. In fact, residence time is in part defining affinity since it substantially affects the ligand off rate. The calculated K_d values in the range of hundreds of nM of WNT will NEVER be reached in vivo and thus do most likely not reflect a biologically meaningful affinity. As an example, binding affinities of WNTs to FZDs in receptor

overexpression systems have been calculated to be in the range of low nM concentrations, more like 2-5 nM employing different methods.

We respectfully disagree with the reviewer on this point. Most of the measurements of KDs between ligands and receptors in general, and Wnts and their (co-)receptors specifically, have been performed in vitro. This includes several in vitro methods with purified receptor binding domains and ligands or HEK293 cell culture-based measurements. We would argue that the outcomes of these studies are questionable as it is unclear how binding affinities differ in a real in vivo setting compared to these in vitro experiments. Therefore, we measure the binding affinities of Ror2-mCh and Wnt5b-GFP in the natural environment, namely in the living zebrafish embryo. We believe this is an important prerequisite for measuring the correct KDs.

Furthermore, the comparison of the concentrations across in vitro and in vivo experiments is also not valid. Obviously, the concentrations of ligands and receptors in a cytoneme tip can be much higher than in a cell culture medium. This raises the question if these in vitro experiments with “soluble Wnt protein” can be meaningful. Cytonemes have the unique possibility to achieve high concentrations of ligands and receptors in a small area, such as the cytoneme tip. This extremely high focal concentration could allow receptor clustering at the contact sites and signal activation in the receiving cells. In comparison, a similar number of molecules in the medium of a cell culture dish would not be able to elicit a response. The statement “will NEVER be reached in vivo” is, therefore, factually wrong, even if it is written in capital letters.

3.22 Lane 133 – the authors claim that their mCherry-ROR2 is biologically active, referring to Ext Fig 4c. How was biological activity of the construct validated?

We agree with the reviewer that tagging of signaling proteins such as Wnt can alter their function regarding the activation of downstream signaling cascades. Therefore, we planned the experiments carefully and divided them into two groups: In the first one, we used Wnt5b-GFP to determine the subcellular localization (Fig. 1-3). These data are complemented by antibody staining against the endogenous protein. However, we would like to mention that – due to the limited number of good antibodies in zebrafish – we use an antibody against Wnt5a/Wnt5b. In the second group, we use non-tagged Wnt5b to determine its function in autocrine and paracrine signaling (Fig. 3-5). We added this information to the text and to all figures e.g. Wnt5b-GFP, Wnt5b, or α -Wnt5b.

However, in addition, we checked for the function of the tagged constructs. For example, the mCherry-Ror2 construct has been expressed in clones in an embryo expressing the PCP/JNK reporter. We show that the tagged Ror2 construct can induce JNK signaling in a paracrine way similar to the non-tagged Ror2 construct used in a similar approach (Extended Fig. 6g).

3.23 Lane 145-147: The authors argue that their FLIM-FRET data argues for ROR-WNT5B interaction.

Given the criticism of controls, I do not see that the experiments support the claim. A comparison of FLIM FRET with the Vhh construct (max FRET) and the Delta-CRD-ROR2 (min FRET) should be done similar to what has been done for the FCCS set up.

We added the requested experiments as described in 3.16 and 3.17.

3.24 Fig. 4: In the quantification of filopodia length, it appears that ROR is active but that addition of WNT5B does not substantially elevate the effect. When it comes to filopodia number WNT5B in combination with ROR inhibits the effect. This discrepancy is not reflected in the text of the result section.

We added this point to the text

3.25 The transfected constructs in Fig. 4 are untagged? How is it controlled that cells express both ROR and WNT? Or are the constructs tagged with the same fluorescent proteins as in Fig. 3? If so that needs to be indicated in the figures and legends.

In Fig. 4 the constructs are untagged. Microinjection of mRNA is a reliable method to overexpress genes in zebrafish embryos. To identify the cells overexpressing the indicated constructs, we mix the mRNAs with the mRNA for memGFP.

3.26 SP600125 is a JNK inhibitor, not a JNK signaling.

We amended the text accordingly.

According to <https://www.pnas.org/doi/full/10.1073/pnas.251194298>, the JNK inhibitor has a K_i value of 190 nM with a 20 fold selectivity over other kinases. Tocris provides values of IC_{50} =40-90 nM. However, the authors use the inhibitor at a concentration of more than 200 fold the K_i (40 μ M), which basically renders it unspecific or at least non selective. I would argue that it is not surprising that delicate structure such as cytonemes collapse at this concentration and that has relatively little to do with WNT-induced and ROR-mediated signaling.

The usage of inhibitors in vivo (zebrafish embryo) is often different from the usage of these compounds in vitro (tissue culture) – specifically in a biochemical assay for JNK signaling the reviewer seems to refer to. The solubility of SP600125 is extremely low and requires the addition of DMSO, as stated in

the original paper. The reduced solubility also impairs penetration into the zebrafish embryo. Therefore, a concentration of 40 μ M of SP600125 in the medium can be justified.

Interestingly, in the same publication the reviewer refers to, the authors state that “At a concentration of 50 μ M, SP600125 did not block the phosphorylation of ERK1 or -2 and did not inhibit the degradation of I κ B α .” Therefore, even in cell culture, the inhibitor seems to be specific at this concentration. Finally, we have described the functioning of the inhibitor in zebrafish cells recently (Brunt et al., 2021; Nat Comms), and also here, we used a concentration of 40 μ M.

Also, the authors should check the size bar in Fig. 4k. The cells in the DMSO control look much larger than in the JNK inhibitor-treated conditions. This could either be a different magnification or inhibitor-induced cell shrinkage. concentration response curve could assist to distinguish between target specific and unspecific effects.

We revisited the data and adjusted the magnification in (Extended data Figure 5a).

3.27. I could not find the reference Brunt et al 2021 in the reference list. Is that a reference to the JNK activity reporter? If not, a reference for the reporter should be included.

We included the missing references.

3.28. Fig. 4: How do the authors relate te observed effects on the JNK reporter with the transplantation paradigm to ROR and WNT signaling. There is no experimental aspect (blockade of ROR) that argues for an involvement of WNT/ROR signaling in the activation of JNK in surrounding cells. This argument is based on pure and relatively weak correlation, so far.

*To strengthen our argument that the transferred Wnt5b/Ror2 complex is activating PCP/JNK signaling in the neighboring cells, we added several new experiments. For example, we induced a Wnt5b/Ror2 clone in the *ror1^{crispant}/MZ ror2^{t13}* mutant background and in embryos expressing a dominant negative Ror2 construct (*Ror2³ⁱ*). We found JNK signaling activation in the neighboring cells, suggesting that the Wnt5b/Ror2 complexes observed in Fig. 1-3 can activate paracrine PCP signaling. To exclude that Wnt5b can activate PCP signaling without Ror2, we generated a Wnt5b positive clone in the same background. We find a strongly reduced capability of these clones to activate JNK/PCP in the neighboring cells. Similarly, these clones are able to repress *lef1* expression in the zebrafish embryo at a distance. Finally, we blocked cytoneme formation in these clones, and we observed that cytoneme-reduced Wnt5b/Ror2 clones are not able to activate paracrine JNK signaling. Therefore, we believe that Wnt5b/Ror2 complexes – transported by cytonemes – can activate Wnt/PCP in the neighboring cells.*

3.29. Figure 5: Are the images for Fig. c, d, g, o taken with identical settings? Background seems higher, but this could obviously be due to a wide-spread cytosolic expression of the reporter.

In all images of one experiment, we use the same microscopy settings. This increase in cytosolic fluorescence depends on the activation of the KTR reporter.

How is the ratio determined in this case, when the nucleus is difficult to identify?

To illustrate the measurement, we have expanded the Extended Data Fig. 6.

3.30 The scale bar in Fig. 5 is 20 μm . It surprises me a bit that the whole field of view (about 80 μm) is equally and evenly affected with regard to the JNK reporter. If cytonemal signaling (cytonemes are not visible despite the memGFP in the transplanted cells) would be responsible for the effect, I would expect a gradient effect or cells that are in contact with memGFP-positive cytonemes showing high JNK signaling. In this scenario it appears more likely that a released and diffusible factor X is mediating the effect over long relatively distance.

This publication and our former publications demonstrate that cytonemes can reach a length of 100 μm (Mattes et al., 2018; Brunt et al., 2021, Routledge et al., 2022). This is supported by the observation of Shh cytonemes in the chick limb, which can be as long as 200 μm (Sanders et al., 2013).

Regarding a possible gradient, we provide data that a very low expression of Wnt5b/Ror2 in the clone can lead to paracrine PCP activation in the four cell rows around a clone, but activation in the fifth cell row is hardly detectable (Fig. 4d).

To exclude a diffusible factor X, we inhibited the formation of Wnt5b/Ror2 cytonemes by Cdc42 T17N and IRSp53 4K and observe a reduced paracrine Wnt/PCP signaling in vivo. The spreading of a diffusible factor could not be inhibited by these measures. Therefore, we believe that this is strong evidence for paracrine dissemination mechanisms using cytonemes in the zebrafish gastrula.

In lane 249-253, the authors argue that ROR expression and ROR/WNT expression show differences over distance referring to the graphs Fig. 5e, f. The p values presented refer most likely to comparisons to baseline, not between the distances. By eye, the effects over distance in these graphs look very similar. Thus, I do not feel that this argument holds.

We revisited the data, re-calculated the p-values, and specified the statistical test and the n numbers in the figure legends.

3.31 Lane 263-262: Functional CDC42 is most likely relevant for the formation of any cytoneme, not only those that are assessed in the context of WNT signaling

We agree and we will amend the text.

3.32 The remaining figures do not – in my eyes – address the interesting concept of ligand-receptor complex transfer, which in fact is the provocative and exciting aspect of this work. I would rather suggest the authors to focus on that aspect and to find a biochemical or biophysical way to detect complex transfer that investigating signaling crosstalk in vivo that deviates from the central message.

We do not agree with the view of the reviewer. The transfer of such a ligand-receptor complex needs to be investigated in its natural environment. Therefore, we use state-of-the-art imaging techniques and biophysical experiments such as FLIM-FRET or FCCS in the living zebrafish embryo. An artificial biochemical approach would have severe limitations and would not allow addressing the consequences of such an important and unexpected dissemination route.

Referee #4 (Remarks to the Author):

My overarching concern is the Wntb/Ror2 signaling complexes proposed to be loaded on cytonemes and transported on them to a receiving cell are not physiological, as the key colocalization, FLIM-FRET and FCCS experiments are performed using overexpression of the component proteins.

*We agree with the reviewer that visualization of the dynamic of this transport mechanism requires the overexpression of tagged constructs. However, the overexpression is in a physiological range. For example, FCCS experiments are performed with a very low concentration of fluorescently tagged proteins (expression of mRNA with a concentration of 50ng), whereas the FLIM-FRET experiments were done with a higher mRNA concentration (ca. 200ng mRNA). 50-200ng of mRNA of Ror2 was sufficient in rescuing the *ror1^{crispant}* / MZ *ror2^{t13}* mutant at 24 and 48hpf (Fig. 3, Extended Data Fig. 4). Furthermore, the localization data are backed-up with antibody staining of the endogenous protein (Fig. 1). Furthermore, we complemented the paracrine signaling experiment with the transplantation of WT cells from the embryonic margin into a MZ *ror2^{t13}* mutant background to show that these cells can activate paracrine JNK signaling*

Second, the role of Ror2 is studied using dominant negative form of the receptor and transplanted mutant clones, while the relevant phenotypes of the zebrafish mutants are not analyzed/reported.

We agree with the reviewer and now include a detailed analysis of the $ror2^{t13}$ mutant, the MZ $ror2^{t13}$ mutant, the $Ror1^{Crispant}$, and the $ror1^{Crispant} / MZ\ ror2^{t13}$ mutant. These data support the idea that $ror1$ and $ror2$ are important co-receptors in the Wnt/PCP signaling pathway. Furthermore, analysis with the double mutant suggests that Wnt5b/Ror2 complexes can be handed over to the receiving cell to activate Wnt/PCP signaling in a paracrine way.

4.1 Figure 1 shows colocalization of Wnt5b/Ror2 puncta. The number of cells and cytonemes analyzed should be provided on panel “b”. The co-localization of a-Wnt5a/b and Ror2 shown in figure 1d is not compelling without seeing individual color channels and needs to be quantified.

We added a new high-resolution data set on suggesting that Wnt5b and Ror2 are co-localizing on cytonemes and in the receiving cell.

Moreover, one of the components is overexpressed. Additional evidence of co-localization at endogenous protein levels is needed such as proximity ligation assay.

We screened several antibodies for Ror2 and Wnt5b in the zebrafish gastrula. Unfortunately, the ones used in this publication are polyclonal anti-rabbit antibodies and cannot be used for a co-staining experiment. However, in Fig. 2, we provide evidence of direct interaction and binding of Wnt5b-GFP and Ror2-mCherry in the living zebrafish embryo by using FLIM-FRET and FCCS.

4.2 The detection of Wnt5a/b needs to be performed on relevant cells and tissues. In Fig. 1 fibroblasts and epiblast cells are used, while the authors wonder “how these lipid-modified ligands are precisely disseminated in embryonic tissue to regulate complex tissue movements like C&E” . Much work is being done on epiblast cells at 8 hpf, but in which embryonic position? How does this position relate to the normal morphogenetic processes?

The imaging in Fig. 1 m and n was performed at embryonic margin cells, which express Ror2 and Wnt5b. Notably, the PAC2 fibroblast line is a cell line generated from a 48h-old embryo (Köster).

4.3 Detection of Wnt5/Ror2 puncta in $ror2$ KO zebrafish fibroblasts. A more complete characterization of the $ror2$ mutant cell line should be performed including a western blot.

We performed an IHC analysis to characterize the Ror2 mutant cell line (Extended Data Fig. 1). In comparison to the control cells, we found no detectable signal for Ror2 in the KO cells. Furthermore, overexpression of a tagged Ror2 construct shows nearly perfect co-localization with the antibody. To show the specificity of the antibody, we have added a detailed quantification of the Ror2 puncta in the Pac2 ror2^{-/-} cell line (Extended Data Fig. 1).

4.3a Only eriments shouldone mutant cell is shown with α -Ror2 Ab positive puncta. Quantifications from larger number of cells from more than one exp be provided.

We provide a quantification of the observation of Ror2 hand-over from WT PAC2 cells to Ror2^{-/-} PAC2 cells (Extended Data Fig. 2)

4.3b. Additional controls should be provided:

i. α -Ror2 Ab staining of ror2 mutant cells not co-cultured with WT cells.

We performed AB staining for Ror2 and Wnt5a/b in WT PAC2 cells and in to Ror2^{-/-} PAC2 cells and Wnt5a/b^{crispant} PAC2 cells (Extended Data Fig. 1)

ii. α -Ror2 Ab staining of ror2 mutant cells cultured in conditioned media from WT cells, to ensure that Ror2 or Ror2/Wnt5 vesicles are not transferred as secreted vesicles rather than “handed over” to adjacent cells.

This is an interesting suggestion. However, it is technically challenging to capture the transfer of Ror2 from WT PAC2 cells to the mutant cells. We can clearly identify events when Ror2 has been handed over and in Fig. 1 we were very lucky to visualize filopodia at the same time. A negative result would not allow us to conclude that Ror2 can be handed over on extracellular vesicles.

4.4 The conclusion about the Wnt/PCP filopodia is premature. In line 189 the authors write “... we overexpressed the indicated PCP constructs (i.e. Wnt5b and Ror2) in clones in the zebrafish embryo and quantified the length and the number of filopodia per cell”. At best, these data support the conclusion “Wnt5b and Ror2 can be loaded on signalling filopodia”, but the following conclusions “referring to them as Wnt/PCP cytonemes” and “the Wnt5b/Ror2 positive cytoneme tip is delivered to the neighbouring cell during zebrafish gastrulation” cannot be inferred from the current data.

We rewrote the conclusion more carefully.

4.5 In subsequent lines, the authors state “We found that activation of Wnt5b, Ror2, and Wnt5b/Ror2 led to the formation of fewer but much longer filopodia” and later (line 193) “We asked if the Wnt/PCP-induced filopodia are dependent...”. And Lines 201-202 “Therefore, we conclude that Wnt/PCP is required to induce long cytonemes in the source cell.” Here the authors move from Wnt5b and Ror2 overexpression to “Wnt-PCP pathway activation without analyzing any Wnt-PCP pathway activity readouts. Later JNK activation is used as the only Wnt/PCP pathway activity readout, but the key hallmarks of Wnt/PCP signaling during gastrulation: cell shape, cell body orientation are not analyzed here.

We removed the term “Wnt/PCP” and replaced it by “Wnt5b/Ror2 expression”. We further analyze the cell shape in Fig. 5 and show that over-expression of Wnt5b/Ror2 can be partially rescued by reducing cytoneme formation.

4.6 The specificity of Wnt5b as a Wnt/PCP ligand is not clear. Other Wnt ligands should serve as controls. First, Wnt8 ligand that can also bind to Ror2 but stimulate canonical Wnt signaling should be used and tested for the ability to activate JNK.

We have added a new data set analyzing Wnt8a clones and find that they are unable to activate the JNK signaling reporter (Extended Data Fig. 6t,u)

4.7 To further probe the conclusion that “Wnt/PCP is required to induce long cytonemes in the source cell” and considering the earlier studies of the authors that Vangl2 (the conserved PCP pathway component from Drosophila to zebrafish to mammals) promotes the formation of long cytonemes to enable Wnt/b-catenin signaling (Nat. Comm., 2021), the experiments with Wnt5b/Ror2 cytoneme formation should be performed in vangl2 zebrafish mutants lacking both maternal and zygotic Vangl2 expression.

We performed the analysis of the MZ $ror2^{t13}$ mutant and the $ror1^{crispant}/MZ\ ror2^{t13}$. Specifically, we added a data set showing that Wnt5b-Ror2 overexpression clones can activate JNK signaling (Fig. 4h-k), whereas the same clones fail to activate JNK signaling if they are induced in the $Vangl2^{m209}$ mutant background (Fig. 4m-o). These results suggest that – although Wnt5b/Ror2 is overexpressed in the clonal cells – downstream signaling components such as Vangl2 are still essential to trigger the Wnt/PCP signaling cascade in the receiving cells.

4.8 Another key thesis of this manuscript is that Ror2 signaling mediated by the cytoneme-transferred Wnt5b/Ror2 complexes activate paracrine JNK signaling. This is tested by employing a fluorescent JNK kinase translocation reporter, JNK-KTR-mCherry that relocates from the nucleus to cytoplasm upon JNK activation and a newly generated *ror2* zebrafish mutant line. The observation that *ror2* mutant

cells transplanted into WT embryos expressing JNK-KTR-mCherry reporter do not alter the reporter intracellular distribution in the neighboring cells whereas WT cells induced JNK-KTR-mCherry activation when transplanted into WT host, is interpreted that that endogenous Ror2 signalling in the grafted cells is required for paracrine JNK activation via the cytoneme-mediated transfer. Similar experiments with transplanted Wnt5b expressing clones in which cytoneme formation is experimentally impaired using the dominant-negative mutants IRSp534K and Cdc42T17N leads the authors to conclude that paracrine Wnt/PCP signalling is strongly dependent on cytoneme appearance in zebrafish development. There are several concerns with these experiments and their interpretation. The *ror2* mutant line harboring a premature stop codon (truncated Ror2 protein) used in these experiments is not phenotypically characterized. What is the phenotype of *ror2*-deficient mutant embryos with respect gastrulation, in particular, convergence and extension phenotypes ?

a. Morphometric analyses should be performed to monitor anteroposterior axis extension and mediolateral narrowing of embryonic tissues.

b. The hallmarks of Wnt/PCP signaling during zebrafish gastrulation, i.e. mediolateral cell elongation and asymmetric Vangl2/Pk localization should be assessed in these mutants.

*We have extensively characterized the phenotype of the Ror2 mutant line (Fig. 3 and Extended Data Fig. 4). We describe that the mutant Ror2^{t13}, which lacks the extracellular Ig-like domain, the CRD domain, the transmembrane domain, and the tyrosine kinase domain. However, the mutant displayed no obvious morphological alteration. Therefore, we generate the maternal-zygotic *ror2^{t13}* mutant, which is also viable, fertile and shows a mild phenotype of a slightly upwards bend tail tip, similar but less pronounced as in the *wnt5b/pipetail* mutant zebrafish (Hammerschmidt et al., 1996; Rauch et al., 1997). We hypothesized that the complementary pseudo-kinase, Ror1, may act functionally redundant and indeed, a *ror1^{Crispant}/MZ ror2^{t13}* mutant embryos show many typical features of a Wnt/PCP phenotype, including a wider and shorter axial mesoderm leading to a shorter body axis, malformation of the trunk and tail, and heart defects. The phenotype of the *ror1^{Crispant}/MZ ror2^{t13}* embryo-larvae is similar to the phenotype observed in the zebrafish double mutant *wnt5b/pipetail / wnt11/silberblick* (Quesada-Hernandez et al., 2003; Westfall et al., 2003).*

c. How does loss of *ror2* function affect JNK signaling? The JNK-KTR-mCherry reporter activity should be compared between WT and *ror2* mutant embryos throughout gastrulation to understand the requirement for the endogenous *ror2* activity in JNK signaling and also to identify appropriate stages of development to perform functional experiments.

*We have characterized the JNK reporter in the *ror1^{Crispant}/MZ ror2^{t13}* mutant and found a strong downregulation at mid-gastrulation stages (Fig. 4). Therefore, we used these stages for the JNK signaling reporter assay (Fig. 4 and Extended Data Fig. 6)*

d. In the experiments quantifying JNK activity, nuclei should be labeled with Histone-FP and the ratio of nuclear JNK versus Histone could be used as an additional measure of JNK activity. E.g. how do we know that there is a cell where the arrow points in Fig. 5b? And how the authors could measure the nuclear versus cytoplasmic level of JNK activity in a cell in which nucleus is not visible?

We have described the analysis of the JNK reporter extensively in our previous publications (Brunt et al., 2021; Routledge et al., 2022, eLife). For clarification, we added these references. Furthermore, we added an explanation of the functioning of the reporter (Fig. 4 and Extended Data Fig. 6).

4.9 Expression of the dominant-negative mutants IRSp534K and Cdc42T17N is used to impair formation of cytonemes in the transplanted cell clusters expressing Wnt5b what leads to the conclusion that cytonemes are needed/mediate Wnt5b/Ror2s paracrine signaling and JNK activation. Against this notion, the authors report that ubiquitous expression of IRSp534K in the zebrafish embryos does not interfere with JNK signaling.

This raises the concern that the phenomena observed with transplanted cell clusters expressing Wnt5b/Ror2 do not reflect physiological processes but rather the effects of overexpressed proteins. Impairment of cytonemes in entire embryos should lead to altered JNK activity and cell polarity if cytoneme-mediated Wnt5b/Ror2 paracrine signaling is required for Wnt/PCP regulated gastrulation morphogenesis. Indeed, in these experiments a control clone expressing memGFP does not interfere with JNK signal activation and only clones overexpressing Wnt5b/Ror2 do.

We thank the reviewer for this comment, which gives us the possibility to explain the experimental procedures in more detail. The clones for Wnt5/Ror2 were generated by overexpression of the constructs in 1 out of 32 blastomeres. Ubiquitous expression of IRSp53^{4K} or Cdc42^{T17N} does not alter the localization of the KTR reporter (Fig. 4; Extended Data Fig. 6). Overexpression of Wnt5b/Ror2 in these clones leads to the activation of paracrine JNK signaling in this tissue at this time point. Furthermore, blocking of cytoneme formation in the clones partially reduces paracrine signaling. Impairment of cytonemes in the entire embryo has a severe effect, as embryos usually stop developing after gastrulation. We have clarified this in the text.

A more appropriate or additional experiment would be to transplant cells from embryonic regions that do or do not express these proteins. In zebrafish, like in Drosophila Wnt/PCP signaling and the processes it regulates are exquisitely sensitive to both loss but also gain of function.

*We agree with the reviewer and therefore, we have performed these experiments in Fig. 4p-s. Here we grafted cells from the embryonic margin (which expresses high levels of Wnt5b and Ror2) into the MZ *ror2*^{t13} mutant background (Fig 4). In detail, we found that the embryonic margin cells from MZ *ror2*^{t13} do not alter JNK signal activation around the clone (Fig. 4p). However, WT cells are able to induce JNK*

activation in the neighboring cells (Fig. 4q). Finally, we reduced cytoneme formation by co-expressing IRSp53^{4K}, and we observed that paracrine JNK signaling was significantly reduced (Fig. 4r) as seen in the quantification in Fig. 4s. We have added this information to the text.

4.10 The finding that challenges the classical notion of intercellular signaling whereby a cell competent to respond to a signal produced by a different cell expresses the cognate receptor is that cell receiving cytoneme-mediated Wnt5b/Ror2 complexes do not require Ror2 activity. This conclusion is reached based on the experiments in which a dominant negative form of the Ror2 receptor, the kinase-dead mutant Ror23i (Hikasa et al., 2002) is expressed in receiving cells what blocks JNK signal activation. Yet such cells can activate JNK signaling when cells expressing Wnt5b/Ror2 are transplanted in their environment. It is puzzling why to inactivate Ror2 in these experiments a dominant negative form of the receptor is used given the *ror2* mutant line described in the manuscript? Such experiments should be also performed using the *ror2* mutant line and JNK reporter.

We agree with the reviewer and added the requested experiments to the manuscript (Fig. 4).

4.11 The *ror2* mutant line provides additional opportunities to perform experiments that would alleviate the concern about overexpressing fluorescent Ror2 fusion proteins. When *ror2* phenotypes are analyzed, the dose of mCh-Ror2 that rescues these phenotypes (i.e. physiological dose) would be identified allowing to perform some of the colocalization studies in more physiological setting. Alternatively, although admittedly more difficult, knock-in technologies now afford generating fluorescent fusion proteins in endogenous genes.

*We agree with the reviewer. Therefore, we grafted WT embryonic margin cells (which express physiological concentrations of *ror2*) into the MZ *ror2*^{t13} mutant. These experiments indicate that physiological levels of *ror2* can activate paracrine JNK signaling (Fig. 4p-s). Furthermore, we performed microinjection of 50-200ng of WT Ror2 mRNA into the *ror1*^{Crispant} / MZ *ror2*^{t13} mutant (fig. 3d, Ext. Fig 4c). We find that we can partially rescue the phenotype suggesting that the microinjection experiments were done in a physiological range.*

4.12 When concluding that a receiving cell does not require Wnt5a/b receptor one needs to consider that Ror2 is only one of the receptors binding Wnt5 ligands during zebrafish gastrulation. In fact Ror2 receptors are thought to work with Frizzled receptors. Is the reception of Wnt5b/Ror2 independent of any receptors or could it be mediated by Frizzled, Ptk7, Ryk or other membrane receptors implicated in Wnt/PCP signaling?

We agree with the reviewer, and indeed, also our lab has provided some evidence that Ror2 can work with and without Fzd receptors, here Fzd7 (Brinkmann et al., 2016, JBC). Our data do not exclude that Ror2 acts in concert with Fzd receptors or with other co-receptors. We have added this information to the discussion.

4.13 To probe the role of Vangl2, which is known to be required for mediating JNK signaling, the authors deploy expression of the dominant-negative mutants Vangl210A rather than well-characterized vangl2 zebrafish mutant lines. Such orthogonal experiments should be performed.

We agree with the reviewer and added a data set showing that Wnt5b-Ror2 overexpression clones can activate JNK signaling (Fig. 4h-K), whereas the same clones fail to activate JNK signaling if they are induced in the Vangl2^{m209} mutant background (Fig. 4m-o). These results suggest that – although Wnt5b/Ror2 is overexpressed in the clonal cells – downstream signaling components such as Vangl2 are still essential to trigger the Wnt/PCP signaling cascade in the receiving cells.

4.14 The requirement for the role of the membrane-localized Vangl2 in transducing Wnt5b/Ror2-cytoneme-mediated signalling in the receiving cell is puzzling given that Ror2 is not required in the receiving cells? This amplifies the point #11 that the more typical Wnt/PCP membrane complexes composed of Fz/Vangl/Celsr could be required for Wnt5b/Ror2-cytoneme mediated transfer?

We postulate that the Wnt5b/Ror2 ligand complex is sufficient to recruit kinases such as Ck1δ/ε for the phosphorylation of the downstream effector Vangl2. Therefore, we believe that Vangl2 can be seen as a phosphorylation substrate. We agree that it would be interesting to investigate the role of Fz/Vangl/Celsr. However, we believe this is not the focus of this manuscript and should be addressed in subsequent work.

4.15 Another key conclusion (line 288) “Cytonemal-delivered Wnt5b/Ror2 complexes can block Wnt/b-catenin signaling” is also questionable. Here, small cell clusters expressing Wnt5b, Ror2 or both are transplanted into the zebrafish margin and Wnt/b-catenin pathway activity is assessed using lef1 expression in the transplanted cells and their neighbors. These experiments show that overexpressed Wnt5b/Ror2 CAN interference with b-catenin activity in autocrine and paracrine manner, and the latter is dependent on cytonemes. Surprisingly to address whether ENDOGENOUS genes are involved, Ror23i and Vangl210a mutant protein overexpression is deployed. As indicated above, a requirement for the endogenous components should be tested using the mutant lines: is lef1 expression altered in ror2 mutants the authors generated? Would cell clusters transplanted from the equivalent embryonic positions from ror2 mutants to the margin of wild-type embryos have such effects.

*We agree with the reviewer and reworded that sentence carefully. We demonstrate that *lef1* expression can be reduced upon *Wnt5b/ror2* signaling and *lef1* expression is broadened in *ror1^{crispant} / MZ ror2^{t13}* mutants (ext. Fig 4j-k). However, we are not able to say if all aspects of *Wnt/β-catenin* signaling are inhibited by *Wnt5b/Ror2*.*

4.16 It is stated in the summary that “cytoneme-dependent spreading of active *Wnt5b/Ror2* affects convergence and extension in the zebrafish gastrula”. These experiments are also problematic as they test the ability of cell clusters overexpressing *Wnt5* and *Ror2* rather than using mutant cells. The ability of many overexpressed proteins to interference with convergence and extension is known

*We agree with the reviewer. In this analysis, we are interested in the paracrine signaling activity of *Wnt5b/Ror2*. Therefore, it is even more surprising that blockage of cytoneme formation by co-expression of *IRSp53^{4k}* reduces the C&E alteration and partially rescues this phenotype.*

Additional points:

- Line 284 “This data suggests ” data is a plural noun.

Corrected.

- The figure numbers should be provided on the figures – their absence made reviewing the manuscript rather cumbersome.

OK

Reviewer Reports on the First Revision:

Referee expertise:

Referee #1: Wnt signalling

Referee #2: Cytonemes, cell-cell signalling

Referee #3: Ligand-receptor interactions, Wnt, molecular biology

Referee #4: Zebrafish, Wnt/PCP signalling

Referees' comments:

Referee #1 (Remarks to the Author):

Review Zhang et al

Cytoneme-mediated transport of active Wnt5b/Ror2 complexes in zebrafish.

This paper presents a new view of ligand-receptor transport between cells that goes counter all the existing cell-biological knowledge of how growth factors activate target cells; namely that specific receptors on the target cells are not necessary. Because of provocative nature of the claims, the reviewers including myself expressed numerous concerns.

In the rebuttal, the authors state that "All referees' comments and suggestions have been addressed". I am afraid that this is not the case for my own concerns.

In my original review, I had asked for a biochemical experiment to support the paper's claims, by purifying the complex between the ligand (Wnt) and the receptor Ror2, and add this to cells. Those cells should then be known for possible expression of all types of Wnt receptors, Fzd LRP, not just Ror2. This may be difficult to accomplish but not impossible, given the fact that several Wnts and receptors have been purified to generate complexes for structural analysis. Cells can easily be characterized for receptor gene expression. The authors reply to this question by showing several indirect experiments, none of them biochemical in nature.

In the absence of such experiments, I believe that the authors should be much more cautious in their interpretation. Not use the rather sweeping terms used in the abstract, introduction and discussion. I suggest that the authors make more efforts to come to their dramatic conclusions.

Referee #2 (Remarks to the Author):

In the revised manuscript, the authors have addressed all my concerns/suggestions with several

additional experiments. The results provide strong evidence that cytonemes can deliver pre-formed ligand-receptor complexes from source to recipient cells, and reception of this receptor-ligand complex in the neighboring cells is required for signaling and morphogenetic outputs.

Some minor comments/suggestions on the new text:

1) References would be great for Lines 48-50 (for cytoneme's CDC42-dependence and control through producing cells and extracellular space) and line 91 (for MyoX & N-WASP in cytonemes).

2) Solid evidence for Wnt5-Ror endocytosis and its need for signaling. Yet, clarification in the text is required. Line 125-126: it is unclear whether the source or the recipient cells overexpress Dyn2KA. Dyn can also directly/indirectly regulate exocytosis/externalization. So, if Dyn2-K44A is present in the source, a possibility is that the protein externalization is affected, and this is why there is a reduction in recipient uptake. Also, a comment on the Ror2 localization in the Rab-5-marked early endosome in the source might be important. Does the Ror2-Wnt5 complex get internalized in the source as well? If so, why Dy2K44A in the source does not affect the no. of endosome localized Ror2 puncta?

3) Scale bars missing in several Fig panels, including extended Figures. Box plots with <10 samples might need data points (e.g., 2m,n, mGFP control graph, but other graphs have >10 data points). Fig 2o, Y-axis label (%) missing. Fig. 4l bar graph - error bar might be S.D.?

Referee #3 (Remarks to the Author):

A. Summary of the key results

The authors promote a provocative mode of receptor-ligand complex transfer from the cytoneme to a receiving cell, where an activated ROR/WNT complex transfers signaling over long distances.

B. Originality and significance: if not novel, please include reference

The message is indeed novel and the significance is high if the concept of "handing over" a cytonemal vesicle including an activated receptor-ligand complex can indeed be conceptualized experimentally.

C. Data & methodology: validity of approach, quality of data, quality of presentation

The methodology is suitable but reaches limitations in defining the mechanism of vesicle fusion and signal complex transfer from the cytoneme to the receiving cell. Data are otherwise well presented and annotations of constructs has improved during revision.

D. Appropriate use of statistics and treatment of uncertainties

Yes, appropriate.

E. Conclusions: robustness, validity, reliability

I fear that the data and the summary scheme in Fig. 1k are not in agreement with how membrane vesicles are transferred between cells.

F. Suggested improvements: experiments, data for possible revision

The mechanism of vesicle transfer must be experimentally addressed. Furthermore, the aspect of long distance signaling to activate the JNK reporter in the absence of (visible) cytonemes must be addressed.

G. References: appropriate credit to previous work?

The underlying mechanism of WNT signaling towards JNK remain obscure, despite the reference to papers (Ref 9 and 10). The authors avoid mentioning FZDs constructively in their work despite a central role of FZDs in WNT/ROR/JNK signaling.

H. Clarity and context: lucidity of abstract/summary, appropriateness of abstract, introduction and conclusions

Yes, clear.

DETAILED COMMENTS

While I am happy to disagree in some points with the authors, some criticism remains. The authors addressed some of my comments satisfactorily, however at some points I have the feeling that they avoided a straightforward answer.

1. (regarding my initial comment 3.13). The figure 1k illustrates the proposed concept and underlines at the same time that the presented findings are not supporting the basic concept. While the experiments with the sec-Vhh are indeed elegant, proving that the WNT-binding domain of ROR2 is on the outside of the cytoneme, it appears energetically disfavored that the cytoneme-derived vesicle would fuse with the plasma membrane of the receiving cell and that that complex would then be internalized by a dynamin-mediated endocytosis. Rather, the ROR2/WNT-containing vesicle released from the cytoneme as a outside-out vesicle – additional comment: is this mechanism, which cannot be a classical exocytosis, common knowledge? How are parts of the cytoneme released?) should be engulfed by the membrane of the receiving cells and endocytosed as a whole. Furthermore, in this endocytosed vesicle ending up in the early endosomes of the receiving cell with the intracellular domain of ROR pointing to the lumen of the endocytosed vesicle, emanating signal transduction is conceptually difficult to imagine. As pointed out in my initial review, the whole concept of receptor/signal transmission is not completely in agreement with current concepts of membrane biology. I cannot get my head around about the receptor orientation in the membrane transport processes from cytoneme to receiving endosome. The authors refer to the receptor complex movement from cytoneme to receiving cell as a “handover” – the mechanism remains fully obscure.

Are free cytoneme-derived ROR2/WNT-positive vesicles visible in the experiments?

While I might be wrong, I am not aware of a membrane fusion mechanism that can handle the fusion of a vesicle with the plasma membrane as depicted in Fig. 1k. Thermodynamically this process of spontaneous vesicle fusion with the plasmamembrane on the extracellular side is highly unfavorable. Ingestion of an intact vesicle is, however, common, dynamin dependent and not experimentally excluded in this manuscript, but the latter concept would render the internalized ROR2/WNT complex incapable of signaling because it is engulfed in an endosome. – These questions need to be clarified.

Also Suppl Video 1 that is claimed to show the handing over of a ROR2/WNT-containing piece of the cytoneme is ambiguous and could be explained by a shift in focus level instead of total retraction of the cytoneme. The example of the cell that undergoes mitosis (which probably affects cytoneme dynamics) is unfortunate. The particle that is ROR2/WNT positive does not look to be handed over to the target cell, it remains floating in between the cells. I wonder what happens after the nine seconds at which the video stopped.

Also in video 2, which aims to control for the effect of the JNK inhibitor on cytoneme formation, the

green particle in the center of the image goes back and forth in between two cells until a third cell intercalates in between the cytoneme-connected cells and the fate of the green particle remains uncertain.

2. Comment 3.3: This point from the initial review remains. The authors have not mentioned the main WNT receptors, the FZDs in the introduction. While I understand the focus on ROR2, it is awkward to not introduce the main receptors to WNT proteins, which in fact might be part of the observed phenomena, even though it is not experimentally addressed. The authors basically glance over their existence, which is counterintuitive; especially since the reference #10 (Nishita M et al 2010 – defines an important role of FZD in ROR-DVL-JNK signaling).

3. In the discussion of residence time and affinity, where I can disagreeingly accept the author's arguments, the authors refused to answer the question about residence time and the previous question, where the ROR2/WNT complex is formed. The membranous staining for ROR2 and WNT of the WNT producing cell argues that the receptor complex is not formed at the tip of the cytoneme, where the authors argue that the relatively low affinity would be compensated by high local concentrations of the ligand, but that the complex is formed at the cell surface prior to cytonemal translocation. The ROR2/WNT complex is transported over long distances through areas of maximal ligand dilution and low local WNT concentration, which can only mean that the residence time is long, which would reflect on a relatively high affinity, potentially higher than the values that were calculated. This comment is not meant to impose my argumentation, I just would like to understand the system under the parameters described and measured.

4. Regarding my previous comment 3.30: Upon my question about long distance effects on the JNK reporter the authors answer and argue with cytoneme length of 100 μm . This is a good point indeed, however, cytonemes are not visible in the preparation despite the intense green label of the providing cells. How do the authors then explain long distance, cytoneme-mediated effect over >80 μm in the absence of cytonemes? There should be a network of cytonemes established to all individual cells to reach the effect shown if it is indeed cytoneme- and not soluble factor- or "any other alternative mechanism"-mediated.

5. My comment 3.32 remains, at least until the paradigm of cytoneme tip endocytosis and the orientation/localization of a putatively active ROR is clarified. Without addressing the mode of vesicle internalization/"hand over" further investigation of signaling in complex tissues remains confusing and my previously mentioned caveat of studying a soluble factor is back on the table. Both IRSp53 and CDC42 are known to regulate diverse cell biological phenomenon in addition to cytoneme formation and appear as a relatively unselective tools especially in complex tissues. CDC42 e.g regulates exocytosis and thus the control experiment with CDC42 aimed to specifically affect cytoneme formation could also inhibit exocytosis of a hypothetical released factor X. Most importantly, the observed phenomena in Fig. 5 can by no means be causally related to the cytonemal transfer of an active ROR2/WNT complex and the studied phenomenon. I am doubtful that the use of IRSp53 4K as an inhibitor of filopodia and cytonemes as well as other membrane protrusions unambiguously links the observed gross alterations in the embryo to cytonemal ROR2/WNT signaling or that the use of these tools could distinguish between non cytonemal and cytonemal WNT signaling through RORs or other receptors; especially since the confusion of the

membrane “hand over” phenomenon is not clarified. The same argumentation is valid for Extended data Fig 7, 8

Referee #4 (Remarks to the Author):

In this revised manuscript the authors have done a very significant effort to address many of the questions and concerns raised by the reviewers.

While the manuscript is significantly improved, this reviewer is still concerned about the interpretation and biological significance of the Wnt5a/b/Ror2 cytonemal transport and Wnt/PCP signaling in cells lacking Ror2 receptors. The authors continue to refer to these protrusions Wnt/PCP cytonemes (e.g. line 233) and argue that these cytonemes transport Wnt5/Ror2 complexes between cells to mediate convergence and extension movements in which Wnt5a/b have been previously implicated by many studies. Indeed, MZ Ror2 mutants combined with Ror1 crispants manifest stronger morphological defects consistent with impaired convergence and extension movements. This new experimental evidence is appreciated. However, the *in vivo* observations are carried out in cells of the blastoderm margin in embryos before convergence and extension movements that shape the notochord and other mesodermal tissues are initiated.

The key “dogma challenging” conclusion is that Wnt5/Ror2 complexes are transported from a signaling to receiving cell in a receptor independent fashion. Here, the authors describe experiments (line 244), in which generated small clones of Wnt5b-GFP in zebrafish embryos and analyzed the GFP puncta around the clone. They observed that “the number of puncta on the membrane of the producing cells and in the neighboring cells is significantly reduced in *ror1*crispant / MZ *ror2t13* (Fig. 3 j-l). Consistently, this phenotype is reversed if the Wnt5b-GFP expressing clones co-express Ror2.” However, Figure 3 j-l only shows that Ror2 promotes spread of Wnt5b-GFP in the sending cells and when overexpressed in WT background more Wnt5b-GFP puncta are generated and spread farther. Could not find the experiments comparing the number of Wntb-GFP puncta and their spread from a Wnt-GFP clone in WT versus *ror1*crispant / MZ *ror2t13* host.

Moreover, based on these experiments the authors posit “ Wnt/PCP signaling induces long Wnt5b/Ror2-bearing cytonemes, which facilitates Wnt5b (and Ror2) spreading.” There is no evidence for Wnt/PCP signaling here, but that Wnt5b/Ror2 co-expression induces long Wnt5b/Ror2 cytonemes ..”.

An interesting set of new experiments explores how an ectopic clone expressing Wnt5b/Ror2 can influence convergence and extension of host WT embryos. These experiments demonstrate that such paracrine ectopic signaling (via cytoneme transferred Wnt5/Ror2) can impair notochord cell polarity and convergence and extension. An outstanding question is how important such paracrine signaling is for normal convergence and extension? The key experiment here would be to ask if ectopic Wnt5/Ror2 clones can RESCUE notochord convergence and extension phenotypes in Ror1/Ror2 mutants the authors generated.

Altogether, this work provides compelling evidence for Wnt5b/Ror2 cytonemes mediating paracrine signaling in receiving cells in Ror2 independent fashion, and modulating JNK signaling via Vangl2-

dependent fashion. This is very interesting work that will be of interest to a broad audience and thus warrants publication. However, the authors conclusions about “pivotal” role of this type of signaling to the normal gastrulation processes should be toned down.

Author Rebuttals to First Revision:

Referees' comments:

Referee #1 (Remarks to the Author):

In my original review, I had asked for a biochemical experiment to support the paper's claims, by purifying the complex between the ligand (Wnt) and the receptor Ror2, and add this to cells. Those cells should then be known for possible expression of all types of Wnt receptors, Fzd LRP, not just Ror2. This may be difficult to accomplish but not impossible, given the fact that several Wnts and receptors have been purified to generate complexes for structural analysis. Cells can easily be characterized for receptor gene expression. The authors reply to this question by showing several indirect experiments, none of them biochemical in nature.

In the absence of such experiments, I believe that the authors should be much more cautious in their interpretation. Not use the rather sweeping terms used in the abstract, introduction and discussion. I suggest that the authors make more efforts to come to their dramatic conclusions.

We thank the reviewer for their comments and appreciate the suggestions to purify a complex of Wnt5b/Ror2 and "add" this to the cell. The reviewer agrees that such an experiment is far from easy, and unfortunately, my lab does not have the expertise to perform the suggested experiment. Even if it would be technically doable, we believe that a membrane-spanning complex becomes unstable upon extraction from lipid membranes and would lose functionality (as demonstrated several times in the literature). That is why we have decided not to carry out this experiment. In agreement, with the reviewer, we have toned down the text and indicated the points in the text.

Referee #2 (Remarks to the Author):

In the revised manuscript, the authors have addressed all my concerns/suggestions with several additional experiments. The results provide strong evidence that cytonemes can deliver pre-formed ligand-receptor complexes from source to recipient cells, and reception of this receptor-ligand complex in the neighboring cells is required for signaling and morphogenetic outputs.

We thank this reviewer for their positive feedback.

Some minor comments/suggestions on the new text:

1) References would be great for Lines 48-50 (for cytoneme's CDC42-dependence and control through producing cells and extracellular space) and line 91 (for MyoX & N-WASP in cytonemes).

We have added references showing the dependency of cytonemes on Cdc42, control of their directionality by the source cells, as well as their co-localisation with MyoX and N-WASP.

2) Solid evidence for Wnt5-Ror endocytosis and its need for signaling. Yet, clarification in the text is required. Line 125-126: it is unclear whether the source or the recipient cells overexpress Dyn2KA. Dyn can also directly/indirectly regulate exocytosis/externalization. So, if Dyn2-K44A is present in the source, a possibility is that the protein externalization is affected, and this is why there is a reduction in recipient uptake. Also, a comment on the Ror2 localization in the Rab-5-marked early endosome in the source might be important. Does the Ror2-Wnt5 complex get internalized in the source as well? If so, why Dy2K44A in the source does not affect the no. of endosome localized Ror2 puncta?

We added the requested information to the text. For clarification, we measure the number of clusters in the adjacent cells. We did not quantify the number of vesicles in the source cell.

3) Scale bars missing in several Fig panels, including extended Figures. Box plots with <10 samples might need data points (e.g., 2m,n, mGFP control graph, but other graphs have >10 data points). Fig 2o, Y-axis label (%) missing. Fig. 4l bar graph - error bar might be S.D.?

We have fixed all of the points mentioned by the reviewer.

Referee #3 (Remarks to the Author):

A. Summary of the key results

The authors promote a provocative mode of receptor-ligand complex transfer from the cytoneme to a receiving cell, where an activated ROR/WNT complex transfers signaling over long distances.

B. Originality and significance: if not novel, please include reference

The message is indeed novel and the significance is high if the concept of “handing over” a cytonemal vesicle including an activated receptor-ligand complex can indeed be conceptualized experimentally.

C. Data & methodology: validity of approach, quality of data, quality of presentation

The methodology is suitable but reaches limitations in defining the mechanism of vesicle fusion and signal complex transfer from the cytoneme to the receiving cell. Data are otherwise well presented and annotations of constructs has improved during revision.

D. Appropriate use of statistics and treatment of uncertainties

Yes, appropriate.

E. Conclusions: robustness, validity, reliability

I fear that the data and the summary scheme in Fig. 1k are not in agreement with how membrane vesicles are transferred between cells.

F. Suggested improvements: experiments, data for possible revision

The mechanism of vesicle transfer must be experimentally addressed. Furthermore, the aspect of long distance signaling to activate the JNK reporter in the absence of (visible) cytonemes must be addressed.

G. References: appropriate credit to previous work?

The underlying mechanism of WNT signaling towards JNK remain obscure, despite the reference to papers (Ref 9 and 10). The authors avoid mentioning FZDs constructively in their work despite a central role of FZDs in WNT/ROR/JNK signaling.

H. Clarity and context: lucidity of abstract/summary, appropriateness of abstract, introduction and conclusions

Yes, clear.

DETAILED COMMENTS

While I am happy to disagree in some points with the authors, some criticism remains. The authors addressed some of my comments satisfactorily, however at some points I have the feeling that they avoided a straightforward answer.

1. (regarding my initial comment 3.13). The figure 1k illustrates the proposed concept and underlines at the same time that the presented findings are not supporting the basic concept. While the experiments with the sec-Vhh are indeed elegant, proving that the WNT-binding domain of ROR2 is

on the outside of the cytoneme, it appears energetically disfavored that the cytoneme-derived vesicle would fuse with the plasma membrane of the receiving cell and that that complex would then be internalized by a dynamin-mediated endocytosis. Rather, the ROR2/WNT-containing vesicle released from the cytoneme as a outside-out vesicle – additional comment: is this mechanism, which cannot be a classical exocytosis, common knowledge? How are parts of the cytoneme released?) should be engulfed by the membrane of the receiving cells and endocytosed as a whole. Furthermore, in this endocytosed vesicle ending up in the early endosomes of the receiving cell with the intracellular domain of ROR pointing to the lumen of the endocytosed vesicle, emanating signal transduction is conceptually difficult to imagine. As pointed out in my initial review, the whole concept of receptor/signal transmission is not completely in agreement with current concepts of membrane biology. I cannot get my head around about the receptor orientation in the membrane transport processes from cytoneme to receiving endosome. The authors refer to the receptor complex movement from cytoneme to receiving cell as a “handover” – the mechanism remains fully obscure.

To support our idea of the mode of transport, we have provided further experimental data. First, we investigate the transport of Ror2-mcherry with high-resolution and find all stages of transport described (new Ext Fig. 1g, Fig. 1K). Then we use Lattice-SIM-based super-resolution imaging to analyze the transport of Wnt5b-GFP, Ror2-GFP, and GFP-Ror2 (Ext Fig. 1g-i). We treat these cell cultures with a VHH-mCherry nanobody to visualize the orientation of the N-terminal or C-terminal GFP on the Ror2 about the plasma membrane. In summary, we show that Wnt5b/Ror2 cytonemes, contact cells (1), and leave a Ror2-vesicle behind (2). Furthermore, we find that the vesicle is eventually internalized (3). We hope that these findings also address the second concern of the reviewer regarding the orientation of the receptor. According to our model (Fig. 1k), endocytosis of the receptor from the target cell membrane will place the signalling active inner domain of the receptor outside of the endosome, which could explain how Ror2 can continue to signal into the target cell.

Are free cytoneme-derived ROR2/WNT-positive vesicles visible in the experiments?

While I might be wrong, I am not aware of a membrane fusion mechanism that can handle the fusion of a vesicle with the plasma membrane as depicted in Fig. 1k. Thermodynamically this process of spontaneous vesicle fusion with the plasmamembrane on the extracellular side is highly unfavorable. Ingestion of an intact vesicle is, however, common, dynamin dependent and not experimentally excluded in this manuscript, but the latter concept would render the internalized ROR2/WNT complex incapable of signaling because it is engulfed in an endosome. – These questions need to be clarified.

We respectfully disagree with the opinion of the reviewer. There are many examples of how extracellular vesicles (e.g. exosomes or microvesicles) can fuse with the plasma membrane of target cells (reviewed in Meldolesi et al., 2018, Curr Biol). For example, Syncytin1/2 has been implicated in

the fusion events of placenta-derived exosomes with the plasma membrane (Yu et al., 2002, J Cell Biol). Such a mechanism serves as a general, non-selective means of extracellular vesicle fusion with the plasma membranes of their recipient cells, as observed in many following works (Prada et al. 2016, Biotechnology; Mulcahy et al., 2014; J. Extracell. Ves.; Chen and Brigstock, 2016, FEBS Lett.; French et al., 2017, Semin. Cell Dev. Biol). Cytoneme-derived vesicle fusion with the plasma membrane – as described in this work - may serve as a further example for such a fusion event.

Also Suppl Video 1 that is claimed to show the handing over of a ROR2/WNT-containing piece of the cytoneme is ambiguous and could be explained by a shift in focus level instead of total retraction of the cytoneme. The example of the cell that undergoes mitosis (which probably affects cytoneme dynamics) is unfortunate. The particle that is ROR2/WNT positive does not look to be handed over to the target cell, it remains floating in between the cells. I wonder what happens after the nine seconds at which the video stopped.

Also in video 2, which aims to control for the effect of the JNK inhibitor on cytoneme formation, the green particle in the center of the image goes back and forth in between two cells until a third cell intercalates in between the cytoneme-connected cells and the fate of the green particle remains uncertain.

We respectfully disagree with this point. The video shows a triple-stained cluster of Wnt5/Ror2 and membrane markers. The zebrafish is a dynamic system, and the cells migrate and divide quickly. Therefore, it is challenging to visualize the entire process. We therefore provide data in cell culture suggesting such a transport mechanism (Fig. 1; Extended Data Figure 1).

2. Comment 3.3: This point from the initial review remains. The authors have not mentioned the main WNT receptors, the FZDs in the introduction. While I understand the focus on ROR2, it is awkward to not introduce the main receptors to WNT proteins, which in fact might be part of the observed phenomena, even though it is not experimentally addressed. The authors basically glance over their existence, which is counterintuitive; especially since the reference #10 (Nishita M et al 2010 – defines an important role of FZD in ROR-DVL-JNK signaling).

Although we focus on Wnt5b/Ror2 interaction, we added the information about Fzd to the text.

3. In the discussion of residence time and affinity, where I can disagreeingly accept the author's arguments, the authors refused to answer the question about residence time and the previous question, where the ROR2/WNT complex is formed. The membranous staining for ROR2 and WNT of the WNT producing cell argues that the receptor complex is not formed at the tip of the cytoneme, where the authors argue that the relatively low affinity would be compensated by high local concentrations of the ligand, but that the complex is formed at the cell surface prior to cytonemal

translocation. The ROR2/WNT complex is transported over long distances through areas of maximal ligand dilution and low local WNT concentration, which can only mean that the residence time is long, which would reflect on a relatively high affinity, potentially higher than the values that were calculated. This comment is not meant to impose my argumentation, I just would like to understand the system under the parameters described and measured.

Our data indicate that the Wnt5b/Ror2 complex is formed at the membrane of the producing cell prior cytoneme formation, as shown in Fig. 1f, the time lapse analysis Fig. 1m, and as measured with in vivo FLIM-FRET and in vivo FCCS (Fig. 2 e, f). The residence time was extracted from time-lapse experiments.

4. Regarding my previous comment 3.30: Upon my question about long distance effects on the JNK reporter the authors answer and argue with cytoneme length of 100 μm . This is a good point indeed, however, cytonemes are not visible in the preparation despite the intense green label of the providing cells. How do the authors then explain long distance, cytoneme-mediated effect over >80 μm in the absence of cytonemes? There should be a network of cytonemes established to all individual cells to reach the effect shown if it is indeed cytoneme- and not soluble factor- or "any other alternative mechanism"-mediated.

We revisited the data set on the JNK signaling analysis and added data showing clones of Wnt5b/Ror2 expressing cells, which form cytonemes with a length of ca. 80 μm (Fig. 4b).

5. My comment 3.32 remains, at least until the paradigm of cytoneme tip endocytosis and the orientation/localization of a putatively active ROR is clarified. Without addressing the mode of vesicle internalization/"hand over" further investigation of signaling in complex tissues remains confusing and my previously mentioned caveat of studying a soluble factor is back on the table. Both IRSp53 and CDC42 are known to regulate diverse cell biological phenomenon in addition to cytoneme formation and appear as a relatively unselective tools especially in complex tissues. CDC42 e.g regulates exocytosis and thus the control experiment with CDC42 aimed to specifically affect cytoneme formation could also inhibit exocytosis of a hypothetical released factor X.

We respectfully disagree with the comments of the reviewer. Cdc42 has multiple roles in the cells, including endocytosis and exocytosis. However, one of the main functions of this small Rho GTPase is the regulation of actin polymerization leading to filopodia formation (Nobes et al., 1995, Cell). We have shown in numerous publications that blockage of Cdc42 (via the mutant or via chemical inhibition) is an effective tool for reducing cytoneme formation. However, we agree that no tool is perfect. Therefore, to complement the experiments, we use IRSp53^{4k}. This bar protein is a key protein in filopodia formation (Ahmed et al., 2010, Cell Dev Biol; Krugmann et al., 2001, Curr Biol). Irsp53 protein

contains an Inverse-Bin-Amphiphysin-Rvs (I-BAR) domain (Mattila et al., 2007, J Cell Biol) that promotes membrane deformation and includes actin binding sites (Yamagishi et al., 2004, J Biol Chem). IRSp53 is essential for filopodia formation via a Cdc42-mediated N-WASP interaction (Oh et al., 2013, Biochem Journal). The dominant-negative form of the protein harbors mutations in the 4 actin-bundling sites (Millard et al., 2005, EMBO J; Lim et al., 2008, J Biochem) within the I-BAR domain. Over-expression of the IRSp53^{4k} changes filopodia to microspikes, which collapse immediately. IRSp53 function has been extensively studied in zebrafish germ cells (Meyen et al., 2015, eLife). Finally, we have shown that IRSp53 is essential for Wnt cytoneme formation in zebrafish gastrulation (Fig. 3g, and Stanganello et al., 2015; Nat Comms; Mattes et al., 2018, eLife; Brunt et al., 2021, Nat Comms), and also in human gastric cancer cell lines (Routledge et al., 2022, eLife) as well as in cancer-associated fibroblasts (Rogers et al, 2022, bioRxiv). Therefore, we believe perturbing Cdc42 or IRSp53 function can lead to meaningful conclusions about the function of cytonemes in various contexts.

Most importantly, the observed phenomena in Fig. 5 can by no means be causally related to the cytonemal transfer of an active ROR2/WNT complex and the studied phenomenon. I am doubtful that the use of IRSp53 4K as an inhibitor of filopodia and cytonemes as well as other membrane protrusions unambiguously links the observed gross alterations in the embryo to cytonemal ROR2/WNT signaling or that the use of these tools could distinguish between non cytonemal and cytonemal WNT signaling through RORs or other receptors; especially since the confusion of the membrane “hand over” phenomenon is not clarified. The same argumentation is valid for Extended data Fig 7, 8

To strengthen the argument that Ror2 can be provided via cytonemes, we included new experiments in Fig. 5. Specifically, we induced Ror2 clones in the double ror1/ror2^{Crispant} background and find that we can partially rescue convergence & extension as well as the cell polarity. In support of our concept, we find that blockage of cytoneme formation via co-expression of IRSp53^{4k} leads to a cell population that seems to be unable to converge to the midline (white arrows), suggesting that these cells lack proper Wnt/PCP signaling.

Referee #4 (Remarks to the Author):

In this revised manuscript the authors have done a very significant effort to address many of the questions and concerns raised by the reviewers.

While the manuscript is significantly improved, this reviewer is still concerned about the interpretation and biological significance of the Wnt5a/b/Ror2 cytonemal transport and Wnt/PCP signaling in cells lacking Ror2 receptors. The authors continue to refer to these protrusions Wnt/PCP cytonemes (e.g. line 233) and argue that these cytonemes transport Wnt5/Ror2 complexes between cells to mediate convergence and extension movements in which Wnt5a/b have been previously implicated by many

studies. Indeed, MZ Ror2 mutants combined with Ror1 crispants manifest stronger morphological defects consistent with impaired convergence and extension movements. This new experimental evidence is appreciated. However, the *in vivo* observations are carried out in cells of the blastoderm margin in embryos before convergence and extension movements that shape the notochord and other mesodermal tissues are initiated.

We thank the reviewer for the encouraging comments.

The key “dogma challenging” conclusion is that Wnt5/Ror2 complexes are transported from a signaling to receiving cell in a receptor independent fashion. Here, the authors describe experiments (line 244), in which generated small clones of Wnt5b-GFP in zebrafish embryos and analyzed the GFP puncta around the clone. They observed that “the number of puncta on the membrane of the producing cells and in the neighboring cells is significantly reduced in *ror1*crispant / MZ *ror2*^{t13} (Fig. 3 j-l). Consistently, this phenotype is reversed if the Wnt5b-GFP expressing clones co-express Ror2.” However, Figure 3 j-l only shows that Ror2 promotes spread of Wnt5b-GFP in the sending cells and when overexpressed in WT background more Wnt5b-GFP puncta are generated and spread farther. Could not find the experiments comparing the number of Wntb-GFP puncta and their spread from a Wnt-GFP clone in WT versus *ror1*crispant / MZ *ror2*^{t13} host.

*We are sorry that the reviewer was unable to find this data set. The quantification of the Wnt puncta and their spread in the *ror1*crispant / MZ *ror2*^{t13} host is quantified in Fig. 3k/i.*

Moreover, based on these experiments, the authors posit “ Wnt/PCP signaling induces long Wnt5b/Ror2-bearing cytonemes, which facilitates Wnt5b (and Ror2) spreading.” There is no evidence for Wnt/PCP signaling here, but that Wnt5b/Ror2 co-expression induces long Wnt5b/Ror2 cytonemes ..”.

We agree with the reviewer and revisited the data set on the JNK signalling and show an image of a clone of Wnt5b/Ror2 expressing cells, which forms a cytoneme with a length of ca. 80µm (Fig. 4b).

An interesting set of new experiments explores how an ectopic clone expressing Wnt5b/Ror2 can influence convergence and extension of host WT embryos. These experiments demonstrate that such paracrine ectopic signalling (via cytoneme transferred Wnt5/Ror2) can impair notochord cell polarity and convergence and extension. An outstanding question is how important such paracrine signaling is for normal convergence and extension. The key experiment here would be to ask if ectopic Wnt5/Ror2 clones can RESCUE notochord convergence and extension phenotypes in *Ror1/Ror2* mutants the authors generated.

We included these suggested experiments in Fig. 5. Specifically, we induced Ror2 clones in the ror1/ror2^{Crispant} background and find that we can partially rescue convergence & extension and the cell shape. In support of our concept, we find that blockage of cytoneme formation via co-expression of IRSp53-4k leads to a cell population which seems to be unable to converge to the midline (white arrows), suggesting that these cells lack Wnt/PCP signaling.

Altogether, this work provides compelling evidence for Wnt5b/Ror2 cytonemes mediating paracrine signaling in receiving cells in Ror2 independent fashion, and modulating JNK signaling via Vangl2-dependent fashion. This is very interesting work that will be of interest to a broad audience and thus warrants publication. However, the authors conclusions about “pivotal” role of this type of signaling to the normal gastrulation processes should be toned down.

We thank this reviewer for their kind words and toned down the wording throughout the text.

Reviewer Reports on the Second Revision:

Referees' comments:

Referee #3 (Remarks to the Author):

The authors have made substantial efforts to clarify my criticism. The experimental data support the proposed model of signaling.

I have some minor comments:

1. Regarding the argumentation about residence time and affinity throughout the review process the authors contradict themselves previously lifting the point of high local WNT concentrations on the cytoneme but now indicating that the WNT/receptor complex is formed at the plasma membrane, where WNT concentration is lower. This argumentation needs to be continued in future work and reaches here the limits of what is technically possible. I still feel that the author should put the low affinity (KD of hundreds of nanomoles) into context and discuss this value in relation to the low nanomolar affinities shown for other Wnt receptors. The authors call 229 nM a "high binding affinity" lane 162/page 6 (and lane 200/page 7). High compared to what? Obviously, as the authors discuss, the ligand-receptor affinity determinations differ from endogenous/in vivo systems to cell systems, however, ligand-receptor affinity turns often out to be higher in systems with endogenously or low expressed receptors compared to overexpressing systems. While I understand that this is most likely a technological issue, presenting absolute affinity values is always a system-intrinsic analysis.
2. Please add cytonemes in the scheme to Fig. 4a that would represent what is actually proposed.

With kind regards,
Gunnar Schulte

Referee #4 (Remarks to the Author):

In this second revision of the manuscript "Cytoneme-mediated transport of active Wnt5b/Ror2 complexes in zebrafish" Scholpp and collaborators have performed additional experiments to address my key concern about the biological significance of the proposed paracrine Wnt5b/Ror2 activity by cytoneme mediated transfer of Wnt5b/Ror2 complexes independent of Ror receptors in receiving cells.

They extended their previous experiments demonstrating that an ectopic cell population expressing Wnt5b/Ror2 placed in lateral plate mesoderm not only can impair mediolateral cell polarization and notochord C&E in WT embryo and do so in cytoneme-dependent manner (impaired by co-expression of IRSp53) but also can RESCUE mediolateral cell polarization and notochord C&E in embryos deficient in Ror1/Ror2. I appreciate some of the concerns of Reviewer #3 about the precise cell biological and molecular mechanisms of how Wnt5b/Ror2 complexes/vesicles are "handed over" to receiving cells. However, the biological phenomenon and mechanism of Wnt5b/Ror intercellular

communication described by the authors is novel and opens new areas of investigation. I am supportive of publishing this work without further delay.